# Entropic Optimal Transport between Unbalanced Gaussian Measures has a Closed Form

**Hicham Janati**
Inria Saclay
Paris-Saclay, France
hicham.janati@inria.fr

**Boris Muzellec**
ENSAE,
Paris-Saclay, France
boris.muzellec@ensae.fr

**Gabriel Peyré**
CNRS and ENS, PSL University
Paris, France
gabriel.peyre@ens.fr

**Marco Cuturi**
Google Brain, ENSAE
Paris Saclay, France
cuturi@google.com

## Abstract

Although optimal transport (OT) problems admit closed form solutions in a very few notable cases, e.g. in 1D or between Gaussians, these closed forms have proved extremely fecund for practitioners to define tools inspired from the OT geometry. On the other hand, the numerical resolution of OT problems using entropic regularization has given rise to many applications, but because there are no known closed-form solutions for entropic regularized OT problems, these approaches are mostly algorithmic, not informed by elegant closed forms. In this paper, we propose to fill the void at the intersection between these two schools of thought in OT by proving that the entropy-regularized optimal transport problem between two Gaussian measures admits a closed form. Contrary to the unregularized case, for which the explicit form is given by the Wasserstein-Bures distance, the closed form we obtain is differentiable everywhere, even for Gaussians with degenerate covariance matrices. We obtain this closed form solution by solving the fixed-point equation behind Sinkhorn's algorithm, the default method for computing entropic regularized OT. Remarkably, this approach extends to the generalized *unbalanced* case — where Gaussian measures are scaled by positive constants. This extension leads to a closed form expression for unbalanced Gaussians as well, and highlights the mass transportation / destruction trade-off seen in unbalanced optimal transport. Moreover, in both settings, we show that the optimal transportation plans are (scaled) Gaussians and provide analytical formulas of their parameters. These formulas constitute the first non-trivial closed forms for entropy-regularized optimal transport, thus providing a ground truth for the analysis of entropic OT and Sinkhorn's algorithm.

## 1 Introduction

Optimal transport (OT) theory [48, 20] has recently inspired several works in data science, where dealing with and comparing probability distributions, and more generally positive measures, is an important staple (see [39] and references therein). For these applications of OT to be successful, a belief now widely shared in the community is that some form of regularization is needed for OT to be both scalable and avoid the curse of dimensionality [17, 21]. Two approaches have emerged in recent years to achieve these goals: either regularize directly the measures themselves, by looking at them through a simplified lens; or regularize the original OT problem using various modifications. The first approach exploits well-known closed-form identities for OT when comparing two univariate

measures or two multivariate Gaussian measures. In this approach, one exploits those formulas and operates by summarizing complex measures as one or possibly many univariate or multivariate Gaussian measures. The second approach builds on the fact that for arbitrary measures, regularizing the OT problem, either in its primal or dual form, can result in simpler computations and possibly improved sample complexity. The latter approach can offer additional benefits for data science: because the original marginal constraints of the OT problem can also be relaxed, regularized OT can also yield useful tools to compare measures with different total mass — the so-called "unbalanced" case [3]— which provides a useful additional degree of freedom. Our work in this paper stands at the intersection of these two approaches. To our knowledge, that intersection was so far empty: no meaningful closed-form formulation was known for regularized optimal transport. We provide closed-form formulas of entropic (OT) of two Gaussian measures for balanced and unbalanced cases.

**Summarizing measures *vs*. regularizing OT.** Closed-form identities to compute OT distances (or more generally recover Monge maps) are known when either (1) both measures are univariate and the ground cost is submodular [43, §2]: in that case evaluating OT only requires integrating that submodular cost w.r.t. the quantile distributions of both measures; or (2) both measures are Gaussian, in a Hilbert space, and the ground cost is the squared Euclidean metric [18, 23], in which case the OT cost is given by the Wasserstein-Bures metric [4, 34]. These two formulas have inspired several works in which data measures are either projected onto 1D lines [40, 6], with further developments in [38, 30, 47]; or represented by Gaussians, to take advantage of the simpler computational possibilities offered by the Wasserstein-Bures metric [28, 37, 11].

Various schemes have been proposed to regularize the OT problem in the primal [14, 22] or the dual [45, 2, 15]. We focus in this work on the formulation obtained by [13], which combines entropic regularization [14] with a more general formulation for unbalanced transport [12, 31, 32]. The advantages of unbalanced entropic transport are numerous: it comes with favorable sample complexity regimes compared to unregularized OT [24], can be cast as a loss with favorable properties [26, 19], and can be evaluated using variations of the Sinkhorn algorithm [25].

**On the absence of closed-form formulas for regularized OT.** Despite its appeal, one of the shortcomings of entropic regularized OT lies in the absence of simple test-cases that admit closed-form formulas. While it is known that regularized OT can be related, in the limit of infinite regularization, to the energy distance [41], the absence of closed-form formulas for a fixed regularization strength poses an important practical problem to evaluate the performance of stochastic algorithms that try to approximate regularized OT: we do not know of any setup for which the ground truth value of entropic OT between continuous densities is known. The purpose of this paper is to fill this gap, and provide closed form expressions for balanced and unbalanced OT for Gaussian measures. We hope these formulas will prove useful in two different ways: as a solution to the problem outlined above, to facilitate the evaluation of new methodologies building on entropic OT, and more generally to propose a more robust yet well-grounded replacement to the Bures-Wasserstein metric.

**Related work.** From an economics theory perspective, Bojilov and Galichon [5] provided a closed form for an "equilibrium 2-sided matching problem" which is equivalent to entropy-regularized optimal transport. Second, a sequence of works in optimal control theory [9, 10, 8] studied stochastic systems, of which entropy regularized optimal transport between Gaussians can be seen as a special case, and found a closed form of the optimal dual potentials. Finally, a few recent concurrent works provided a closed form of entropy regularized OT between Gaussians: first Gerolin et al. [27] found a closed form in the univariate case, then Mallasto et al. [35] and del Barrio and Loubes [16] generalized the formula for multivariate Gaussians. The closest works to this paper are certainly those of Mallasto et al. [35] and del Barrio and Loubes [16] where the authors solved the balanced entropy regularized OT and studied the Gaussian barycenters problem. To the best of our knowledge, the closed form formula we provide for unbalanced OT is novel. Other differences between this paper and the aforementioned papers are highlighted below.

**Contributions.** Our contributions can be summarized as follows:

- Theorem 1 provides a closed form expression of the entropic (OT) plan $\pi$, which is shown to be a Gaussian measure itself (also shown in [5, 8, 35, 16]). Here, we furthermore study the properties of the OT loss function: it remains well defined, convex and differentiable even for singular covariance matrices unlike the Bures metric.

- Using the definition of debiased Sinkhorn barycenters [33, 29], Theorem 2 shows that the entropic barycenter of Gaussians is Gaussian and its covariance verifies a fixed point equation similar to that of Agueh and Carlier [1]. Mallasto et al. [35] and del Barrio and Loubes [16] provided similar fix point equations however by restricting the barycenter problem to the set of Gaussian measures whereas we consider the larger set of sub-Gaussian measures.

- As in the balanced case, Theorem 3 provides a closed form expression of the unbalanced Gaussian transport plan. The obtained formula sheds some light on the link between mass destruction and the distance between the means of $\alpha, \beta$ in Unbalanced OT.

**Notations.** $\mathcal{S}^d$ denotes the set of square symmetric matrices in $\mathbb{R}^{d \times d}$. $\mathcal{S}^d_{++}$ and $\mathcal{S}^d_+$ denote the cones of positive definite and positive semi-definite matrices in $\mathcal{S}^d$ respectively. Let $\mathcal{N}(\mathbf{a}, \mathbf{A})$ denote the multivariate Gaussian distribution with mean $\mathbf{a} \in \mathbb{R}^d$ and variance $\mathbf{A} \in \mathcal{S}^d_{++}$. $f = \mathcal{Q}(\mathbf{a}, \mathbf{A})$ denotes the quadratic form $f : x \mapsto -\frac{1}{2}(x^\top \mathbf{A} x - 2\mathbf{a}^\top x)$ with $\mathbf{A} \in \mathcal{S}^d$. For short, we denote $\mathcal{Q}(\mathbf{A}) = \mathcal{Q}(0, \mathbf{A})$. Whenever relevant, we follow the convention $0 \log 0 = 0$. $\mathcal{M}^+_p$ denotes the set of non-negative measures in $\mathbb{R}^d$ with a finite p-th order moment and its subset of probablity measures $\mathcal{P}_p$. For a non-negative measure $\alpha \in \mathcal{M}^+_p(\mathbb{R}^d)$, $\mathcal{L}_2(\alpha)$ denotes the set of functions $f : \mathbb{R}^d \to \mathbb{R}$ such that $\mathbb{E}_\alpha(|f|^2) = \int_{\mathbb{R}^d} |f|^2 \mathrm{d}\alpha < +\infty$. With $\mathbf{C} \in \mathcal{S}^d_{++}$ and $\mathbf{a}, \mathbf{b} \in \mathbb{R}^d$, we denote the squared Mahalanobis distance: $\|\mathbf{a} - \mathbf{b}\|^2_\mathbf{C} = (\mathbf{a} - \mathbf{b})^\top \mathbf{C}(\mathbf{a} - \mathbf{b})$.

## 2  Reminders on Optimal Transport

**The Kantorovich problem.** Let $\alpha, \beta \in \mathcal{P}_2$ and let $\Pi(\alpha, \beta)$ denote the set of probability measures in $\mathcal{P}_2$ with marginal distributions equal to $\alpha$ and $\beta$. The 2-Wasserstein distance is defined as:

$$W_2^2(\alpha, \beta) \stackrel{\text{def}}{=} \min_{\pi \in \Pi(\alpha, \beta)} \int_{\mathbb{R}^{d \times d}} \|x - y\|^2 \mathrm{d}\pi(x, y). \tag{1}$$

This is known as the *Kantorovich* formulation of optimal transport. When $\alpha$ is absolutely continuous with respect to the Lebesgue measure (i.e. when $\alpha$ has a density), Equation (1) can be equivalently rewritten using the *Monge* formulation, where $T_\sharp \mu = \nu$ *i.f.f.* for all Borel sets $A$, $\nu(T(A)) = \mu(A)$:

$$W_2^2(\alpha, \beta) = \min_{T : T_\sharp \alpha = \nu} \int_{\mathbb{R}^d} \|x - T(x)\|^2 \mathrm{d}\alpha(x). \tag{2}$$

The optimal map $T^*$ in Equation (2) is called the Monge map.

**The Wasserstein-Bures metric.** Let $\mathcal{N}(m, \Sigma)$ denote the Gaussian distribution on $\mathbb{R}^d$ with mean $m \in \mathbb{R}^d$ and covariance matrix $\Sigma \in S^d_{++}$. A well-known fact [18, 46] is that Equation (1) admits a closed form for Gaussian distributions, called the Wasserstein-Bures distance (a.k.a. the *Fréchet* distance):

$$W_2^2(\mathcal{N}(a, \mathbf{A}), \mathcal{N}(b, \mathbf{B})) = \|a - b\|^2 + \mathfrak{B}^2(\mathbf{A}, \mathbf{B}), \tag{3}$$

where $\mathfrak{B}$ is the *Bures* distance [4] between positive matrices:

$$\mathfrak{B}^2(\mathbf{A}, \mathbf{B}) \stackrel{\text{def}}{=} \mathrm{Tr}\mathbf{A} + \mathrm{Tr}\mathbf{B} - 2\mathrm{Tr}(\mathbf{A}^{\frac{1}{2}} \mathbf{B} \mathbf{A}^{\frac{1}{2}})^{\frac{1}{2}}. \tag{4}$$

Moreover, the Monge map between two Gaussian distributions admits a closed form: $T^\star : x \to \mathbf{T^{AB}}(x - \mathbf{a}) + \mathbf{b}$, with

$$\mathbf{T^{AB}} \stackrel{\text{def}}{=} \mathbf{A}^{-\frac{1}{2}}(\mathbf{A}^{\frac{1}{2}} \mathbf{B} \mathbf{A}^{\frac{1}{2}})^{\frac{1}{2}} \mathbf{A}^{-\frac{1}{2}} = \mathbf{B}^{\frac{1}{2}}(\mathbf{B}^{\frac{1}{2}} \mathbf{A} \mathbf{B}^{\frac{1}{2}})^{-\frac{1}{2}} \mathbf{B}^{\frac{1}{2}}, \tag{5}$$

which is related to the Bures gradient (w.r.t. the Frobenius inner product):

$$\nabla_\mathbf{A} \mathfrak{B}^2(\mathbf{A}, \mathbf{B}) = \mathrm{Id} - \mathbf{T^{AB}}. \tag{6}$$

$\mathfrak{B}^2(\mathbf{A}, \mathbf{B})$ and its gradient can be computed efficiently on GPUs using Newton-Schulz iterations which are provided in **??** along with numerical experiments in the appendix.

# 3 Entropy-Regularized Optimal Transport between Gaussians

Solving (1) can be quite challenging, even in a discrete setting [39]. Adding an entropic regularization term to (1) results in a problem which can be solved efficiently using Sinkhorn's algorithm [14]. Let $\sigma > 0$. This corresponds to solving the following problem:

$$\mathrm{OT}_\sigma(\alpha, \beta) \stackrel{\mathrm{def}}{=} \min_{\pi \in \Pi(\alpha,\beta)} \int_{\mathbb{R}^d \times \mathbb{R}^d} \|x - y\|^2 \mathrm{d}\pi(x, y) + 2\sigma^2 \,\mathrm{KL}(\pi\|\alpha \otimes \beta), \tag{7}$$

where $\mathrm{KL}(\pi\|\alpha \otimes \beta) \stackrel{\mathrm{def}}{=} \int_{\mathbb{R}^d} \log\left(\frac{\mathrm{d}\pi}{\mathrm{d}\alpha\mathrm{d}\beta}\right) \mathrm{d}\pi$ is the Kullback-Leibler divergence (or relative entropy). As in the original case (1), $\mathrm{OT}_\sigma$ can be studied with centered measures (i.e zero mean) with no loss of generality:

**Lemma 1.** *Let $\alpha, \beta \in \mathcal{P}$ and $\bar{\alpha}, \bar{\beta}$ their respective centered transformations. It holds that*

$$\mathrm{OT}_\sigma(\alpha, \beta) = \mathrm{OT}_\sigma(\bar{\alpha}, \bar{\beta}) + \|\mathbf{a} - \mathbf{b}\|^2. \tag{8}$$

**Dual problem and Sinkhorn's algorithm.**  Compared to (1), (7) enjoys additional properties, such as the uniqueness of the solution $\pi^*$. Moreover, problem (7) has the following dual formulation:

$$\mathrm{OT}_\sigma(\alpha, \beta) = \max_{\substack{f \in \mathcal{L}_1(\alpha), \\ g \in \mathcal{L}_1(\beta)}} \mathbb{E}_\alpha(f) + \mathbb{E}_\beta(g) - 2\sigma^2 \left(\int_{\mathbb{R}^d \times \mathbb{R}^d} e^{\frac{f(x)+g(y)-\|x-y\|^2}{2\sigma^2}} \mathrm{d}\alpha(x)\mathrm{d}\beta(y) - 1\right). \tag{9}$$

If $\alpha$ and $\beta$ have finite second order moments, a pair of dual potentials $(f, g)$ is optimal if and only they verify the following optimality conditions $\beta$-a.s and $\alpha$-a.s respectively [36]:

$$e^{\frac{f(x)}{2\sigma^2}}\left(\int_{\mathbb{R}^d} e^{\frac{-\|x-y\|^2+g(y)}{2\sigma^2}} \mathrm{d}\beta(y)\right) = 1, \quad e^{\frac{g(x)}{2\sigma^2}}\left(\int_{\mathbb{R}^d} e^{\frac{-\|x-y\|^2+f(y)}{2\sigma^2}} \mathrm{d}\alpha(y)\right) = 1. \tag{10}$$

Moreover, given a pair of optimal dual potentials $(f, g)$, the optimal transportation plan is given by

$$\frac{\mathrm{d}\pi^\star}{\mathrm{d}\alpha\mathrm{d}\beta}(x, y) = e^{\frac{f(x)+g(y)-\|x-y\|^2}{2\sigma^2}}. \tag{11}$$

Starting from a pair of potentials $(f_0, g_0)$, the optimality conditions (10) lead to an alternating dual ascent algorithm, which is equivalent to Sinkhorn's algorithm in log-domain:

$$\begin{aligned}
g_{n+1} &= \left(y \in \mathbb{R}^d \to -2\sigma^2 \log \int_{\mathbb{R}^d} e^{\frac{-\|x-y\|^2+f_n(x)}{2\sigma^2}} \mathrm{d}\alpha(x)\right), \\
f_{n+1} &= \left(x \in \mathbb{R}^d \to -2\sigma^2 \log \int_{\mathbb{R}^d} e^{\frac{-\|x-y\|^2+g_{n+1}(y)}{2\sigma^2}} \mathrm{d}\beta(y)\right).
\end{aligned} \tag{12}$$

Séjourné et al. [44] showed that when the support of the measures is compact, Sinkhorn's algorithm converges to a pair of dual potentials. Here in particular, we study Sinkhorn's algorithm when $\alpha$ and $\beta$ are Gaussian measures.

**Closed form expression for Gaussian measures.**

**Theorem 1.** *Let $\mathbf{A}, \mathbf{B} \in \mathcal{S}_{++}^d$ and $\alpha \sim \mathcal{N}(\mathbf{a}, \mathbf{A})$ and $\beta \sim \mathcal{N}(\mathbf{b}, \mathbf{B})$. Define $\mathbf{D}_\sigma = (4\mathbf{A}^{\frac{1}{2}}\mathbf{B}\mathbf{A}^{\frac{1}{2}} + \sigma^4 \,\mathrm{Id})^{\frac{1}{2}}$. Then,*

$$\mathrm{OT}_\sigma(\alpha, \beta) = \|\mathbf{a} - \mathbf{b}\|^2 + \mathcal{B}_\sigma^2(\mathbf{A}, \mathbf{B}), \text{ where} \tag{13}$$

$$\mathcal{B}_\sigma^2(\mathbf{A}, \mathbf{B}) = \mathrm{Tr}(\mathbf{A}) + \mathrm{Tr}(\mathbf{B}) - \mathrm{Tr}(\mathbf{D}_\sigma) + d\sigma^2(1 - \log(2\sigma^2)) + \sigma^2 \log \det\left(\mathbf{D}_\sigma + \sigma^2 \,\mathrm{Id}\right). \tag{14}$$

*Moreover, with $\mathbf{C}_\sigma = \frac{1}{2}\mathbf{A}^{\frac{1}{2}}\mathbf{D}_\sigma\mathbf{A}^{-\frac{1}{2}} - \frac{\sigma^2}{2}\,\mathrm{Id}$, the Sinkhorn optimal transportation plan is also a Gaussian measure over $\mathbb{R}^d \times \mathbb{R}^d$ given by*

$$\pi^\star \sim \mathcal{N}\left(\begin{pmatrix}\mathbf{a}\\\mathbf{b}\end{pmatrix}, \begin{pmatrix}\mathbf{A} & \mathbf{C}_\sigma\\\mathbf{C}_\sigma^\top & \mathbf{B}\end{pmatrix}\right). \tag{15}$$

**Remark 1.** *While for our proof it is necessary to assume that $\mathbf{A}$ and $\mathbf{B}$ are positive definite in order for them to have a Lebesgue density, notice that the closed form formula given by Theorem 1 remains well-defined for positive semi-definite matrices. Moreover, unlike the Bures-Wasserstein metric, $\mathrm{OT}_\sigma$ is differentiable even when $\mathbf{A}$ or $\mathbf{B}$ are singular.*

The proof of 1 is broken down into smaller results, Propositions 1 to 3 and lemma 2. Using Lemma 1, we can focus in the rest of this section on centered Gaussians without loss of generality.

**Sinkhorn's algorithm and quadratic potentials.** We obtain a closed form solution of $\mathrm{OT}_\sigma$ by considering quadratic solutions of (10). The following key proposition characterizes the obtained potential after a pair of Sinkhorn iterations with quadratic forms.

**Proposition 1.** *Let $\alpha \sim \mathcal{N}(0, \mathbf{A})$ and $\beta \sim \mathcal{N}(0, \mathbf{B})$ and the Sinkhorn transform $T_\alpha : \mathbb{R}^{\mathbb{R}^d} \to \mathbb{R}^{\mathbb{R}^d}$:*

$$T_\alpha(h)(x) \overset{\text{def}}{=} -\log \int_{\mathbb{R}^d} e^{\frac{-\|x-y\|^2}{2\sigma^2} + h(y)} \mathrm{d}\alpha(y). \tag{16}$$

*Let $\mathbf{X} \in \mathcal{S}_d$. If $h = m + \mathcal{Q}(\mathbf{X})$ i.e $h(x) = m - \frac{1}{2}x^\top \mathbf{X} x$ for some $m \in \mathbb{R}$, then $T_\alpha(h)$ is well-defined if and only if $\mathbf{X}' \overset{\text{def}}{=} \sigma^2 \mathbf{X} + \sigma^2 \mathbf{A}^{-1} + \mathrm{Id} \succ 0$. In that case,*

*(i) $T_\alpha(h) = \mathcal{Q}(\mathbf{Y}) + m'$ where $\mathbf{Y} = \frac{1}{\sigma^2}(\mathbf{X}'^{-1} - \mathrm{Id})$ and $m' \in \mathbb{R}$ is an additive constant,*

*(ii) $T_\beta(T_\alpha(h))$ is well-defined and is also a quadratic form up to an additive constant, since $\mathbf{Y}' \overset{\text{def}}{=} \sigma^2 \mathbf{Y} + \sigma^2 \mathbf{B}^{-1} + \mathrm{Id} = \mathbf{X}'^{-1} + \sigma^2 \mathbf{B}^{-1} \succ 0$ and (i) applies.*

Consider the null inialization $f_0 = 0 = \mathcal{Q}(0)$. Since $\sigma^2 \mathbf{A}^{-1} + \mathrm{Id} \succ 0$, Proposition 1 applies with $\mathbf{X} = 0$ and a simple induction shows that $(f_n, g_n)$ remain quadratic forms for all $n$. Sinkhorn's algorithm can thus be written as an algorithm on positive definite matrices.

**Proposition 2.** *Starting with null potentials, Sinkhorn's algorithm is equivalent to the iterations:*

$$\mathbf{F}_{n+1} = \sigma^2 \mathbf{A}^{-1} + \mathbf{G}_n^{-1}, \qquad \mathbf{G}_{n+1} = \sigma^2 \mathbf{B}^{-1} + \mathbf{F}_{n+1}^{-1}, \tag{17}$$

*with $\mathbf{F}_0 = \sigma^2 \mathbf{A}^{-1} + \mathrm{Id}$ and $\mathbf{G}_0 = \sigma^2 \mathbf{B}^{-1} + \mathrm{Id}$.*

Moreover, the sequence $(\mathbf{F}_n, \mathbf{G}_n)$ is contractive (in the matrix operator norm) and converges towards a pair of positive definite matrices $(\mathbf{F}, \mathbf{G})$. At optimality, the dual potentials are determined up to additive constants $f_0$ and $g_0$: $\frac{f}{2\sigma^2} = \mathcal{Q}(\mathbf{U}) + f_0$ and $\frac{g}{2\sigma^2} = \mathcal{Q}(\mathbf{V}) + g_0$ where $\mathbf{U}$ and $\mathbf{V}$ are given by

$$\mathbf{F} = \sigma^2 \mathbf{U} + \sigma^2 \mathbf{A}^{-1} + \mathrm{Id}, \qquad \mathbf{G} = \sigma^2 \mathbf{V} + \sigma^2 \mathbf{B}^{-1} + \mathrm{Id} . \tag{18}$$

**Closed form solution.** Taking the limit of Sinkhorn's equations (17) along with the change of variable (18), there exists a pair of optimal potentials determined up to an additive constant:

$$\frac{f}{2\sigma^2} = \mathcal{Q}(\mathbf{U}) = \mathcal{Q}\left(\frac{1}{\sigma^2}(\mathbf{G}^{-1} - \mathrm{Id})\right), \qquad \frac{g}{2\sigma^2} = \mathcal{Q}(\mathbf{V}) = \mathcal{Q}\left(\frac{1}{\sigma^2}(\mathbf{F}^{-1} - \mathrm{Id})\right), \tag{19}$$

where $(\mathbf{F}, \mathbf{G})$ is the solution of the fixed point equations

$$\mathbf{F} = \sigma^2 \mathbf{A}^{-1} + \mathbf{G}^{-1}, \qquad \mathbf{G} = \sigma^2 \mathbf{B}^{-1} + \mathbf{F}^{-1}. \tag{20}$$

Let $\mathbf{C} \overset{\text{def}}{=} \mathbf{A}\mathbf{G}^{-1}$. Combining both equations of (20) in one leads to $\mathbf{G} = \sigma^2 \mathbf{B}^{-1} + (\mathbf{G}^{-1} + \sigma^2 \mathbf{A}^{-1})^{-1}$, which can be shown to be equivalent to

$$\mathbf{C}^2 + \sigma^2 \mathbf{C} - \mathbf{A}\mathbf{B} = 0. \tag{21}$$

Notice that since $\mathbf{A}$ and $\mathbf{G}^{-1}$ are positive definite, their product $\mathbf{C} = \mathbf{A}\mathbf{G}^{-1}$ is similar to $\mathbf{A}^{\frac{1}{2}}\mathbf{G}^{-1}\mathbf{A}^{\frac{1}{2}}$. Thus it has positive eigenvalues. Proposition 3 provides the only feasible solution of (21).

**Proposition 3.** *Let $\sigma^2 \geq 0$ and $\mathbf{C}$ satisfying Equation (21). Then,*

$$\mathbf{C} = \left(\mathbf{A}\mathbf{B} + \frac{\sigma^4}{4}\mathrm{Id}\right)^{\frac{1}{2}} - \frac{\sigma^2}{2}\mathrm{Id} = \mathbf{A}^{\frac{1}{2}}(\mathbf{A}^{\frac{1}{2}}\mathbf{B}\mathbf{A}^{\frac{1}{2}} + \frac{\sigma^4}{4}\mathrm{Id})^{\frac{1}{2}}\mathbf{A}^{-\frac{1}{2}} - \frac{\sigma^2}{2}\mathrm{Id} . \tag{22}$$

**Corollary 1.** *The optimal dual potentials of (19) can be given in closed form by:*

$$\mathbf{U} = \frac{\mathbf{B}}{\sigma^2}(\mathbf{C} + \sigma^2 \mathrm{Id})^{-1} - \frac{\mathrm{Id}}{\sigma^2}, \qquad \mathbf{V} = (\mathbf{C} + \sigma^2 \mathrm{Id})^{-1}\frac{\mathbf{A}}{\sigma^2} - \frac{\mathrm{Id}}{\sigma^2}. \tag{23}$$

*Moreover, $\mathbf{U}$ and $\mathbf{V}$ remain well-defined even for singular matrices $\mathbf{A}$ and $\mathbf{B}$.*

**Optimal transportation plan and** $\mathrm{OT}_\sigma$**.**   Using Corollary 1 and (19), Equation (11) leads to a closed form expression of $\pi$. To conclude the proof of Theorem 1, we introduce lemma 2 that computes the $\mathrm{OT}_\sigma$ loss at optimality. Detailed technical proofs are provided in the appendix.

**Lemma 2.** *Let* $\mathbf{A}, \mathbf{B}, \mathbf{C}$ *be invertible matrices such that* $\mathbf{H} = \left( \begin{smallmatrix} \mathbf{A} & \mathbf{C} \\ \mathbf{C}^\top & \mathbf{B} \end{smallmatrix} \right) \succ 0$. *Let* $\alpha = \mathcal{N}(0, \mathbf{A}), \beta = \mathcal{N}(0, \mathbf{B})$, *and* $\pi = \mathcal{N}(0, \mathbf{H})$. *Then,*

$$\int_{\mathbb{R}^d \times \mathbb{R}^d} \|x - y\|^2 \mathrm{d}\pi(x, y) = \mathrm{Tr}(\mathbf{A}) + \mathrm{Tr}(\mathbf{B}) - 2\mathrm{Tr}(\mathbf{C}), \tag{24}$$

$$\mathrm{KL}\left( \pi \| \alpha \otimes \beta \right) = \tfrac{1}{2} \left( \log \det \mathbf{A} + \log \det \mathbf{B} - \log \det \left( \begin{smallmatrix} \mathbf{A} & \mathbf{C} \\ \mathbf{C}^T & \mathbf{B} \end{smallmatrix} \right) \right). \tag{25}$$

**Properties of** $\mathrm{OT}_\sigma$**.**   Theorem 1 shows that $\pi$ has a Gaussian density. Proposition 4 allows to reformulate this optimization problem over couplings in $\mathbb{R}^{d \times d}$ with a positivity constraint.

**Proposition 4.** *Let* $\alpha = \mathcal{N}(0, \mathbf{A}), \beta = \mathcal{N}(0, \mathbf{B})$, *and* $\sigma^2 > 0$. *Then,*

$$\mathrm{OT}_\sigma(\alpha, \beta) = \min_{\mathbf{C}: \left( \begin{smallmatrix} \mathbf{A} & \mathbf{C} \\ \mathbf{C}^T & \mathbf{B} \end{smallmatrix} \right) \geq 0} \left\{ \mathrm{Tr}(\mathbf{A}) + \mathrm{Tr}(\mathbf{B}) - 2\mathrm{Tr}(\mathbf{C}) + \sigma^2 (\log \det \mathbf{AB} - \log \det \left( \begin{smallmatrix} \mathbf{A} & \mathbf{C} \\ \mathbf{C}^T & \mathbf{B} \end{smallmatrix} \right)) \right\} \tag{26}$$

$$= \min_{\mathbf{K} \in \mathbb{R}^{d \times d}: \|\mathbf{K}\|_{op} \leq 1} \mathrm{Tr}\mathbf{A} + \mathrm{Tr}\mathbf{B} - 2\mathrm{Tr}\mathbf{A}^{\frac{1}{2}}\mathbf{K}\mathbf{B}^{\frac{1}{2}} - \sigma^2 \ln \det(\mathrm{Id} - \mathbf{K}\mathbf{K}^\top). \tag{27}$$

*Moreover, both* (26) *and* (27) *are convex problems.*

We now study the convexity and differentiability of $\mathrm{OT}_\sigma$, which are more conveniently derived from the dual problem of (26) given as a positive definite program:

**Proposition 5.** *The dual problem of* (26) *can be written with no duality gap as*

$$\max_{\mathbf{F}, \mathbf{G} \succ 0} \left\{ \langle \mathrm{Id} - \mathbf{F}, \, \mathbf{A} \rangle + \langle \mathrm{Id} - \mathbf{G}, \, \mathbf{B} \rangle + \sigma^2 \log \det \left( \frac{\mathbf{FG} - \mathrm{Id}}{\sigma^4} \right) + \sigma^2 \log \det \mathbf{AB} + 2d\sigma^2 \right\}. \tag{28}$$

Feydy et al. [19] showed that on compact spaces, the gradient of $\mathrm{OT}_\sigma$ is given by the optimal dual potentials. This result was later extended by Janati et al. [29] to sub-Gaussian measures with unbounded supports. The following proposition re-establishes this statement for Gaussians.

**Proposition 6.** *Assume* $\sigma > 0$ *and consider the pair* $\mathbf{U}, \mathbf{V}$ *of Corollary 1. Then*

   (i) *The optimal pair* $(\mathbf{F}^*, \mathbf{G}^*)$ *of* (28) *is a solution to the fixed point problem* (20),

   (ii) $\mathfrak{B}_{\sigma^2}$ *is differentiable and:* $\nabla \mathfrak{B}_{\sigma^2}(\mathbf{A}, \mathbf{B}) = -(\sigma^2 \mathbf{U}, \sigma^2 \mathbf{V})$. *Thus:* $\nabla_\mathbf{A} \mathfrak{B}_{\sigma^2}(\mathbf{A}, \mathbf{B}) = \mathrm{Id} - \mathbf{B}^{\frac{1}{2}} \left( (\mathbf{B}^{\frac{1}{2}}\mathbf{A}\mathbf{B}^{\frac{1}{2}} + \frac{\sigma^4}{4}\mathrm{Id})^{\frac{1}{2}} + \frac{\sigma^2}{2}\mathrm{Id} \right)^{-1} \mathbf{B}^{\frac{1}{2}}$,

   (iii) $(\mathbf{A}, \mathbf{B}) \mapsto \mathfrak{B}_{\sigma^2}(\mathbf{A}, \mathbf{B})$ *is convex in* $\mathbf{A}$ *and in* $\mathbf{B}$ *but not jointly.*

   (iv) *For a fixed* $\mathbf{B}$ *with its spectral decomposition* $\mathbf{B} = \mathbf{P}\Sigma\mathbf{P}^\top$, *the function* $\phi_\mathbf{B} : \mathbf{A} \mapsto \mathfrak{B}_{\sigma^2}(\mathbf{A}, \mathbf{B})$ *is minimized at* $\mathbf{A}_0 = \mathbf{P}(\Sigma - \sigma^2 \mathrm{Id})_+ \mathbf{P}^\top$ *where the thresholding operator* $_+$ *is defined by* $x_+ = \max(x, 0)$ *for any* $x \in \mathbb{R}$ *and extended element-wise to diagonal matrices.*

When $\mathbf{A}$ and $\mathbf{B}$ are not singular, by letting $\sigma \to 0$ in $\nabla_\mathbf{A} \mathfrak{B}_{\sigma^2}(\mathbf{A}, \mathbf{B})$, we recover the gradient of the Bures metric given in (6). Moreover, (iv) illustrates the entropy bias of $\mathfrak{B}_{\sigma^2}$. Feydy et al. [19] showed that it can be circumvented by considering the Sinkhorn divergence:

$$S_\sigma : (\alpha, \beta) \mapsto \mathrm{OT}_\sigma(\alpha, \beta) - \frac{1}{2}(\mathrm{OT}_\sigma(\alpha, \alpha) + \mathrm{OT}_\sigma(\beta, \beta)) \tag{29}$$

which is non-negative and equals 0 if and only if $\alpha = \beta$. Using the differentiability and convexity of $S_\sigma$ on sub-Gaussian measures [29], we conclude this section by showing that the debiased Sinkhorn barycenter of Gaussians remains Gaussian:

**Theorem 2.** *Consider the restriction of* $\mathrm{OT}_\sigma$ *to the set of sub-Gaussian measures* $\mathcal{G} \stackrel{\mathrm{def}}{=} \{\mu \in \mathcal{P}_2 | \exists q > 0, \mathbb{E}_\mu(e^{q\|X\|^2}) < +\infty\}$ *and let* $K$ *Gaussian measures* $\alpha_k \sim \mathcal{N}(\mathbf{a}_k, \mathbf{A}_k)$ *with a sequence of positive weights* $(w_k)_k$ *summing to 1. Then, the weighted debiased barycenter defined by:*

$$\beta \stackrel{\mathrm{def}}{=} \mathrm{argmin}_{\beta \in \mathcal{G}} \sum_{k=1}^K w_k S_\sigma(\alpha_k, \beta) \tag{30}$$

is a Gaussian measure given by $\mathcal{N}\left(\sum_{k=1}^{K} w_k \mathbf{a}_k, \mathbf{B}\right)$ where $\mathbf{B} \in \mathcal{S}_+^d$ is a solution of the equation:

$$\sum_{k=1}^{K} w_k (\mathbf{B}^{\frac{1}{2}} \mathbf{A}_k \mathbf{B}^{\frac{1}{2}} + \frac{\sigma^4}{4} \operatorname{Id})^{\frac{1}{2}} = (\mathbf{B}^2 + \frac{\sigma^4}{4} \operatorname{Id})^{\frac{1}{2}} \tag{31}$$

## 4 Entropy Regularized OT between Unbalanced Gaussians

We proceed by considering a more general setting, in which measures $\alpha, \beta \in \mathcal{M}_2^+(\mathbb{R}^d)$ have finite integration masses $m_\alpha = \alpha(\mathbb{R}^d)$ and $m_\beta = \beta(\mathbb{R}^d)$ that are not necessarily the same. We remind the reader of entropy-regularized unbalanced OT:

$$\operatorname{UOT}_{2\sigma^2}^{\otimes}(\alpha, \beta) \stackrel{\text{def}}{=} \inf_{\pi \in \mathcal{M}_1^+} \int_{\mathbb{R}^d \times \mathbb{R}^d} \|x - y\|^2 d\pi(x, y) + 2\sigma^2 \operatorname{KL}(\pi \| \alpha \otimes \beta) + \gamma \operatorname{KL}(\pi_1 \| \alpha) + \gamma \operatorname{KL}(\pi_2 \| \beta), \tag{32}$$

where $\gamma > 0$ and $\pi_1$, $\pi_2$ are the marginal distributions of the coupling $\pi$.

**Duality and optimality conditions.** By definition of the KL divergence, the term $\operatorname{KL}(\pi \| \alpha \otimes \beta)$ in (32) is finite if and only if $\pi$ admits a density with respect to $\alpha \otimes \beta$. Therefore (32) can be formulated as a variational problem:

$$\operatorname{UOT}_\sigma(\alpha, \beta) \stackrel{\text{def}}{=} \inf_{r \in \mathcal{L}_1(\alpha \otimes \beta)} \left\{ \int_{\mathbb{R}^d \times \mathbb{R}^d} \|x - y\|^2 r(x, y) d\alpha(x) d\beta(y) \right. \tag{33}$$
$$\left. + 2\sigma^2 \operatorname{KL}(r \| \alpha \otimes \beta) + \gamma \operatorname{KL}(r_1 \| \alpha) + \gamma \operatorname{KL}(r_2 \| \beta) \right\},$$

where $r_1 \stackrel{\text{def}}{=} \int_{\mathbb{R}^d} r(., y) d\beta(y)$ and $r_2 \stackrel{\text{def}}{=} \int_{\mathbb{R}^d} r(x, .) d\alpha(x)$ correspond to the marginal density functions and the Kullback-Leibler divergence is defined as: $\operatorname{KL}(f \| \mu) = \int_{\mathbb{R}^d} (f \log(f) + 1 - f) d\mu$. As in [13], Fenchel-Rockafellar duality provides the following dual problem:

$$\operatorname{UOT}_\sigma(\alpha, \beta) = \sup_{\substack{f \in \mathcal{L}_\infty(\alpha) \\ g \in \mathcal{L}_\infty(\beta)}} \left\{ \gamma \int_{\mathbb{R}^d} (1 - e^{-\frac{f}{\gamma}}) d\alpha + \gamma \int_{\mathbb{R}^d} (1 - e^{-\frac{g}{\gamma}}) d\beta \right. \tag{34}$$
$$\left. - 2\sigma^2 \int_{\mathbb{R}^d \times \mathbb{R}^d} (e^{\frac{-\|x-y\|^2 + f(x) + g(y)}{2\sigma^2}} - 1) d\alpha(x) d\beta(y) \right\}.$$

For which strong duality holds. Moreover, a maximizing sequence of potentials $(f_n, g_n)$ weakly converges towards a pair of measurable functions $(f, g)$ if and only if [42]:

$$\frac{f(x)}{2\sigma^2} \stackrel{a.s}{=} -\tau \log \int_{\mathbb{R}^d} e^{\frac{g(y) - \|x-y\|^2}{2\sigma^2}} d\beta(y), \quad \frac{g(x)}{2\sigma^2} \stackrel{a.s}{=} -\tau \log \int_{\mathbb{R}^d} e^{\frac{f(y) - \|x-y\|^2}{2\sigma^2}} d\alpha(y), \tag{35}$$

where $\tau \stackrel{\text{def}}{=} \frac{\gamma}{\gamma + 2\sigma^2}$.

In which case the (unique) optimal transportation plan is given by:

$$\frac{d\pi}{d\alpha \otimes d\beta}(x, y) = e^{\frac{f(x) + g(y) - \|x-y\|^2}{2\sigma^2}}. \tag{36}$$

The following proposition provides a simple formula to compute $\operatorname{UOT}_\sigma$ at optimality. It shows that it is sufficient to know the total transported mass $\pi(\mathbb{R}^d \times \mathbb{R}^d)$.

**Proposition 7.** *Assume there exists an optimal transportation plan $\pi^*$, solution of* (32). *Then*

$$\operatorname{UOT}_\sigma(\alpha, \beta) = \gamma(m_\alpha + m_\beta) + 2\sigma^2 m_\alpha m_\beta - 2(\sigma^2 + \gamma)\pi^*(\mathbb{R}^d \times \mathbb{R}^d). \tag{37}$$

**Unbalanced OT for scaled Gaussians.** Let $\alpha$ and $\beta$ be unbalanced Gaussian measures. Formally, $\alpha = m_\alpha \mathcal{N}(\mathbf{a}, \mathbf{A})$ and $\beta = m_\beta \mathcal{N}(\mathbf{b}, \mathbf{B})$ with $m_\alpha, m_\beta > 0$. Unlike balanced OT, $\alpha$ and $\beta$ cannot be assumed to be centered without loss of generality. However, we can still derive a closed form formula for $\operatorname{UOT}_\sigma(\alpha, \beta)$ by considering quadratic potentials of the form

$$\frac{f(\mathbf{x})}{2\sigma^2} = -\frac{1}{2}(x^\top \mathbf{U} x - 2x^\top \mathbf{u}) + \log(m_u), \quad \frac{g(x)}{2\sigma^2} = -\frac{1}{2}(x^\top \mathbf{V} x - 2x^\top \mathbf{v}) + \log(m_v). \tag{38}$$

Let $\sigma$ and $\gamma$ be the regularization parameters as in Equation (33), and $\tau \stackrel{\text{def}}{=} \frac{\gamma}{2\sigma^2+\gamma}$, $\lambda \stackrel{\text{def}}{=} \frac{\sigma^2}{1-\tau} = \sigma^2 + \frac{\gamma}{2}$. Let us define the following useful quantities:

$$\mu = \begin{pmatrix} \mathbf{a} + \mathbf{A}\mathbf{X}^{-1}(\mathbf{b} - \mathbf{a}) \\ \mathbf{b} + \mathbf{B}\mathbf{X}^{-1}(\mathbf{a} - \mathbf{b}) \end{pmatrix} \tag{39}$$

$$\mathbf{H} = \begin{pmatrix} (\mathrm{Id} + \frac{1}{\lambda}\mathbf{C})(\mathbf{A} - \mathbf{A}\mathbf{X}^{-1}\mathbf{A}) & \mathbf{C} + (\mathrm{Id} + \frac{1}{\lambda}\mathbf{C})\mathbf{A}\mathbf{X}^{-1}\mathbf{B} \\ \mathbf{C}^\top + (\mathrm{Id} + \frac{1}{\lambda}\mathbf{C}^\top)\mathbf{B}\mathbf{X}^{-1}\mathbf{A} & (\mathrm{Id} + \frac{1}{\lambda}\mathbf{C}^\top)(\mathbf{B} - \mathbf{B}\mathbf{X}^{-1}\mathbf{B}) \end{pmatrix} \tag{40}$$

$$m_\pi = \sigma^{\frac{d\sigma^2}{\gamma+\sigma^2}} \left( m_\alpha m_\beta \det(\mathbf{C}) \sqrt{\frac{\det(\widetilde{\mathbf{A}}\widetilde{\mathbf{B}})^\tau}{\det(\mathbf{A}\mathbf{B})}} \right)^{\frac{1}{\tau+1}} \frac{e^{-\frac{\|\mathbf{a}-\mathbf{b}\|^2_{\mathbf{X}^{-1}}}{2(\tau+1)}}}{\sqrt{\det(\mathbf{C} - \frac{2}{\gamma}\widetilde{\mathbf{A}}\widetilde{\mathbf{B}})}}, \tag{41}$$

with

$$\mathbf{X} = \mathbf{A} + \mathbf{B} + \lambda\,\mathrm{Id}, \qquad\qquad \widetilde{\mathbf{A}} = \frac{\gamma}{2}(\mathrm{Id} - \lambda(\mathbf{A} + \lambda\,\mathrm{Id})^{-1}),$$

$$\widetilde{\mathbf{B}} = \frac{\gamma}{2}(\mathrm{Id} - \lambda(\mathbf{B} + \lambda\,\mathrm{Id})^{-1}), \qquad\qquad \mathbf{C} = \left(\frac{1}{\tau}\widetilde{\mathbf{A}}\widetilde{\mathbf{B}} + \frac{\sigma^4}{4}\,\mathrm{Id}\right)^{\frac{1}{2}} - \frac{\sigma^2}{2}\,\mathrm{Id}.$$

**Theorem 3.** *Let $\alpha = m_\alpha \mathcal{N}(\mathbf{a}, \mathbf{A})$ and $\beta = m_\beta \mathcal{N}(\mathbf{b}, \mathbf{B})$ be two unbalanced Gaussian measures. Let $\tau = \frac{\gamma}{2\sigma^2+\gamma}$ and $\lambda \stackrel{\text{def}}{=} \frac{\sigma^2}{1-\tau} = \sigma^2 + \frac{\gamma}{2}$ and $\mu$, $\mathbf{H}$, and $m_\pi$ be as above. Then*

*(i) The unbalanced optimal transport plan, minimizer of (32), is also an unbalanced Gaussian over $\mathbb{R}^d \times \mathbb{R}^d$ given by $\pi = m_\pi \mathcal{N}(\mu, \mathbf{H})$,*

*(ii) $\mathrm{UOT}_\sigma$ can be obtained in closed form using Proposition 7 with $\pi(\mathbb{R}^d \times \mathbb{R}^d) = m_\pi$.*

**Remark 2.** *The exponential term in the closed form formula above provides some intuition on how transportation occurs in unbalanced OT. When the difference between the means is too large, the transported mass $m_\pi^\star$ goes to 0 and thus no transport occurs. However for fixed means $\mathbf{a}, \mathbf{b}$, when $\gamma \to +\infty$, $\mathbf{X}^{-1} \to 0$ and the exponential term approaches 1.*

## 5  Numerical Experiments

**Empirical validation of the closed form formulas.** Figure 1 illustrates the convergence towards the closed form formulas of both theorems. For each dimension $d$ in [5, 10], we select a pair of Gaussians $\alpha \sim \mathcal{N}(\mathbf{a}, \mathbf{A})$ and $\beta \sim m_\beta \mathcal{N}(\mathbf{b}, \mathbf{B})$ with $m_\beta$ equals 1 (balanced) or 2 (unbalanced) and randomly generated means $\mathbf{a}, \mathbf{b}$ (uniform in $[-1, 1]^d$) and covariances $\mathbf{A}, \mathbf{B} \in S_{++}^d$ following the Wishart distribution $W_d(0.2 * \mathrm{Id}, d)$. We generate i.i.d datasets $\alpha_n \sim \mathcal{N}(\mathbf{a}, \mathbf{A})$ and $\beta_n \sim m_\beta \mathcal{N}(\mathbf{b}, \mathbf{B})$ with $n$ samples and compute $\mathrm{OT}_\sigma$ / $\mathrm{UOT}_\sigma$. We report means and $\pm$ shaded standard-deviation areas over 20 independent trials for each value of $n$.

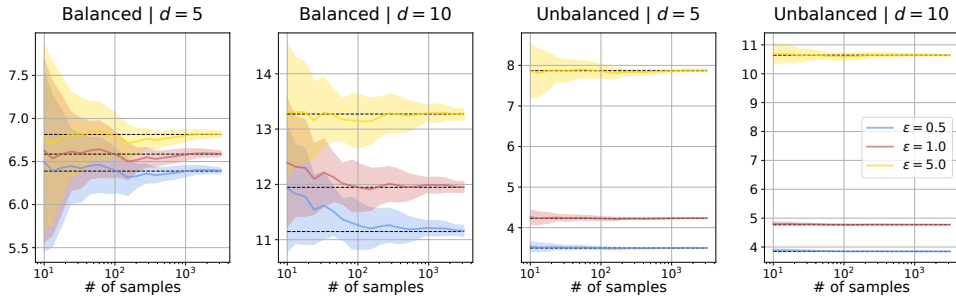

Figure 1: Numerical convergence the (n-samples) empirical estimation of $\mathrm{OT}(\alpha_n, \beta_n)$ computed using Sinkhorn's algorithm towards the closed form of $\mathrm{OT}_\sigma(\alpha, \beta)$ and $\mathrm{UOT}_\sigma(\alpha, \beta)$ (the theoretical limit is dashed) given by Theorem 1 and Theorem 3 for random Gaussians $\alpha, \beta$. For unbalanced OT, $\gamma = 1$.

**Transport plan visualization with $d = 1$.** Figure 2 confronts the expected theoretical plans (contours in black) given by theorems 1 and 3 to empirical ones (weights in shades of red) obtained

with Sinkhorn's algorithm using 2000 Gaussian samples. The density functions (black) and the empirical histograms (red) of $\alpha$ (resp. $\beta$) with 200 bins are displayed on the left (resp. top) of each transport plan. The red weights are computed via a 2d histogram of the transport plan returned by Sinkhorn's algorithm with (200 x 200) bins. Notice the blurring effect of $\varepsilon$ and increased mass transportation of the Gaussian tails in unbalanced transport with larger $\gamma$.

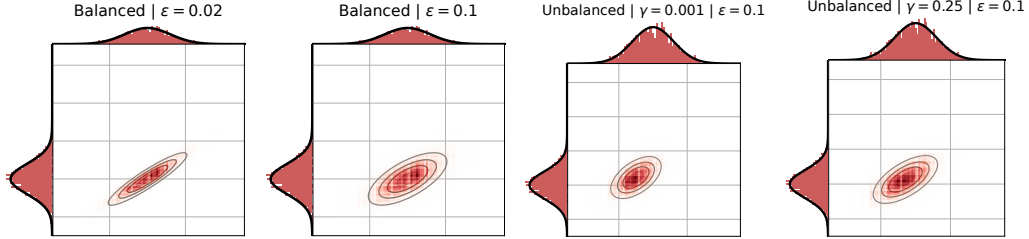

Figure 2: Effect of $\varepsilon$ in balanced OT and $\gamma$ in unbalanced OT. Empirical plans (red) correspond to the expected Gaussian contours depicted in black. Here $\alpha = \mathcal{N}(0, 0.04)$ and $\beta = m_\beta \mathcal{N}(0.5, 0.09)$ with $m_\beta = 1$ (balanced) and $m_\beta = 2$ (unbalanced). In unbalanced OT, the right tail of $\beta$ is not transported, and the mean of the transportation plan is shifted compared to that of the balanced case – as expected from Theorem 3 specially for low $\gamma$.

**Empirical estimation of the closed form mean and covariance of the unbalanced transport plan**
Figure 3 illustrates the convergence towards the closed form formulas of $\mu$ and $\mathbf{H}$ of theorem 3. For each dimension $d$ in [1, 2, 5, 10], we select a pair of Gaussians $\alpha \sim \mathcal{N}(\mathbf{a}, \mathbf{A})$ and $\beta \sim m_\beta \mathcal{N}(\mathbf{b}, \mathbf{B})$ with $m_\beta = 1.1$ and randomly generated means $\mathbf{a}, \mathbf{b}$ (uniform in $[-1, 1]^d$) and covariances $\mathbf{A}, \mathbf{B} \in S_{++}^d$ following the Wishart distribution $W_d(0.2 * \mathrm{Id}, d)$. We generate i.i.d datasets $\alpha_n \sim \mathcal{N}(\mathbf{a}, \mathbf{A})$ and $\beta_n \sim m_\beta \mathcal{N}(\mathbf{b}, \mathbf{B})$ with $n$ samples and compute $\mathrm{OT}_\sigma$ / $\mathrm{UOT}_\sigma$. We set $\varepsilon \overset{\text{def}}{=} 2\sigma^2 - 0.5$ and $\gamma = 0.1$. Using the obtained empirical Sinkhorn transportation plan, we computed its empirical mean $\mu_n$ and covariance matrix $\Sigma_n$ and display their relative $\ell_\infty$ distance to $\mu$ and $\mathbf{H}$ ($\Sigma$ in the figure) of theorem 3. The means and $\pm$ sd intervals are computed over 50 independent trials for each value of $n$.

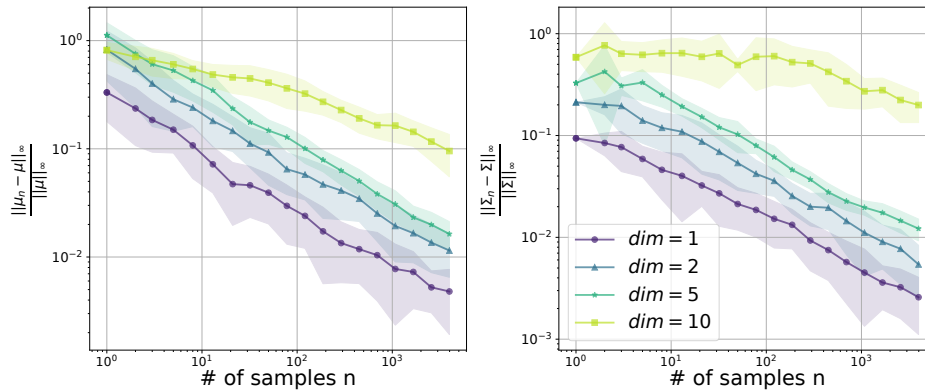

Figure 3: Numerical convergence the (n-samples) empirical estimation of the theoretical mean $\mu$ and covariance $\mathbf{H}$ of theorem 3. Empirical moments are computed computed using Sinkhorn's algorithm.

## Broader Impact

We expect this work to benefit research on sample complexity issues in regularized optimal transport, such as [24] for balanced regularized OT, and future work on unbalanced regularized OT. By providing the first continuous test-case, we hope that researchers will be able to better test their theoretical bounds and benchmark their methods.

## Acknowledgments

H. Janati, B. Muzellec and M. Cuturi were supported by a "Chaire d'excellence de l'IDEX Paris Saclay". H. Janati acknowledges the support of the ERC Starting Grant SLAB ERC-YStG-676943. The work of G. Peyré was supported by the European Research Council (ERC project NORIA) and by the French government under management of Agence Nationale de la Recherche as part of the "Investissements d'avenir" program, reference ANR19-P3IA-0001 (PRAIRIE 3IA Institute).

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
