[Supplementary Material]

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

# Appendix

## 5.1 The Newton-Schulz algorithm

The main bottleneck in computing $\mathbf{T^{AB}}$ is that of computing matrix square roots. This can be performed using singular value decomposition (SVD) or, as suggested in [39], using Newton-Schulz (NS) iterations [30, §5.3]. In particular, Newton-Schulz iterations have the advantage of yielding both roots, and inverse roots. Hence, to compute $\mathbf{T^{AB}}$, one would run NS a first time to obtain $\mathbf{A}^{\frac{1}{2}}$ and $\mathbf{A}^{-\frac{1}{2}}$, and a second time to get $(\mathbf{A}^{\frac{1}{2}}\mathbf{B}\mathbf{A}^{\frac{1}{2}})^{\frac{1}{2}}$.

In fact, as a direct application of [30, Theorem 5.2], one can even compute both $\mathbf{T^{AB}}$ and $\mathbf{T^{BA}} = (\mathbf{T^{AB}})^{-1}$ in a single run by initializing the Newton-Schulz algorithm

---

**Algorithm 1** NS Monge Iterations

---
**Input:** PSD matrix $\mathbf{A}, \mathbf{B}, \epsilon > 0$
   $\mathbf{Y} \leftarrow \frac{\mathbf{B}}{(1+\epsilon)\|\mathbf{B}\|}, \mathbf{Z} \leftarrow \frac{\mathbf{A}}{(1+\epsilon)\|\mathbf{A}\|}$
   **while** not converged **do**
      $\mathbf{T} \leftarrow (3\mathbf{I} - \mathbf{ZY})/2$
      $\mathbf{Y} \leftarrow \mathbf{YT}$
      $\mathbf{Z} \leftarrow \mathbf{TZ}$
   **end while**
   $\mathbf{Y} \leftarrow \sqrt{\frac{\|\mathbf{B}\|}{\|\mathbf{A}\|}}\mathbf{Y}, \mathbf{Z} \leftarrow \sqrt{\frac{\|\mathbf{A}\|}{\|\mathbf{B}\|}}\mathbf{Z}$
**Output:** $\mathbf{Y} = \mathbf{T^{AB}}, \mathbf{Z} = \mathbf{T^{BA}}$

---

with $\mathbf{A}$ and $\mathbf{B}$, as in Algorithm 1. Using (6), and noting that $\mathfrak{B}^2(\mathbf{A}, \mathbf{B}) = \mathrm{Tr}\mathbf{A} + \mathrm{Tr}\mathbf{B} - 2\mathrm{Tr}(\mathbf{T^{AB}A})$, this implies that a single run of NS is sufficient to compute $\mathfrak{B}^2(\mathbf{A}, \mathbf{B})$, $\nabla_{\mathbf{A}}\mathfrak{B}^2(\mathbf{A}, \mathbf{B})$ and $\nabla_{\mathbf{B}}\mathfrak{B}^2(\mathbf{A}, \mathbf{B})$ using basic matrix operations. The main advantage of Newton-Schultz over SVD is that it its efficient scalability on GPUs, as illustrated in Figure 4.

Newton-Schulz iterations are quadratically convergent under the condition $\| \mathrm{Id} - \left( \begin{smallmatrix} \mathbf{A} & 0 \\ 0 & \mathbf{B} \end{smallmatrix} \right)^2 \| < 1$, as shown in [30, Theorem 5.8]. To meet this condition, it is sufficient to rescale $\mathbf{A}$ and $\mathbf{B}$ so that their norms equal $(1 + \varepsilon)^{-1}$ for some $\varepsilon > 0$, as in the first step of Algorithm 1 (which can be skipped if $\|\mathbf{A}\| < 1$ (resp. $\|\mathbf{B}\| < 1$)). Finally, the output of the iterations are scaled back, using the homogeneity (resp. inverse homogonity) of eq. (5) w.r.t. $\mathbf{A}$ (resp. $\mathbf{B}$).

A rough theoretical analysis shows that both Newton-Schulz and SVD have a $O(d^3)$ complexity in the dimension. Figure 4 compares the running times of Newton-Schulz iterations and SVD on CPU or GPU used to compute both $\mathbf{A}^{\frac{1}{2}}$ and $\mathbf{A}^{-\frac{1}{2}}$. We simulate a batch of positive definite matrices $\mathbf{A}$ following the Wishart distribution $W(\mathrm{Id}_d, d)$ to which we add $0.1\,\mathrm{Id}$ to avoid numerical issues when computing inverse square roots. We display the average run-time of 50 different trials along with its $\pm$ std interval. Notice the different magnitudes between CPUs and GPUs. As a termination criterion, we first run EVD to obtain $\mathbf{A}_{evd}^{\frac{1}{2}}$ and $\mathbf{A}_{evd}^{-\frac{1}{2}}$ and stop the Newton-Schultz algorithm when its n-th running estimate $\mathbf{A}_n^{\frac{1}{2}}$ verifies: $\|\mathbf{A}_n^{\frac{1}{2}} - \mathbf{A}_{evd}^{\frac{1}{2}}\|_1 \leq 10^{-4}$. Notice the different order of magnitude between CPUs and GPUs. Moreover, the computational advantage of Newton-Schultz on GPUs can be further increased when computing multiple square roots in parallel.

Figure 4: Average run-time of Newton-Schulz and EVD to compute on CPUs and GPUs.

Figure 5: Effect of regularization on transportation plans. When $\sigma$ goes to 0 (left), the transportation plan concentrates on the graph of the linear Monge map. When $\sigma$ goes to infinity (right), the transportation plan converges to the independent coupling.

## 5.2 Effects of regularization strength.

We provide numerical experiments to illustrate the behaviour of transportation plans and corresponding distances as $\sigma$ goes to 0 or to infinity. As can be seen from eq. (14), when $\sigma \to 0$ we recover the Wasserstein-Bures distance (3), and the optimal transportation plan converges to the Monge map (5). When on the contrary $\sigma \to \infty$, Sinkhorn divergences $\mathfrak{S}_\varepsilon(\alpha, \beta) \overset{\text{def}}{=} \mathrm{OT}_\varepsilon(\alpha, \beta) - \frac{1}{2}(\mathrm{OT}_\varepsilon(\alpha, \alpha) + \mathrm{OT}_\varepsilon(\beta, \beta))$ convergence to MMD with a $-c$ kernel (where $c$ is the optimal transport ground cost) [27]. With a $-\ell_2$ kernel, MMD is degenerate and equals 0 for centered measures.

Figure 7: Bures, Sinkhorn-Bures, and Euclidean geodesics. Sinkhorn-Bures trajectories converge to Bures geodesics as $\sigma$ goes to 0, and to Euclidean geodesics as $\sigma$ goes to infinity.

Figure 6: Numerical convergence of $\mathfrak{B}_\sigma^2(\mathbf{A}, \mathbf{B}) - \frac{1}{2}(\mathfrak{B}_\sigma^2(\mathbf{A}, \mathbf{A}) + \mathfrak{B}_\sigma^2(\mathbf{B}, \mathbf{B}))$ to $\mathfrak{B}^2(\mathbf{A}, \mathbf{B})$ as $\sigma$ goes to $0$ and to $0$ as $\sigma$ goes to infinity.

## 5.3 Proofs of technical results

We provide in this appendix the proofs of the results in the paper, as well as some technical lemmas used in solving Sinkhorn's equations in closed form.

**Proof of Lemma 1.**

*Proof.* Let $\mathrm{d}\bar{\alpha}(x) = \mathrm{d}\alpha(x + \mathbf{a})$ (resp. $\mathrm{d}\bar{\beta}(y) = \mathrm{d}\beta(y + \mathbf{b})$, $\mathrm{d}\bar{\pi}(x, y) = \mathrm{d}\pi(x + \mathbf{a}, y + \mathbf{b})$, such that $\bar{\alpha}, \bar{\beta}$ and $\bar{\pi}$ are centered. Then, $\forall \pi \in \Pi(\alpha, \beta)$,

  (i)  $\bar{\pi} \in \Pi(\bar{\alpha}, \bar{\beta})$,

  (ii) $\mathrm{KL}(\pi \| \alpha \otimes \beta) = \mathrm{KL}(\bar{\pi} \| \bar{\alpha} \otimes \bar{\beta})$

  (iii) $\int_{\mathbb{R}^d \times \mathbb{R}^d} \|x - y\|^2 \mathrm{d}\bar{\pi}(x, y) = \int_{\mathbb{R}^d \times \mathbb{R}^d} \|(x - \mathbf{a}) - (y - \mathbf{b})\|^2 \mathrm{d}\pi(x, y) = \|\mathbf{a} - \mathbf{b}\|^2 + \int_{\mathbb{R}^d \times \mathbb{R}^d} \|x - y\|^2 \mathrm{d}\pi(x, y)$

Plugging (i)-(iii) into (7), we get $\mathrm{OT}_\sigma(\alpha, \beta) = \mathrm{OT}_\sigma(\bar{\alpha}, \bar{\beta}) + \|\mathbf{a} - \mathbf{b}\|^2$.  □

**Proof of Proposition 1.**

*Proof.* The exponent inside the integral can be written as:

$$e^{\frac{-\|x-y\|^2}{2\sigma^2}+h(y)}\mathrm{d}\alpha(y) \propto e^{\frac{-\|x-y\|^2}{2\sigma^2}-\frac{1}{2}(y^\top \mathbf{X}y - y^\top \mathbf{A}^{-1}y)}\mathrm{d}y$$

$$\propto e^{-\frac{1}{2}(y^\top(\frac{\mathrm{Id}}{\sigma^2}+\mathbf{X}+\mathbf{A}^{-1})y)+\frac{x^\top y}{\sigma^2}}\mathrm{d}y$$

which is integrable if and only if $\mathbf{X} + \mathbf{A}^{-1} + \frac{1}{\sigma^2}\,\mathrm{Id} \succ 0$. Moreover, up to a multiplicative factor, the exponentiated Sinkhorn transform is equivalent to a Gaussian convolution of an exponentiated quadratic form. Lemma 5 applies:

$$e^{-T_\alpha(h)} = \int_{\mathbb{R}^d} e^{\frac{-\|x-y\|^2}{2\sigma^2}+f(y)}\mathrm{d}\alpha(y)$$

$$\propto \int_{\mathbb{R}^d} e^{\frac{-\|x-y\|^2}{2\sigma^2}+\mathcal{Q}(\mathbf{X})(y)+\mathcal{Q}(\mathbf{A}^{-1})(y)}\mathrm{d}y$$

$$\propto \exp\left(\mathcal{Q}\left(\tfrac{\mathrm{Id}}{\sigma^2}\right)\right) \star \exp\left(\mathcal{Q}(\mathbf{X}) + \mathcal{Q}(\mathbf{A}^{-1})\right)$$

$$\propto \exp\left(\mathcal{Q}\left(\tfrac{\mathrm{Id}}{\sigma^2}\right)\right) \star \exp\left(\mathcal{Q}(\mathbf{X} + \mathbf{A}^{-1})\right)$$

$$\propto \exp\left(\mathcal{Q}((\mathrm{Id} + \sigma^2\mathbf{X} + \sigma^2\mathbf{A}^{-1})^{-1}(\mathbf{X} + \mathbf{A}^{-1}))\right).$$

$$\propto \exp\left(\mathcal{Q}(\tfrac{1}{\sigma^2}\mathbf{X}'^{-1}(\mathbf{X}' - \mathrm{Id}))\right).$$

$$\propto \exp\left(\mathcal{Q}(\tfrac{1}{\sigma^2}(\mathrm{Id} - \mathbf{X}'^{-1}))\right).$$

Therefore $T_\alpha(h)$ is up to an additive constant given by $\mathcal{Q}(\frac{1}{\sigma^2}(\mathbf{X}'^{-1} - \mathrm{Id}))$.

Finally, since $\mathbf{B}$ and $\mathbf{X}'$ are positive definite, the positivity condition of $\mathbf{Y}'$ holds and $T_\beta$ can be applied again to get $T_\beta(T_\alpha(h))$. □

**Proof of Proposition 2.**

*Proof.* Let $\mathbf{U}_0 = \mathbf{V}_0 = 0$. Applying Proposition 1 to the initial pair of potentials $\mathcal{Q}(\mathbf{U}_0), \mathcal{Q}(\mathbf{V}_0)$ leads to the sequence of quadratic Sinkhorn potentials $\frac{f_n}{2\sigma^2} = \mathcal{Q}(\mathbf{U}_n)$ and $\frac{g_n}{2\sigma^2} = \mathcal{Q}(\mathbf{V}_n)$ where:

$$\mathbf{V}_{n+1} = \frac{1}{\sigma^2}((\sigma^2\mathbf{U}_n + \sigma^2\mathbf{A}^{-1} + \mathrm{Id})^{-1} - \mathrm{Id})$$

$$\mathbf{U}_{n+1} = \frac{1}{\sigma^2}((\sigma^2\mathbf{V}_{n+1} + \sigma^2\mathbf{B}^{-1} + \mathrm{Id})^{-1} - \mathrm{Id}).$$

The change of variable:

$$\mathbf{F}_n = \sigma^2\mathbf{U}_n + \sigma^2\mathbf{A}^{-1} + \mathrm{Id}$$

$$\mathbf{G}_n = \sigma^2\mathbf{V}_n + \sigma^2\mathbf{B}^{-1} + \mathrm{Id}$$

leads to (17).

We turn to show that this algorithm converges. First, note that since $\mathbf{F}_0, \mathbf{G}_0 \in \mathcal{S}^d_{++}$, a straightforward induction shows that $\forall n \geq 0, \mathbf{F}_n, \mathbf{G}_n \in \mathcal{S}^d_{++}$. Next, let us write the decoupled iteration on $\mathbf{F}$:

$$\mathbf{F} \leftarrow \sigma^2\mathbf{A}^{-1} + (\sigma^2\mathbf{B}^{-1} + \mathbf{F}^{-1})^{-1} \tag{42}$$

Let $\forall \mathbf{X} \in \mathcal{S}^d_{++}, \phi(\mathbf{X}) \overset{\mathrm{def}}{=} \sigma^2\mathbf{A}^{-1} + (\sigma^2\mathbf{B}^{-1} + \mathbf{X}^{-1})^{-1} \in \mathcal{S}^d_{++}$. The first differential of $\phi$ admits the following expression:

$$\forall \mathbf{X} \in \mathcal{S}^d_{++}, \forall \mathbf{H} \in \mathbb{R}^{d \times d}, D\phi(\mathbf{X})[\mathbf{H}] = (\mathrm{Id} + \sigma^2\mathbf{X}\mathbf{B}^{-1})^{-1}\mathbf{H}(\sigma^2\mathbf{B}^{-1}\mathbf{X} + \mathrm{Id})^{-1}. \tag{43}$$

Hence, $\|D\phi(\mathbf{X})[\mathbf{H}]\|_{\mathrm{op}} \leq \|(\mathrm{Id} + \sigma^2\mathbf{X}\mathbf{B}^{-1})^{-1}\|^2_{\mathrm{op}}\|\mathbf{H}\|_{\mathrm{op}}$. Plugging $\mathbf{H} = \mathrm{Id}$, we get that $\|D\phi(\mathbf{X})\|_{\mathrm{op}} = \|(\mathrm{Id} + \sigma^2\mathbf{X}\mathbf{B}^{-1})^{-1}\|^2_{\mathrm{op}}$. Finally, by matrix similarity

$$\|(\mathrm{Id} + \sigma^2\mathbf{X}\mathbf{B}^{-1})^{-1}\|_{\mathrm{op}} = \|(\mathrm{Id} + \sigma^2\mathbf{B}^{-\frac{1}{2}}\mathbf{X}\mathbf{B}^{-\frac{1}{2}})^{-1}\|_{\mathrm{op}} < 1 \ ,$$

which implies that $\|D\phi(\mathbf{X})\|_{\mathrm{op}} < 1$ for $\mathbf{X} \in \mathcal{S}^d_{++}$ and $\sigma^2 > 0$. The same arguments hold for the iterates $(\mathbf{G}_n)_{n \geq 0}$.

From (42) and using Weyl's inequality, we can bound the smallest eigenvalue of $\mathbf{F}_n$ from under: $\forall n, \lambda_d(\mathbf{F}_n) \geq \frac{\sigma^2}{\lambda_1(\mathbf{A})}$ (where $\lambda_d(\mathbf{F})$ is the smallest eigenvalue of $\mathbf{F}$ and $\lambda_1(\mathbf{A})$ is the biggest eigenvalue of $\mathbf{A}$). Hence, the iterates live in $\mathcal{A} \stackrel{\text{def}}{=} \mathcal{S}^d_{++} \cap \{\mathbf{X} : \lambda_d(\mathbf{X}) \geq \frac{\sigma^2}{\lambda_1(\mathbf{A})}\}$. Finally, for all $\mathbf{X} \in \mathcal{A}$,

$$
\begin{aligned}
\|(\mathrm{Id} + \sigma^2 \mathbf{B}^{-\frac{1}{2}} \mathbf{X} \mathbf{B}^{-\frac{1}{2}})^{-1}\|_{\mathrm{op}} &= \frac{1}{\lambda_d(\mathrm{Id} + \sigma^2 \mathbf{B}^{-1/2} \mathbf{X} \mathbf{B}^{-1/2})} \\
&= \frac{1}{1 + \sigma^2 \lambda_d(\mathbf{B}^{-1/2} \mathbf{X} \mathbf{B}^{-1/2})} \\
&\leq \frac{1}{1 + \sigma^2 \lambda_d(\mathbf{B}^{-1}) \lambda_d(\mathbf{X})} \\
&\leq \frac{1}{1 + \frac{\sigma^4}{\lambda_1(\mathbf{B})\lambda_1(\mathbf{A})}}
\end{aligned}
$$

Which proves the uniform bound ∎

$\square$

**Proof of Proposition 3.**

*Proof.* Combining the two equations in (20) yields

$$
\begin{aligned}
\mathbf{G} &= \sigma^2 \mathbf{B}^{-1} + (\mathbf{G}^{-1} + \sigma^2 \mathbf{A}^{-1})^{-1} \\
\Leftrightarrow \mathbf{G} \mathbf{A}^{-1} &= \sigma^2 \mathbf{B}^{-1} \mathbf{A}^{-1} + (\mathbf{A} \mathbf{G}^{-1} + \sigma^2 \mathrm{Id})^{-1} \\
\Leftrightarrow \mathbf{C}^{-1} &= \sigma^2 (\mathbf{A} \mathbf{B})^{-1} + (\mathbf{C} + \sigma^2 \mathrm{Id})^{-1} \\
\Leftrightarrow \mathbf{C}^{-1}(\mathbf{C} + \sigma^2 \mathrm{Id}) &= \sigma^2 (\mathbf{A} \mathbf{B})^{-1}(\mathbf{C} + \sigma^2 \mathrm{Id}) + \mathrm{Id} \\
\Leftrightarrow \mathrm{Id} + \sigma^2 \mathbf{C}^{-1} &= \sigma^2 (\mathbf{A} \mathbf{B})^{-1}(\mathbf{C} + \sigma^2 \mathrm{Id}) + \mathrm{Id} \\
\Leftrightarrow \mathbf{C} + \sigma^2 \mathrm{Id} &= \sigma^2 (\mathbf{A} \mathbf{B})^{-1}(\mathbf{C} + \sigma^2 \mathrm{Id}) \mathbf{C} + \mathbf{C} \\
\Leftrightarrow \mathbf{C}^2 + \sigma^2 \mathbf{C} - \mathbf{A} \mathbf{B} &= 0. \tag{44}
\end{aligned}
$$

Given that $\mathbf{A}$ and $\mathbf{G}^{-1}$ are positive, their product $\mathbf{C} = \mathbf{A} \mathbf{G}^{-1}$ can be written: $\mathbf{A} \mathbf{G}^{-1} = \mathbf{A}^{\frac{1}{2}}(\mathbf{A}^{\frac{1}{2}} \mathbf{G}^{-1} \mathbf{A}^{\frac{1}{2}}) \mathbf{A}^{-\frac{1}{2}}$, thus $\mathbf{A} \mathbf{G}^{-1}$ is similar to the positive matrix $\mathbf{A}^{\frac{1}{2}} \mathbf{G}^{-1} \mathbf{A}^{\frac{1}{2}}$. Therefore, one can write an eigenvalue decomposition of $\mathbf{C} = \mathbf{P} \Sigma \mathbf{P}^{-1}$ with a positive diagonal matrix $\Sigma$. Substituting in (21), it follows that $\mathbf{C}$ and $\mathbf{A} \mathbf{B}$ share the same eigenvectors with modified eigenvalues. Thus, it is sufficient to find the real roots of the polynomial $x \mapsto x^2 + \sigma^2 x - ab$ with $a, b \in \mathbb{R}_{++}$ which are given by: $x_1 = -\frac{\sigma^2}{2} - \sqrt{ab + \frac{\sigma^4}{4}}$ and $x_2 = -\frac{\sigma^2}{2} + \sqrt{ab + \frac{\sigma^4}{4}}$. Since $\mathbf{C}$ is the product of the positive definite matrices $\mathbf{G}^{-1}$ and $\mathbf{A}$, its eigenvalues are all positive. Discarding the negative root, the closed form follows immediately.

Indeed, by direct calculation, computing the square of the solution $\mathbf{C}$ leads to the equation (21):

$$
\begin{aligned}
\mathbf{C}^2 &= \mathbf{A} \mathbf{B} + \frac{\sigma^4}{2} \mathrm{Id} - \sigma^2 \left( \mathbf{A} \mathbf{B} + \frac{\sigma^4}{4} \mathrm{Id} \right)^{\frac{1}{2}} \\
&= \mathbf{A} \mathbf{B} - \sigma^2 \mathbf{C}.
\end{aligned}
$$

The second equality is obtained by observing that

$$
(\mathbf{A}^{\frac{1}{2}}(\mathbf{A}^{\frac{1}{2}} \mathbf{B} \mathbf{A}^{\frac{1}{2}} + \frac{\sigma^4}{4} \mathrm{Id})^{\frac{1}{2}} \mathbf{A}^{-\frac{1}{2}})^2 = \mathbf{A}^{\frac{1}{2}}(\mathbf{A}^{\frac{1}{2}} \mathbf{B} \mathbf{A}^{\frac{1}{2}} + \frac{\sigma^4}{4} \mathrm{Id}) \mathbf{A}^{-\frac{1}{2}} = \mathbf{A} \mathbf{B} + \frac{\sigma^4}{4} \mathrm{Id},
$$

i.e. that

$$
\left( \mathbf{A} \mathbf{B} + \frac{\sigma^4}{4} \mathrm{Id} \right)^{\frac{1}{2}} = \mathbf{A}^{\frac{1}{2}}(\mathbf{A}^{\frac{1}{2}} \mathbf{B} \mathbf{A}^{\frac{1}{2}} + \frac{\sigma^4}{4} \mathrm{Id})^{\frac{1}{2}} \mathbf{A}^{-\frac{1}{2}}.
$$

$\square$

**Proof of Lemma 2**

*Proof.* It follows from elementary properties of Gaussian measures that the first and second marginals of $\pi$ are respectively $\alpha$ and $\beta$. Hence,

$$\int_{\mathbb{R}^d \times \mathbb{R}^d} \|x - y\|^2 \mathrm{d}\pi(x, y) = \int_{\mathbb{R}^d \times \mathbb{R}^d} \|x\|^2 \mathrm{d}\pi(x, y) + \int_{\mathbb{R}^d \times \mathbb{R}^d} \|y\|^2 \mathrm{d}\pi(x, y) - 2 \int_{\mathbb{R}^d \times \mathbb{R}^d} \langle x,\, y \rangle \mathrm{d}\pi(x, y) \tag{45}$$

$$= \int_{\mathbb{R}^d} \|x\|^2 \mathrm{d}\alpha(x) + \int_{\mathbb{R}^d} \|y\|^2 \mathrm{d}\beta(y) - 2 \int_{\mathbb{R}^d \times \mathbb{R}^d} \langle x,\, y \rangle \mathrm{d}\pi(x, y) \tag{46}$$

$$= \mathrm{Tr}(\mathbf{A}) + \mathrm{Tr}(\mathbf{B}) - 2\mathrm{Tr}(\mathbf{C}). \tag{47}$$

Next, using the closed form expression of the Kullback-Leibler divergence between Gaussian measures,

$$\mathrm{KL}\left(\pi \| \alpha \otimes \beta \right) = \tfrac{1}{2} \left( \mathrm{Tr}\left[ \left( \begin{smallmatrix} \mathbf{A} & 0 \\ 0 & \mathbf{B} \end{smallmatrix} \right)^{-1} \left( \begin{smallmatrix} \mathbf{A} & \mathbf{C} \\ \mathbf{C}^T & \mathbf{B} \end{smallmatrix} \right) \right] - 2n + \log \det \left( \begin{smallmatrix} \mathbf{A} & 0 \\ 0 & \mathbf{B} \end{smallmatrix} \right) - \log \det \left( \begin{smallmatrix} \mathbf{A} & \mathbf{C} \\ \mathbf{C}^T & \mathbf{B} \end{smallmatrix} \right) \right) \tag{48}$$

$$= \tfrac{1}{2} \left( \log \det \mathbf{A} + \log \det \mathbf{B} - \log \det \left( \begin{smallmatrix} \mathbf{A} & \mathbf{C} \\ \mathbf{C}^T & \mathbf{B} \end{smallmatrix} \right) \right). \tag{49}$$

$\square$

**Optimal transport plan and $\mathrm{OT}_\sigma$**

$$\frac{\mathrm{d}\pi}{\mathrm{d}x\mathrm{d}y}(x, y) = \exp\left( \frac{f(x) + g(y) - \|x - y\|^2}{2\sigma^2} \right) \frac{\mathrm{d}\alpha}{\mathrm{d}x}(x) \frac{\mathrm{d}\beta}{\mathrm{d}y}(y)$$

$$\propto \exp\left( \mathcal{Q}(\mathbf{A}^{-1})(x) + \frac{f(x) + g(y) - \|x - y\|^2}{2\sigma^2} + \mathcal{Q}(\mathbf{B}^{-1})(y) \right)$$

$$\propto \exp\left( \mathcal{Q}(\mathbf{U} + \mathbf{A}^{-1})(x) + \mathcal{Q}(\mathbf{V} + \mathbf{B}^{-1})(y) + \mathcal{Q}\left( \begin{smallmatrix} \frac{\mathrm{Id}}{\sigma^2} & -\frac{\mathrm{Id}}{\sigma^2} \\ -\frac{\mathrm{Id}}{\sigma^2} & \frac{\mathrm{Id}}{\sigma^2} \end{smallmatrix} \right)(x, y) \right)$$

$$= \exp\left( \mathcal{Q}\left( \begin{smallmatrix} \mathbf{U}+\mathbf{A}^{-1} & 0 \\ 0 & \mathbf{V}+\mathbf{B}^{-1} \end{smallmatrix} \right)(x, y) + \mathcal{Q}\left( \begin{smallmatrix} \frac{\mathrm{Id}}{\sigma^2} & -\frac{\mathrm{Id}}{\sigma^2} \\ -\frac{\mathrm{Id}}{\sigma^2} & \frac{\mathrm{Id}}{\sigma^2} \end{smallmatrix} \right)(x, y) \right)$$

$$= \exp\left( \mathcal{Q}\left( \begin{smallmatrix} \frac{\mathrm{Id}}{\sigma^2}+\mathbf{U}+\mathbf{A}^{-1} & -\frac{\mathrm{Id}}{\sigma^2} \\ -\frac{\mathrm{Id}}{\sigma^2} & \frac{\mathrm{Id}}{\sigma^2}+\mathbf{V}+\mathbf{B}^{-1} \end{smallmatrix} \right)(x, y) \right)$$

$$= \exp\left( \mathcal{Q}\left( \begin{smallmatrix} \frac{\mathbf{F}}{\sigma^2} & -\frac{\mathrm{Id}}{\sigma^2} \\ -\frac{\mathrm{Id}}{\sigma^2} & \frac{\mathbf{G}}{\sigma^2} \end{smallmatrix} \right)(x, y) \right)$$

$$= \exp\left( \mathcal{Q}(\Gamma)(x, y) \right)$$

with $\Gamma \overset{\mathrm{def}}{=} \left( \begin{smallmatrix} \frac{\mathbf{F}}{\sigma^2} & -\frac{\mathrm{Id}}{\sigma^2} \\ -\frac{\mathrm{Id}}{\sigma^2} & \frac{\mathbf{G}}{\sigma^2} \end{smallmatrix} \right)$. Moreover, since $\frac{\mathbf{G}}{2\sigma^2} \succ 0$, and its Schur complement satisfies $\frac{\mathbf{F}}{\sigma^2} - \frac{1}{\sigma^2}\mathbf{G}^{-1} = \mathbf{A}^{-1} \succ 0$, we have that $\Gamma \succ 0$. Therefore $\pi$ is a Gaussian $\mathcal{N}(\mathbf{H})$ with the covariance matrix given by the block inverse formula:

$$\mathbf{H} = \Gamma^{-1} \tag{50}$$

$$= \sigma^2 \begin{pmatrix} (\mathbf{F} - \mathbf{G}^{-1})^{-1} & (\mathbf{GF} - \mathrm{Id})^{-1} \\ (\mathbf{FG} - \mathrm{Id})^{-1} & (\mathbf{G} - \mathbf{F}^{-1})^{-1} \end{pmatrix} \tag{51}$$

$$= \begin{pmatrix} \mathbf{A} & \mathbf{C} \\ \mathbf{C}^\top & \mathbf{B} \end{pmatrix}, \tag{52}$$

where we used the optimality equations (20) and the definition of $\mathbf{C} = \mathbf{AG}^{-1}$.

We can now conclude the proof of Theorem 1 by computing $\mathrm{OT}_\sigma(\alpha, \beta)$ using Lemma 2. Let $\mathbf{R} = \mathbf{A}^{\frac{1}{2}} \mathbf{B} \mathbf{A}^{\frac{1}{2}}$. Using the closed form expression of $\mathbf{C}$ in (22), it first holds that

$$\mathbf{Z} \overset{\mathrm{def}}{=} \mathbf{A}^{-\frac{1}{2}} \mathbf{C} \mathbf{A}^{\frac{1}{2}} = \left( \mathbf{R} + \tfrac{\sigma^4}{4} \mathrm{Id} \right)^{\frac{1}{2}} - \tfrac{\sigma^2}{2} \mathrm{Id}. \tag{53}$$

Moreover, since $\mathbf{R} = \mathbf{R}^\top$, it holds that $\mathbf{Z} = \mathbf{Z}^\top$. Hence,

$$
\begin{aligned}
\det \left( \begin{smallmatrix} \mathbf{A} & \mathbf{C} \\ \mathbf{C}^T & \mathbf{B} \end{smallmatrix} \right) &= \det(\mathbf{A})\det(\mathbf{B} - \mathbf{C}^\top \mathbf{A}^{-1}\mathbf{C}) \\
&= \det(\mathbf{A}^{\frac{1}{2}}\mathbf{B}\mathbf{A}^{\frac{1}{2}} - \mathbf{A}^{\frac{1}{2}}\mathbf{C}^\top \mathbf{A}^{-1}\mathbf{C}\mathbf{A}^{\frac{1}{2}}) \\
&= \det(\mathbf{R} - \mathbf{Z}^\top \mathbf{Z}) \\
&= \det(\mathbf{R} - \mathbf{Z}^2) \qquad\qquad (54) \\
&= \det(\sigma^2(\mathbf{R} + \frac{\sigma^4}{4}\,\mathrm{Id})^{\frac{1}{2}} - \frac{\sigma^4}{2}\,\mathrm{Id}) \\
&= (\frac{\sigma^2}{2})^d \det((4\mathbf{R} + \sigma^4\,\mathrm{Id})^{\frac{1}{2}} - \sigma^2\,\mathrm{Id}).
\end{aligned}
$$

Since the matrices inside the determinant commute, we can use the identity $\mathbf{P} - \mathbf{Q} = (\mathbf{P}^2 - \mathbf{Q}^2)(\mathbf{P} + \mathbf{Q})^{-1}$ to get rid of the negative sign. Equation (54) then becomes:

$$
\begin{aligned}
(\frac{\sigma^2}{2})^d \det((4\mathbf{R} + \sigma^4\,\mathrm{Id})^{\frac{1}{2}} - \sigma^2\,\mathrm{Id}) &= (\frac{\sigma^2}{2})^d \det(4\mathbf{R})\det\left(((4\mathbf{R} + \sigma^4\,\mathrm{Id})^{\frac{1}{2}} + \sigma^2\,\mathrm{Id})^{-1}\right) \\
&= (2\sigma^2)^d \det(\mathbf{AB})\det\left(((4\mathbf{R} + \sigma^4\,\mathrm{Id})^{\frac{1}{2}} + \sigma^2\,\mathrm{Id})^{-1}\right).
\end{aligned}
$$

Plugging this expression in (25), the determinant of $\mathbf{A}$ and $\mathbf{B}$ cancel out and we finally get:

$$
\begin{aligned}
\mathfrak{B}_\sigma^2(\mathbf{A}, \mathbf{B}) = \mathrm{Tr}(\mathbf{A}) + \mathrm{Tr}(\mathbf{B}) - \mathrm{Tr}(4\mathbf{A}^{\frac{1}{2}}\mathbf{B}\mathbf{A}^{\frac{1}{2}} + \sigma^4\,\mathrm{Id})^{\frac{1}{2}} + d\sigma^2 - \\
\sigma^2 d\log(2\sigma^2) + \sigma^2 \log\det\left((4\mathbf{A}^{\frac{1}{2}}\mathbf{B}\mathbf{A}^{\frac{1}{2}} + \sigma^4\,\mathrm{Id})^{\frac{1}{2}} + \sigma^2\,\mathrm{Id}\right).
\end{aligned}
$$

**Proof of Proposition 4**

*Proof.* Using Lemma 2, eq. (7) becomes

$$
\mathfrak{B}_\sigma^2(\mathbf{A}, \mathbf{B}) = \min_{\mathbf{C}:\left( \begin{smallmatrix} \mathbf{A} & \mathbf{C} \\ \mathbf{C}^T & \mathbf{B} \end{smallmatrix} \right) \geq 0} \left\{ \mathrm{Tr}(\mathbf{A}) + \mathrm{Tr}(\mathbf{B}) - 2\mathrm{Tr}(\mathbf{C}) + \sigma^2(\log\det\mathbf{A} + \log\det\mathbf{B} - \log\det\left( \begin{smallmatrix} \mathbf{A} & \mathbf{C} \\ \mathbf{C}^T & \mathbf{B} \end{smallmatrix} \right)) \right\},
$$

which gives eq. (26). Let us now prove eq. (27). A necessary and sufficient condition for $\left( \begin{smallmatrix} \mathbf{A} & \mathbf{C} \\ \mathbf{C}^T & \mathbf{B} \end{smallmatrix} \right) \geq 0$ is that there exists a contraction $\mathbf{K}$ (i.e. $\mathbf{K} \in \mathbb{R}^d : \|\mathbf{K}\|_{\mathrm{op}} \leq 1$) such that $\mathbf{C} = \mathbf{A}^{\frac{1}{2}}\mathbf{K}\mathbf{B}^{\frac{1}{2}}$ [4, Ch. 1].[1]
With this parameterization, we have (using Schur complements) that

$$
\begin{aligned}
\det \left( \begin{smallmatrix} \mathbf{A} & \mathbf{C} \\ \mathbf{C}^T & \mathbf{B} \end{smallmatrix} \right) &= \det\mathbf{B}\det(\mathbf{A} - \mathbf{C}\mathbf{B}^{-1}\mathbf{C}^\top) \\
&= \det\mathbf{B}\det\mathbf{A}\det(\mathrm{Id} - \mathbf{K}\mathbf{K}^\top)
\end{aligned}
$$

Hence, injecting this in Equation (26), we have the following equivalent problem:

$$
\mathfrak{B}_\sigma^2(\mathbf{A}, \mathbf{B}) = \min_{\mathbf{K} \in \mathbb{R}^{d \times d}: \|\mathbf{K}\|_{\mathrm{op}} \leq 1} \mathrm{Tr}\mathbf{A} + \mathrm{Tr}\mathbf{B} - 2\mathrm{Tr}\mathbf{A}^{\frac{1}{2}}\mathbf{K}\mathbf{B}^{\frac{1}{2}} - \sigma^2 \ln\det(\mathrm{Id} - \mathbf{K}\mathbf{K}^\top) \qquad (55)
$$

Let's prove that both problems are convex.

- (26): The set $\{\mathbf{C} : \left( \begin{smallmatrix} \mathbf{A} & \mathbf{C} \\ \mathbf{C}^T & \mathbf{B} \end{smallmatrix} \right) \geq 0\}$ is convex, since $\left( \begin{smallmatrix} \mathbf{A} & \mathbf{C}_1 \\ \mathbf{C}_1^T & \mathbf{B} \end{smallmatrix} \right) \geq 0$ and $\left( \begin{smallmatrix} \mathbf{A} & \mathbf{C}_2 \\ \mathbf{C}_2^T & \mathbf{B} \end{smallmatrix} \right) \geq 0$ implies that $\left( \begin{smallmatrix} \mathbf{A} & (1-\theta)\mathbf{C}_1 + \theta\mathbf{C}_2 \\ (1-\theta)\mathbf{C}_1^T + \theta\mathbf{C}_2^T & \mathbf{B} \end{smallmatrix} \right) = (1-\theta)\left( \begin{smallmatrix} \mathbf{A} & \mathbf{C}_1 \\ \mathbf{C}_1^T & \mathbf{B} \end{smallmatrix} \right) + \theta\left( \begin{smallmatrix} \mathbf{A} & \mathbf{C}_2 \\ \mathbf{C}_2^T & \mathbf{B} \end{smallmatrix} \right) \geq 0$. Following the same decomposition, the concavity of the $\log\det$ function implies that $\mathbf{C} \to \log\det\left( \begin{smallmatrix} \mathbf{A} & \mathbf{C} \\ \mathbf{C}^T & \mathbf{B} \end{smallmatrix} \right)$ is concave, and hence that the objective function of (26) is convex.

- (27): The ball $\mathcal{B}_{\mathrm{op}} \stackrel{\mathrm{def}}{=} \{\mathbf{K} \in \mathbb{R}^{d \times d} : \|\mathbf{K}\|_{\mathrm{op}} \leq 1\}$ is obviously convex. Hence, there remains to prove that $f(\mathbf{K}) : \mathbf{K} \in \mathcal{B}_{\mathrm{op}} \to \log\det(\mathrm{Id} - \mathbf{K}\mathbf{K}^\top)$ is concave. Indeed, it holds that $f(\mathbf{K}) = \log\det\left( \begin{smallmatrix} \mathrm{Id} & \mathbf{K} \\ \mathbf{K}^T & \mathrm{Id} \end{smallmatrix} \right)$. Hence, $\forall \mathbf{K}, \mathbf{H} \in \mathcal{B}_{\mathrm{op}}, \forall t \in [0,1]$,

$$
\begin{aligned}
f((1-t)\mathbf{K} + t\mathbf{H}) &= \log\det\left\{ (1-t)\left( \begin{smallmatrix} \mathrm{Id} & \mathbf{K} \\ \mathbf{K}^T & \mathrm{Id} \end{smallmatrix} \right) + t\left( \begin{smallmatrix} \mathrm{Id} & \mathbf{H} \\ \mathbf{H}^T & \mathrm{Id} \end{smallmatrix} \right) \right\} \\
&\geq (1-t)\log\det\left( \begin{smallmatrix} \mathrm{Id} & \mathbf{K} \\ \mathbf{K}^T & \mathrm{Id} \end{smallmatrix} \right) + t\log\det\left( \begin{smallmatrix} \mathrm{Id} & \mathbf{H} \\ \mathbf{H}^T & \mathrm{Id} \end{smallmatrix} \right) \\
&= (1-t)f(\mathbf{K}) + tf(\mathbf{H}),
\end{aligned}
$$

where the second line follows from the concavity of $\log\det$.

□

**Proof of Proposition 5**

*Proof.* By Proposition 4, (26) is convex, hence strong duality holds. Ignoring the terms not depending on $\mathbf{C}$, problem (26) can be written using the redundant parameterization $\mathbf{X} = \left( \begin{smallmatrix} \mathbf{X}_1 & \mathbf{X}_2 \\ \mathbf{X}_3 & \mathbf{X}_4 \end{smallmatrix} \right)$:

$$\mathfrak{D}(\mathbf{A}, \mathbf{B}) \overset{\text{def}}{=} \min_{\substack{\mathbf{X} \succ 0 \\ \mathbf{X}_1 = \mathbf{A}, \mathbf{X}_4 = \mathbf{B}}} - \operatorname{Tr}(\mathbf{X}_2) - \operatorname{Tr}(\mathbf{X}_3) - \sigma^2 \log \det(\mathbf{X}) \tag{56}$$

$$= \min_{\substack{\mathbf{X} \succ 0 \\ \mathbf{X}_1 = \mathbf{A}, \mathbf{X}_4 = \mathbf{B}}} - \langle \mathbf{X}, \left( \begin{smallmatrix} 0 & \mathrm{Id} \\ \mathrm{Id} & 0 \end{smallmatrix} \right) \rangle - \sigma^2 \log \det(\mathbf{X}) \tag{57}$$

$$= \min_{\substack{\mathbf{X} \succ 0 \\ \mathbf{X}_1 = \mathbf{A}, \mathbf{X}_4 = \mathbf{B}}} \mathcal{F}(\mathbf{X}), \tag{58}$$

where the functional $\mathcal{F}$ is convex. Moreover, its Legendre transform is given by:

$$\mathcal{F}^\star(\mathbf{Y}) = \max_{\mathbf{X} \succ 0} \langle \mathbf{X}, \mathbf{Y} + \left( \begin{smallmatrix} 0 & \mathrm{Id} \\ \mathrm{Id} & 0 \end{smallmatrix} \right) \rangle + \sigma^2 \log \det(\mathbf{X})$$

$$= \left( -\sigma^2 \log \det \right)^\star \left( \mathbf{Y} + \left( \begin{smallmatrix} 0 & \mathrm{Id} \\ \mathrm{Id} & 0 \end{smallmatrix} \right) \right)$$

$$= \sigma^2 (-\log \det)^\star \left( \frac{1}{\sigma^2} \left( \mathbf{Y} + \left( \begin{smallmatrix} 0 & \mathrm{Id} \\ \mathrm{Id} & 0 \end{smallmatrix} \right) \right) \right)$$

$$= -\sigma^2 \log \det \left( -\frac{1}{\sigma^2} \left( \mathbf{Y} + \left( \begin{smallmatrix} 0 & \mathrm{Id} \\ \mathrm{Id} & 0 \end{smallmatrix} \right) \right) \right) - 2\sigma^2 d$$

$$= -\sigma^2 \log \det \left( - \left( \mathbf{Y} + \left( \begin{smallmatrix} 0 & \mathrm{Id} \\ \mathrm{Id} & 0 \end{smallmatrix} \right) \right) \right) - 2d(\sigma^2 - \sigma^2 \log(\sigma^2)).$$

Let $\mathcal{H}$ be the linear operator $\mathcal{H} : \mathbf{X} \mapsto (\mathbf{X}_1, \mathbf{X}_4)$. Its conjugate operator is defined on $\mathcal{S}_{++}^d \times \mathcal{S}_{++}^d$ and is given by $\mathcal{H}^\star(\mathbf{F}, \mathbf{G}) = \left( \begin{smallmatrix} \mathbf{F} & 0 \\ 0 & \mathbf{G} \end{smallmatrix} \right)$. Therefore, Fenchel's duality theorem leads to:

$$\mathfrak{D}(\mathbf{A}, \mathbf{B}) = \max_{\mathbf{F}, \mathbf{G} \succ 0} - \langle \mathbf{F}, \mathbf{A} \rangle - \langle \mathbf{G}, \mathbf{B} \rangle - \mathcal{F}^\star \left( -\mathcal{H}^\star(\mathbf{F}, \mathbf{G}) \right)$$

$$= \max_{\mathbf{F}, \mathbf{G} \succ 0} - \langle \mathbf{F}, \mathbf{A} \rangle - \langle \mathbf{G}, \mathbf{B} \rangle + \sigma^2 \log \det \left( \begin{smallmatrix} \mathbf{F} & -\mathrm{Id} \\ -\mathrm{Id} & \mathbf{G} \end{smallmatrix} \right) + 2d(\sigma^2 - \sigma^2 \log(\sigma^2))$$

$$= \max_{\mathbf{F}, \mathbf{G} \succ 0} - \langle \mathbf{F}, \mathbf{A} \rangle - \langle \mathbf{G}, \mathbf{B} \rangle + \sigma^2 \log \det (\mathbf{F}\mathbf{G} - \mathrm{Id}) + 2d(\sigma^2 - \sigma^2 \log(\sigma^2))$$

Where the last equality follows from the fact that $\mathrm{Id}$ and $\mathbf{G}$ commute. Therefore, reinserting the discarded trace terms, the dual problem of (26) can be written as

$$\max_{\mathbf{F}, \mathbf{G} \succ 0} \Big\{ - \langle \mathbf{F},\ \mathbf{A} \rangle - \langle \mathbf{G},\ \mathbf{B} \rangle + \sigma^2 \log \det (\mathbf{F}\mathbf{G} - \mathrm{Id})$$

$$+ \operatorname{Tr}(\mathbf{A}) + \operatorname{Tr}(\mathbf{B}) + \sigma^2 \log \det \mathbf{A}\mathbf{B} + 2d\sigma^2 (1 - \log \sigma^2)) \Big\}. \tag{59}$$

□

**Proof of Proposition 6**

*Proof.* (i) *Optimality:* Canceling out the gradients in eq. (28) leads to the following optimality conditions:

$$-A + \sigma^2 \mathbf{G}(\mathbf{F}\mathbf{G} - \mathrm{Id})^{-1} = 0$$
$$-B + \sigma^2 (\mathbf{F}\mathbf{G} - \mathrm{Id})^{-1} \mathbf{F} = 0, \tag{60}$$

i.e.

$$\mathbf{F} = \sigma^2 \mathbf{A}^{-1} + \mathbf{G}^{-1}$$
$$\mathbf{G} = \sigma^2 \mathbf{B}^{-1} + \mathbf{F}^{-1} \tag{61}$$

Thus $(\mathbf{F}, \mathbf{G})$ is a solution of the Sinkhorn fixed point equation (20).

(ii) *Differentiabilty:* Using Danskin's theorem on problem (28) leads to the formula of the gradient as a function of the optimal dual pair $(\mathbf{F}, \mathbf{G})$. Indeed, keeping in mind that $\nabla_{\mathbf{A}} \log \det(\mathbf{A}) = -\mathbf{A}^{-1}$ and using the change of variable of Proposition 2, we recover the dual potentials of Corollary 1:

$$\nabla \mathfrak{B}_{\sigma^2}(\mathbf{A}, \mathbf{B}) = \left(\mathrm{Id} - \mathbf{F}^* + \sigma^2 \mathbf{A}^{-1}, \mathrm{Id} - \mathbf{G}^* + \sigma^2 \mathbf{B}^{-1}\right)$$
$$= -\sigma^2(\mathbf{U}, \mathbf{V})$$

Using Corollary 1, it holds that

$$\nabla_{\mathbf{A}} \mathfrak{B}_{\sigma^2}(\mathbf{A}, \mathbf{B}) = -\sigma^2 \mathbf{U}$$
$$= \mathrm{Id} - \mathbf{B}(\mathbf{C} + \sigma^2 \mathrm{Id})^{-1}$$
$$= \mathrm{Id} - \mathbf{B}\left((\mathbf{AB} + \frac{\sigma^4}{4}\mathrm{Id})^{\frac{1}{2}} + \frac{\sigma^2}{2}\mathrm{Id}\right)^{-1}$$
$$= \mathrm{Id} - \mathbf{B}^{\frac{1}{2}}\left((\mathbf{B}^{\frac{1}{2}}\mathbf{AB}^{\frac{1}{2}} + \frac{\sigma^4}{4}\mathrm{Id})^{\frac{1}{2}} + \frac{\sigma^2}{2}\mathrm{Id}\right)^{-1}\mathbf{B}^{\frac{1}{2}}$$
$$= \mathrm{Id} - \mathbf{B}^{\frac{1}{2}}\left(\mathbf{D}^{\frac{1}{2}} + \frac{\sigma^2}{2}\mathrm{Id}\right)^{-1}\mathbf{B}^{\frac{1}{2}},$$

where $\mathbf{D} \stackrel{\text{def}}{=} \mathbf{B}^{\frac{1}{2}}\mathbf{AB}^{\frac{1}{2}} + \frac{\sigma^4}{4}\mathrm{Id}$.

(iii) *Convexity:* Assume without loss of generality that $\mathbf{B}$ is fixed and let $G : \mathbf{B} \mapsto \nabla_{\mathbf{A}} \mathfrak{B}_{\sigma^2}(\mathbf{A}, \mathbf{B})$. As long as $\sigma > 0$, $G$ is differentiable as a composition of differentiable functions. Let's show that the Hessian of $\psi : \mathbf{A} \mapsto \mathfrak{B}_{\sigma^2}(\mathbf{A}, \mathbf{B})$ is a positive quadratic form. Take a direction $\mathbf{H} \in \mathcal{S}^d_+$. It holds:

$$\nabla^2_{\mathbf{A}} \mathfrak{B}_{\sigma^2}(\mathbf{A}, \mathbf{B})(\mathbf{H}, \mathbf{H}) = \langle \mathbf{H}, \mathrm{Jac}_G(\mathbf{A})(\mathbf{H})\rangle$$
$$= \mathrm{Tr}(\mathbf{H} \, \mathrm{Jac}_G(\mathbf{A})(\mathbf{H})).$$

For the sake of clarity, let's write $G(\mathbf{A}) = \mathrm{Id} - L(W(\phi(\mathbf{A})))$ with the following intermediary functions:

$$L : \mathbf{A} \mapsto \mathbf{B}^{\frac{1}{2}}\mathbf{AB}^{\frac{1}{2}}$$

$$Q : \mathbf{A} \mapsto \mathbf{A}^{\frac{1}{2}}$$

$$\phi : \mathbf{A} \mapsto Q(L(\mathbf{A}) + \frac{\sigma^4}{4}\mathrm{Id})$$

$$W : \mathbf{A} \mapsto (\mathbf{A} + \frac{\sigma^2}{2}\mathrm{Id})^{-1}.$$

Moreover, their derivatives are given by:

$$\mathrm{Jac}_L(\mathbf{A})(\mathbf{H}) = \mathbf{B}^{\frac{1}{2}}\mathbf{HB}^{\frac{1}{2}}$$
$$\mathrm{Jac}_W(\mathbf{A})(\mathbf{H}) = -(\mathbf{A} + \frac{\sigma^2}{2}\mathrm{Id})^{-1}\mathbf{H}(\mathbf{A} + \frac{\sigma^2}{2}\mathrm{Id})^{-1}$$
$$\mathrm{Jac}_Q(\mathbf{A})(\mathbf{H}) = \mathbf{Z},$$

where $\mathbf{Z} \in \mathcal{S}^d_+$ is the unique solution of the Sylvester equation: $\mathbf{ZA}^{\frac{1}{2}} + \mathbf{A}^{\frac{1}{2}}\mathbf{Z} = \mathbf{H}$.

Using the chain rule:

$$\mathrm{Jac}_G(\mathbf{A})(\mathbf{H}) = -\mathrm{Jac}_L(W(\phi(\mathbf{A})))(\mathrm{Jac}_W(\phi(\mathbf{A}))(\mathrm{Jac}_\phi(\mathbf{A})(\mathbf{H})))$$
$$= -\mathbf{B}^{\frac{1}{2}}\mathrm{Jac}_W(\phi(\mathbf{A}))(\mathrm{Jac}_\phi(\mathbf{A})(\mathbf{H}))\mathbf{B}^{\frac{1}{2}}$$
$$= \mathbf{B}^{\frac{1}{2}}\left(\phi(\mathbf{A}) + \frac{\sigma^2}{2}\mathrm{Id}\right)^{-1}\mathrm{Jac}_\phi(\mathbf{A})(\mathbf{H})\left(\phi(\mathbf{A}) + \frac{\sigma^2}{2}\mathrm{Id}\right)^{-1}\mathbf{B}^{\frac{1}{2}}$$
$$= \mathbf{B}^{\frac{1}{2}}\left(\mathbf{D}^{\frac{1}{2}} + \frac{\sigma^2}{2}\mathrm{Id}\right)^{-1}\mathrm{Jac}_\phi(\mathbf{A})(\mathbf{H})\left(\mathbf{D}^{\frac{1}{2}} + \frac{\sigma^2}{2}\mathrm{Id}\right)^{-1}\mathbf{B}^{\frac{1}{2}}.$$

Again using the chain rule:

$$\mathbf{Y} \overset{\text{def}}{=} \operatorname{Jac}_\phi(\mathbf{A})(\mathbf{H}) = \operatorname{Jac}_Q\!\left(L(\mathbf{A}) + \frac{\sigma^4}{4}\operatorname{Id}\right)((\operatorname{Jac}_L(\mathbf{A}))(\mathbf{H}))$$

$$= \operatorname{Jac}_Q\!\left(L(\mathbf{A}) + \frac{\sigma^4}{4}\operatorname{Id}\right)(\mathbf{B}^{\frac{1}{2}}\mathbf{H}\mathbf{B}^{\frac{1}{2}})$$

$$= \operatorname{Jac}_Q(\mathbf{D})(\mathbf{B}^{\frac{1}{2}}\mathbf{H}\mathbf{B}^{\frac{1}{2}}).$$

Therefore, $\mathbf{Y} \succ 0$ is the unique solution of the Sylvester equation:

$$\mathbf{Y}\mathbf{D}^{\frac{1}{2}} + \mathbf{D}^{\frac{1}{2}}\mathbf{Y} = \mathbf{B}^{\frac{1}{2}}\mathbf{H}\mathbf{B}^{\frac{1}{2}}.$$

Combining everything:

$$\nabla^2_{\mathbf{A}}\mathfrak{B}_{\sigma^2}(\mathbf{A},\mathbf{B})(\mathbf{H},\mathbf{H}) = \langle \mathbf{H}, \operatorname{Jac}_G(\mathbf{A})(\mathbf{H}) \rangle$$

$$= \operatorname{Tr}\left(\mathbf{H}\operatorname{Jac}_G(\mathbf{A})(\mathbf{H})\right)$$

$$= \operatorname{Tr}\left(\mathbf{H}\mathbf{B}^{\frac{1}{2}}\left(\mathbf{D}^{\frac{1}{2}} + \frac{\sigma^2}{2}\operatorname{Id}\right)^{-1}\mathbf{Y}\left(\mathbf{D}^{\frac{1}{2}} + \frac{\sigma^2}{2}\operatorname{Id}\right)^{-1}\mathbf{B}^{\frac{1}{2}}\right)$$

$$= \operatorname{Tr}\left(\mathbf{B}^{\frac{1}{2}}\mathbf{H}\mathbf{B}^{\frac{1}{2}}\left(\mathbf{D}^{\frac{1}{2}} + \frac{\sigma^2}{2}\operatorname{Id}\right)^{-1}\mathbf{Y}\left(\mathbf{D}^{\frac{1}{2}} + \frac{\sigma^2}{2}\operatorname{Id}\right)^{-1}\right).$$

Since $\mathbf{H}$ and $\mathbf{Y}$ are positive, the matrices $\mathbf{B}^{\frac{1}{2}}\mathbf{H}\mathbf{B}^{\frac{1}{2}}$ and $\left(\mathbf{D}^{\frac{1}{2}} + \frac{\sigma^2}{2}\operatorname{Id}\right)^{-1}\mathbf{Y}\left(\mathbf{D}^{\frac{1}{2}} + \frac{\sigma^2}{2}\operatorname{Id}\right)^{-1}$ are positive semi-definite as well. Their product is similar to a positive semi-definite matrix, therefore the trace above is non-negative.

Given that $\mathbf{A}$ and $\mathbf{H}$ are arbitrary positive semi-definite matrices, it holds that

$$\nabla^2_{\mathbf{A}}\mathfrak{B}_{\sigma^2}(\mathbf{A},\mathbf{B})(\mathbf{H},\mathbf{H}) \geq 0$$

Therefore, $\mathbf{A} \mapsto \mathfrak{B}_{\sigma^2}(\mathbf{A},\mathbf{B})$ is convex.

*Counter-example of joint convexity:* If $\mathfrak{B}_{\sigma^2}$ were jointly convex , then $\delta \overset{\text{def}}{=} : \mathbf{A} \to \mathfrak{B}_{\sigma^2}(\mathbf{A},\mathbf{A})$ would be a convex function.

In the 1-dimensional case with $\sigma = 1$, one can see that this would be equivalent to $x \to \ln((x^2 + 1)^{\frac{1}{2}} + 1) - (x^2 + 1)^{\frac{1}{2}}$ being convex, whereas it is in fact strictly concave.

*(iv) Minimizer of $\phi_{\mathbf{B}}$* With fixed $\mathbf{B}$, cancelling the gradient of $\phi_{\mathbf{B}} \overset{\text{def}}{=} : \mathbf{A} \mapsto \mathfrak{B}_{\sigma^2}(\mathbf{A},\mathbf{B})$ leads to $\mathbf{A} = \mathbf{B} - \sigma^2 \operatorname{Id}$ which is well defined if and only if $\mathbf{B} \succeq \sigma^2 \operatorname{Id}$. However, if $\mathbf{B} - \sigma^2 \operatorname{Id}$ is not positive semi-definite, write the eigenvalue decomposition: $\mathbf{B} = \mathbf{P}\Sigma\mathbf{P}^\top$ and define $\mathbf{A}_0 \overset{\text{def}}{=} \mathbf{P}(\Sigma - \sigma^2 \operatorname{Id})_+\mathbf{P}^\top$ where the operator $x_+ = \max(x,0)$ is applied element-wise. Then:

$$\nabla_{\mathbf{A}}\phi_{\mathbf{B}}(\mathbf{A}_0) = \operatorname{Id} - \mathbf{P}\Sigma^{\frac{1}{2}}\mathbf{P}^\top\left((\mathbf{P}(\Sigma^2 - \sigma^2\Sigma)_+\mathbf{P}^\top + \frac{\sigma^4}{4}\operatorname{Id})^{\frac{1}{2}} + \frac{\sigma^2}{2}\operatorname{Id}\right)^{-1}\mathbf{P}\Sigma^{\frac{1}{2}}\mathbf{P}^\top$$

$$= \operatorname{Id} - \mathbf{P}\Sigma^{\frac{1}{2}}\left(((\Sigma^2 - \sigma^2\Sigma)_+ + \frac{\sigma^4}{4}\operatorname{Id})^{\frac{1}{2}} + \frac{\sigma^2}{2}\operatorname{Id}\right)^{-1}\Sigma^{\frac{1}{2}}\mathbf{P}^\top$$

$$= \operatorname{Id} - \mathbf{P}\Sigma^{\frac{1}{2}}\left((\Sigma - \sigma^2 \operatorname{Id})_+ + \sigma^2 \operatorname{Id}\right)^{-1}\Sigma^{\frac{1}{2}}\mathbf{P}^\top$$

$$= \mathbf{P}(\operatorname{Id} - \Sigma^{\frac{1}{2}}\left((\Sigma - \sigma^2 \operatorname{Id})_+ + \sigma^2 \operatorname{Id}\right)^{-1}\Sigma^{\frac{1}{2}})\mathbf{P}^\top$$

$$= \frac{1}{\sigma^2}\mathbf{P}(\sigma^2 \operatorname{Id} - \Sigma)_+\mathbf{P}^\top$$

Thus, given that $(\Sigma - \sigma^2 \operatorname{Id})_+(\sigma^2 \operatorname{Id} - \Sigma)_+ = 0$, it holds, for any $\mathbf{H} \in \mathcal{S}_+^d$:

$$\langle \mathbf{H} - \mathbf{A}_0, \nabla_{\mathbf{A}}\phi_{\mathbf{B}}(\mathbf{A}_0)\rangle = \langle \mathbf{P}^\top \mathbf{H}\mathbf{P} - (\Sigma - \sigma^2 \operatorname{Id})_+, (\sigma^2 \operatorname{Id} - \Sigma)_+\rangle$$

$$= \langle \mathbf{P}^\top \mathbf{H}\mathbf{P}, (\sigma^2 \operatorname{Id} - \Sigma)_+\rangle$$

$$= \operatorname{Tr}(\mathbf{P}^\top \mathbf{H}\mathbf{P}(\sigma^2 \operatorname{Id} - \Sigma)_+) \geq 0$$

Where the last inequality holds since both matrices are positive semi-definite. Given that $\phi_{\mathbf{B}}$ is convex, the first order optimality condition holds so $\phi_{\mathbf{B}}$ is minimized at $\mathbf{A}_0$. $\qquad\square$

**Proof of Theorem 2**

*Proof.* This theorem is a generalization of [31, Thm 3] for multivariate Gaussians. First we are going to break it down using the centering lemma 1. For any probability measure $\mu$, let $\bar{\mu}$ denote its centered transformation. The debiased barycenter problem is equivalent to:

$$
\min_{\beta \in \mathcal{G}} \sum_{k=1}^{K} w_k S_\sigma(\alpha_k, \beta)
$$

$$
= \min_{\beta \in \mathcal{G}} \sum_{k=1}^{K} w_k \operatorname{OT}_\sigma(\alpha_k, \beta) - \frac{1}{2}(\operatorname{OT}_\sigma(\alpha_k, \alpha_k) + \operatorname{OT}_\sigma(\beta, \beta))
$$

$$
= \min_{\beta \in \mathcal{G}} \sum_{k=1}^{K} w_k \|\mathbf{a}_k - \mathbb{E}_\beta(X)\|^2 + w_k \operatorname{OT}_\sigma(\bar{\alpha}_k, \bar{\beta}) - \frac{1}{2}(w_k \operatorname{OT}_\sigma(\bar{\alpha}_k, \bar{\alpha}_k) + \operatorname{OT}_\sigma(\bar{\beta}, \bar{\beta}))
$$

$$
= \min_{\substack{\mathbf{b} \in \mathbb{R}^d \\ \beta \in \mathcal{G}, \mathbb{E}_\beta(\mathbf{X})=0}} \sum_{k=1}^{K} w_k \|\mathbf{a}_k - \mathbf{b}\|^2 + w_k \operatorname{OT}_\sigma(\bar{\alpha}_k, \beta) - \frac{1}{2}(w_k \operatorname{OT}_\sigma(\bar{\alpha}_k, \bar{\alpha}_k) + \operatorname{OT}_\sigma(\beta, \beta))
$$

(62)

Therefore, since both arguments are independent, we can first minimize over $\mathbf{b}$ to obtain $\mathbb{E}_\beta(\mathbf{X}) = \mathbf{b} = \sum_{k=1}^{K} w_k \mathbf{a}_k$. Without loss of generality, we assume from now on that $\mathbf{a}_k = 0$ for all $k$.

The rest of this proof is adapted from [31], Thm 3 to $d \geq 1$. Janati et al. [31] showed that $S_\sigma$ is differentiable and convex (w.r.t. one measure at a time) on sub-Gaussian measures where the notion of differentiability is different from the usual Fréchet differentiability: a function $F : \mathcal{G} \to \mathbb{R}$ is differentiable at $\alpha$ if there exists $\nabla F(\alpha) \in \mathcal{C}(\mathbb{R}^d)$ such that for any displacement $t\delta\alpha$ with $t > 0$ and $\delta\alpha = \alpha_1 - \alpha_2$ with $\alpha_1, \alpha_2 \in \mathcal{G}$, and

$$
F(\alpha + t\delta\alpha) = F(\alpha) + t\langle \delta\alpha, \nabla F(\alpha) \rangle + o(t) \ ,
$$

(63)

where $\langle \delta\alpha, \nabla F(\alpha) \rangle = \int_{\mathbb{R}^d} \nabla F(\alpha) \mathrm{d}\delta\alpha$.

Moreover, $F$ is convex if and only if for any $\alpha, \alpha' \in \mathcal{G}$:

$$
F(\alpha) \geq F(\alpha') + \langle \alpha - \alpha', \nabla F(\alpha') \rangle \ ,
$$

(64)

Let $(f_k, g_k)$ denote the potentials associated with $\operatorname{OT}_\sigma(\alpha_k, \beta)$ and $h_\beta$ the autocorrelation potential associated with $\operatorname{OT}_\sigma(\beta, \beta)$. If $\beta$ is sub-Gaussian, it holds: $\nabla_\beta S_\sigma(\alpha_k, \beta) = g_k - h$. Therefore, from (64) a probability measure $\beta$ is the debiased barycenter if and only if for any direction $\mu \in \mathcal{G}$, the optimality condition holds:

$$
\langle \sum_{k=1}^{K} w_k \nabla_\beta S_\sigma(\alpha_k, \beta), \mu - \beta \rangle \geq 0
$$

$$
\Leftrightarrow \sum_{k=1}^{K} w_k \langle g_k - h_\beta, \mu - \beta \rangle \geq 0
$$

(65)

Moreover, the potentials $(f_k), (g_k)$ and $h$ must verify the Sinkhorn optimality conditions (10) for all $k$ and for all x $\beta$-a.s and y $\alpha$-a.s:

$$
\begin{cases}
e^{\frac{f_k(x)}{2\sigma^2}} \left( \int_{\mathbb{R}^d} e^{\frac{-\|x-y\|^2 + g_k(y)}{2\sigma^2}} \mathrm{d}\beta(y) \right) = 1, \quad e^{\frac{g_k(x)}{2\sigma^2}} \left( \int_{\mathbb{R}^d} e^{\frac{-\|x-y\|^2 + f_k(y)}{2\sigma^2}} \mathrm{d}\alpha_k(y) \right) = 1. \\
e^{\frac{h(x)}{2\sigma^2}} \left( \int_{\mathbb{R}^d} e^{\frac{-\|x-y\|^2 + h_\beta(y)}{2\sigma^2}} \mathrm{d}\beta(y) \right) = 1.
\end{cases}
$$

(66)

We are going to show that for the Gaussian measure $\beta$ given in the statement of the theorem is well-defined and verifies all optimality conditions (66). Indeed, assume that $\beta$ is a Gaussian measure given by $\mathcal{N}(\mathbf{B})$ for some unknown $\mathbf{B} \in S_+^d$ (remember that $\beta$ is necessarily centered, following the developments (62)). The Sinkhorn equations can therefore be written as a system on positive definite matrices:

$$
\mathbf{F}_k = \sigma^2 \mathbf{A}_k^{-1} + \mathbf{G}_k^{-1}, \quad \mathbf{G}_k = \sigma^2 \mathbf{B} + \mathbf{F}_k^{-1}, \quad \mathbf{H} = \sigma^2 \mathbf{B} + \mathbf{H}^{-1}
$$

where for all $k$:

$$\begin{aligned}
\frac{f_k}{2\sigma^2} &= \mathcal{Q}(\frac{1}{\sigma^2}(\mathbf{G}_k^{-1} - \mathrm{Id})) + \frac{f_k(0)}{2\sigma^2} \\
\frac{g_k}{2\sigma^2} &= \mathcal{Q}(\frac{1}{\sigma^2}(\mathbf{F}_k^{-1} - \mathrm{Id})) + \frac{g_k(0)}{2\sigma^2} \\
\frac{h_\beta}{2\sigma^2} &= \mathcal{Q}(\frac{1}{\sigma^2}(\mathbf{H}^{-1} - \mathrm{Id})) + \frac{h_\beta(0)}{2\sigma^2}
\end{aligned} \tag{67}$$

Moreover, provided $\mathbf{B}$ exists and is positive definite, the system (67) has a unique set of solutions $(\mathbf{F}_k)_k, (\mathbf{G}_k)_k, \mathbf{H}$ given by:

$$\mathbf{F}_k = \mathbf{B}\mathbf{C}_k^{-1}, \quad \mathbf{G}_k = \mathbf{C}_k^{-1}\mathbf{A}_k, \quad \mathbf{H} = \mathbf{B}^{-1}\mathbf{J} \tag{68}$$

where $\mathbf{C}_k = (\mathbf{A}_k\mathbf{B} + \frac{\sigma^4}{4}\,\mathrm{Id})^{\frac{1}{2}} - \frac{\sigma^2}{2}\,\mathrm{Id}$ and $\mathbf{J} = (\mathbf{B}^2 + \frac{\sigma^4}{4}\,\mathrm{Id})^{\frac{1}{2}} + \frac{\sigma^2}{2}\,\mathrm{Id}$. Therefore, the gradient in (65) can be written:

$$\begin{aligned}
\sum_{k=1}^{K} w_k(g_k - h_\beta) &= \mathcal{Q}(2(\sum_{k=1}^{K} w_k\mathbf{F}_k^{-1} - \mathbf{H}^{-1})) + \sum_{w=1}^{K} w_k g_k(0) - h_\beta(0) \\
&= \mathcal{Q}(2(\sum_{k=1}^{K} w_k\mathbf{F}_k^{-1} - \mathbf{H}^{-1})) + m \ ,
\end{aligned} \tag{69}$$

for some constant $m \in \mathbb{R}$. Let's compute the matrix defining the quadratic form:

$$\begin{aligned}
&\sum_{k=1}^{K} w_k\mathbf{C}_k\mathbf{B}^{-1} - \mathbf{J}^{-1}\mathbf{B} \\
&= \sum_{k=1}^{K} w_k\mathbf{B}^{-\frac{1}{2}}(\mathbf{B}^{\frac{1}{2}}\mathbf{A}_k\mathbf{B}^{\frac{1}{2}} + \frac{\sigma^4}{4}\,\mathrm{Id})^{\frac{1}{2}}\mathbf{B}^{-\frac{1}{2}} - \mathbf{B}^{-1}(\mathbf{B}^2 + \frac{\sigma^4}{4}\,\mathrm{Id})^{\frac{1}{2}} \\
&= \sum_{k=1}^{K} w_k\mathbf{B}^{-\frac{1}{2}}(\mathbf{B}^{\frac{1}{2}}\mathbf{A}_k\mathbf{B}^{\frac{1}{2}} + \frac{\sigma^4}{4}\,\mathrm{Id})^{\frac{1}{2}}\mathbf{B}^{-\frac{1}{2}} - \mathbf{B}^{-\frac{1}{2}}(\mathbf{B}^2 + \frac{\sigma^4}{4}\,\mathrm{Id})^{\frac{1}{2}}\mathbf{B}^{-\frac{1}{2}} \\
&= \mathbf{B}^{-\frac{1}{2}}\left(\sum_{k=1}^{K} w_k(\mathbf{B}^{\frac{1}{2}}\mathbf{A}_k\mathbf{B}^{\frac{1}{2}} + \frac{\sigma^4}{4}\,\mathrm{Id})^{\frac{1}{2}} - (\mathbf{B}^2 + \frac{\sigma^4}{4}\,\mathrm{Id})^{\frac{1}{2}}\right)\mathbf{B}^{-\frac{1}{2}}
\end{aligned} \tag{70}$$

which is null if $\mathbf{B}$ is a solution of the equation:

$$\sum_{k=1}^{K} w_k(\mathbf{B}^{\frac{1}{2}}\mathbf{A}_k\mathbf{B}^{\frac{1}{2}} + \frac{\sigma^4}{4}\,\mathrm{Id})^{\frac{1}{2}} = (\mathbf{B}^2 + \frac{\sigma^4}{4}\,\mathrm{Id})^{\frac{1}{2}}. \tag{71}$$

Therefore, the gradient is constant and equal to $m$. For any probability measure $\mu \in \mathcal{G}$:

$$\begin{aligned}
\langle \sum_{k=1}^{K} w_k\nabla_\beta S_{2\sigma^2}(\alpha_k, \beta), \mu - \beta\rangle &= \langle \sum_{k=1}^{K} w_k g_k - h_\beta, \mu - \beta\rangle \\
&= \langle m, \mu - \beta\rangle \\
&= 0
\end{aligned} \tag{72}$$

since both measures integrate to 1. Therefore, the optimality condition holds.

To end the proof, all we need to show is that (71) admits a positive definite solution. To show the existence of a solution, the same proof of Agueh and Carlier [1] applies. Indeed, let $\lambda_k$ and $\Lambda_k$ denote respectively the smallest and largest eigenvalue of $\mathbf{A}_k$. Let $\lambda = \min_k \lambda_k$ and $\Lambda = \max_k \Lambda_k$. Let $K_{\lambda,\Lambda}$ be the convex compact subset of positive definite matrices $\mathbf{B}$ such that $\Lambda\,\mathrm{Id} \succeq \mathbf{B} \succeq \lambda\,\mathrm{Id}$. Define the map:

$$T : K_{\lambda,\Lambda} \to \mathcal{S}_{++}^d$$

$$\mathbf{B} \mapsto \left(\left(\sum_{k=1}^{K} w_k(\mathbf{B}^{\frac{1}{2}}\mathbf{A}_k\mathbf{B}^{\frac{1}{2}} + \frac{\sigma^4}{4}\,\mathrm{Id})^{\frac{1}{2}}\right)^2 - \frac{\sigma^4}{4}\,\mathrm{Id}\right)^{\frac{1}{2}}$$

Now for any $\mathbf{B} \in K_{\lambda, \Lambda}$, it holds:

$$\lambda \operatorname{Id} \preceq T(\mathbf{B}) \preceq \Lambda \operatorname{Id}. \tag{73}$$

$T$ is therefore a continuous function that maps $K_{\lambda, \Lambda}$ to itself, thus Brouwer's fixed-point theorem guarantees the existence of a solution. $\square$

**Proof of Proposition 7**

*Proof.* Using Fubini-Tonelli along with the optimality conditions (35), the double integral can be written:

$$\pi(\mathbb{R}^d \times \mathbb{R}^d) = \int_{\mathbb{R}^d \times \mathbb{R}^d} e^{\frac{-\|x-y\|^2 + f(x) + g(y)}{2\sigma^2}} \, \mathrm{d}\alpha(x) \mathrm{d}\beta(y)$$

$$= \int_{\mathbb{R}^d} \left( \int_{\mathbb{R}^d} e^{\frac{-\|x-y\|^2 + f(x)}{2\sigma^2}} \, \mathrm{d}\alpha(x) \right) e^{\frac{g(y)}{2\sigma^2}} \, \mathrm{d}\beta(y)$$

$$= \int_{\mathbb{R}^d} e^{\frac{g(y)}{2\sigma^2}(1-\frac{1}{\tau})} \, \mathrm{d}\beta(y)$$

$$= \int_{\mathbb{R}^d} e^{-\frac{g(y)}{\gamma}} \, \mathrm{d}\beta(y)$$

And similarly: $\pi(\mathbb{R}^d \times \mathbb{R}^d) = \int_{\mathbb{R}^d} e^{-\frac{f(x)}{\gamma}} \, \mathrm{d}\alpha(x)$. Therefore, the three integrals in the dual objective (34) are equal to $\pi(\mathbb{R}^d \times \mathbb{R}^d)$ which ends the proof. $\square$

**Lemma 3.** *[Sum of factorized quadratic forms] Let* $\mathbf{A}, \mathbf{B} \in S_d$ *such that* $\mathbf{A} \neq \mathbf{B}$ *and* $\mathbf{a}, \mathbf{b} \in \mathbb{R}^d$. *Denote* $\alpha = (\mathbf{A}, \mathbf{a})$ *and* $\beta = (\mathbf{B}, \mathbf{b})$. *Let* $P_\alpha(\mathbf{x}) = -\frac{1}{2}(\mathbf{x} - \mathbf{a})^\top \mathbf{A}(\mathbf{x} - \mathbf{a})$ *and* $P_\beta(\mathbf{x}) = -\frac{1}{2}(\mathbf{x} - \mathbf{b})^\top \mathbf{B}(\mathbf{x} - \mathbf{b})$. *Then:*

$$P_\alpha(x) + P_\beta(x) = -\frac{1}{2} \left( (\mathbf{x} - \mathbf{c})^\top \mathbf{C}(\mathbf{x} - \mathbf{c}) + q_{\alpha,\beta} \right) \tag{74}$$

*where:*

$$\begin{cases} \mathbf{C} & = \mathbf{A} + \mathbf{B} \\ (\mathbf{A} + \mathbf{B})\mathbf{c} & = (\mathbf{A}\mathbf{a} + \mathbf{B}\mathbf{b}) \\ q_{\alpha,\beta} & = \mathbf{a}^\top \mathbf{A}\mathbf{a} + \mathbf{b}^\top \mathbf{B}\mathbf{b} - c^\top \mathbf{C}\mathbf{c} \end{cases} \tag{75}$$

*In particular, if* $\mathbf{C} = \mathbf{A} + \mathbf{B}$ *is invertible, then:*

$$\begin{cases} \mathbf{c} = \mathbf{C}^{-1}(\mathbf{A}\mathbf{a} + \mathbf{B}\mathbf{b}) \\ \mathbf{c}^\top \mathbf{C}\mathbf{c} = (\mathbf{A}\mathbf{a} + \mathbf{B}\mathbf{b})^\top \mathbf{C}^{-1}(\mathbf{A}\mathbf{a} + \mathbf{B}\mathbf{b}) \end{cases} \tag{76}$$

*Proof.* On one hand,

$$P_\alpha(x) + P_\beta(x) = -\frac{1}{2} \left( (\mathbf{x} - \mathbf{a})^\top \mathbf{A}(\mathbf{x} - \mathbf{a}) + (\mathbf{x} - \mathbf{b})^\top \mathbf{B}(\mathbf{x} - \mathbf{b}) \right)$$

$$= -\frac{1}{2} \left( \mathbf{x}^\top (\mathbf{A} + \mathbf{B})\mathbf{x} - 2\mathbf{x}^\top (\mathbf{A}\mathbf{a} + \mathbf{B}\mathbf{b}) + \mathbf{a}^\top \mathbf{A}\mathbf{a} + \mathbf{b}^\top \mathbf{B}\mathbf{b} \right)$$

On the other hand, for an arbitrary $\gamma = (\mathbf{c}, \mathbf{C})$ and $q \in \mathbb{R}$:

$$P_\gamma(x) - \frac{q}{2} = -\frac{1}{2} \left( (\mathbf{x} - \mathbf{c})^\top \mathbf{C}(\mathbf{x} - \mathbf{C}) + q \right)$$

$$= -\frac{1}{2} \left( x^\top \mathbf{C}x - 2x^\top \mathbf{C}\mathbf{c} + \mathbf{c}^\top \mathbf{C}\mathbf{c} + q \right)$$

If $\mathbf{A} \neq \mathbf{B}$, identification of the parameters of both quadratic forms leads to (75). $\square$

**Lemma 4.** *[Gaussian convolution of factorized quadratic forms] Let $\mathbf{A} \in S_d$ and $\mathbf{a} \in \mathbb{R}^d$ and $\sigma > 0$ such that $\sigma^2 \mathbf{A} + \mathrm{Id} \succ 0$. Let $Q_\alpha(\mathbf{x}) = -\frac{1}{2}(\mathbf{x} - \mathbf{a})^\top \mathbf{A}(\mathbf{x} - \mathbf{a})$. Then the convolution of $e^{Q_\alpha}$ by the Gaussian kernel $\mathcal{N}(0, \frac{\mathrm{Id}}{\sigma^2})$ is given by:*

$$\mathcal{N}(0, \frac{\mathrm{Id}}{\sigma^2}) \star \exp\left(Q_\alpha\right) \stackrel{\text{def}}{=} \int_{\mathbb{R}^d} \frac{1}{(2\pi\sigma^2)^{\frac{n}{2}}} \exp\left(-\frac{1}{2\sigma^2}\|. - y\|^2 + Q_\alpha(y)\right) \mathrm{d}y = c_\alpha \exp(Q(\mathbf{a}, \mathbf{J}))$$

(77)

*where:*

$$\mathbf{J} = (\sigma^2 \mathbf{A} + \mathrm{Id})^{-1} \mathbf{A}$$

$$c_\alpha = \frac{1}{\sqrt{\det(\sigma^2 \mathbf{A} + \mathrm{Id})}}$$

*Proof.* Using Lemma 3 one can write for any $x \in \mathbb{R}^d$ considered fixed:

$$-\frac{1}{2\sigma^2}\|x - y\|^2 + Q_\alpha(y) = Q(x, \frac{\mathrm{Id}}{\sigma^2})(y) + Q(\mathbf{a}, \mathbf{A})(y)$$

$$= Q(\mathbf{Aa} + \frac{x}{\sigma^2}, \mathbf{A} + \frac{\mathrm{Id}}{\sigma^2})(y) + h(x)$$

with $h(x) = -\frac{1}{2}\left(\mathbf{a}^\top \mathbf{Aa} + \frac{1}{\sigma^2}\|x\|^2 - \frac{1}{\sigma^2}(\sigma^2 \mathbf{Aa} + x)^\top(\sigma^2 \mathbf{A} + \mathrm{Id})^{-1}(\sigma^2 \mathbf{Aa} + x)\right)$. Therefore, the convolution integral is finite if and only if $\mathbf{A} + \frac{\mathrm{Id}}{\sigma^2} \succ 0$ in which case we get the integral of a Gaussian density:

$$\frac{1}{(2\pi\sigma^2)^{\frac{n}{2}}} \int_{\mathbb{R}^d} \exp\left(Q(\mathbf{Aa} + \frac{x}{\sigma^2}, \mathbf{A} + \frac{\mathrm{Id}}{\sigma^2})(y) + h(x)\right) \mathrm{d}(y) = \sqrt{\frac{\det(2\pi(\mathbf{A} + \frac{\mathrm{Id}}{\sigma^2})^{-1})}{(2\pi\sigma^2)^n}} e^{h(x)}$$

$$= \frac{e^{h(x)}}{\sqrt{\det(\sigma^2 \mathbf{A} + \mathrm{Id})}}$$

For the sake of clarity, let's separate the terms of $h$ depending on their order in $x$: $h(x) = -\frac{1}{2}\left(h_2(x) + h_1(x) + h_0\right)$ where:

$$h_2(x) = \frac{1}{\sigma^2}(\|x\|^2 - x^\top(\sigma^2 \mathbf{A} + \mathrm{Id})^{-1}x$$

$$h_1(x) = -2x^\top(\sigma^2 \mathbf{A} + \mathrm{Id})^{-1}\mathbf{Aa}$$

$$h_0 = \mathbf{aAa} - \sigma^2 \mathbf{a}^\top \mathbf{A}(\sigma^2 \mathbf{A} + \mathrm{Id})^{-1}\mathbf{Aa}$$

Finally, we can factorize $h_2$ and $h_0$ using Woodbury's matrix identity which holds even for a singular matrix $\mathbf{A}$:

$$(\sigma^2 \mathbf{A} + \mathrm{Id})^{-1} = \mathrm{Id} - \sigma^2(\sigma^2 \mathbf{A} + \mathrm{Id})^{-1}\mathbf{A} \qquad \text{(Woodbury's identity)}$$

Let $\mathbf{J} = (\sigma^2 \mathbf{A} + \mathrm{Id})^{-1}\mathbf{A}$.

$$h_2(x) = \frac{1}{\sigma^2}(\|x\|^2 - x^\top(\mathrm{Id} - \sigma^2(\sigma^2 \mathbf{A} + \mathrm{Id})^{-1}\mathbf{A})x$$

$$= x^\top(\sigma^2 \mathbf{A} + \mathrm{Id})^{-1}\mathbf{A}x$$

$$= x^\top \mathbf{J}x$$

$$h_1(x) = -2x^\top \mathbf{Ja}$$

$$h_0 = \mathbf{aAa} - \sigma^2 \mathbf{a}^\top \mathbf{A}(\sigma^2 \mathbf{A} + \mathrm{Id})^{-1}\mathbf{Aa}$$

$$= \mathbf{a}^\top \mathbf{A}(\mathrm{Id} - \sigma^2(\sigma^2 \mathbf{A} + \mathrm{Id})^{-1}\mathbf{A})\mathbf{a}$$

$$= \mathbf{a}^\top \mathbf{A}(\sigma^2 \mathbf{A} + \mathrm{Id})^{-1}\mathbf{a}$$

$$= \mathbf{a}^\top(\sigma^2 \mathbf{A} + \mathrm{Id})^{-1}\mathbf{Aa}$$

$$= \mathbf{a}^\top \mathbf{Ja}$$

Therefore, $h(x) = -\frac{1}{2}\left(x^\top \mathbf{J}x - 2x^\top \mathbf{Ja} + \mathbf{a}^\top \mathbf{Ja}\right) = -\frac{1}{2}(x - \mathbf{a})^\top \mathbf{J}(x - \mathbf{a}) = Q(\mathbf{a}, \mathbf{J})(x)$.  □

**Lemma 5.** *[Gaussian convolution of generic quadratic forms] Let $\mathbf{A} \in S_d$ and $\mathbf{a} \in \mathbb{R}^d$ and $\sigma > 0$ such that $\sigma^2 \mathbf{A} + \mathrm{Id} \succ 0$. Let $Q_\alpha(\mathbf{x}) = -\frac{1}{2}(\mathbf{x}^\top \mathbf{A}\mathbf{x} - 2\mathbf{x}^\top \mathbf{a})$. Then the convolution of $e^{Q_\alpha}$ by the Gaussian kernel $\mathcal{N}(0, \frac{\mathrm{Id}}{\sigma^2})$ is given by:*

$$\mathcal{N}(0, \frac{\mathrm{Id}}{\sigma^2}) \star \exp(Q_\alpha) \overset{\text{def}}{=} \int_{\mathbb{R}^d} \frac{1}{(2\pi\sigma^2)^{\frac{n}{2}}} \exp\left(-\frac{1}{2\sigma^2}\|. - y\|^2 + Q_\alpha(y)\right) \mathrm{d}y = c_\alpha \exp(Q(\mathbf{G}\mathbf{a}, \mathbf{G}\mathbf{A}))$$

(78)

*where:*

$$\mathbf{G} = (\sigma^2 \mathbf{A} + \mathrm{Id})^{-1}$$

$$c_\alpha = \frac{e^{\frac{\sigma^2 \mathbf{a}^\top \mathbf{G}\mathbf{a}}{2}}}{\sqrt{\det(\sigma^2 \mathbf{A} + \mathrm{Id})}}$$

*Proof.* Using Lemma 3 one can write for any $x \in \mathbb{R}^d$ considered fixed:

$$-\frac{1}{2\sigma^2}\|x - y\|^2 + Q_\alpha(y) = Q(x, \frac{\mathrm{Id}}{\sigma^2})(y) + Q(\mathbf{a}, \mathbf{A})(y)$$

$$= Q(\mathbf{a} + \frac{x}{\sigma^2}, \mathbf{A} + \frac{\mathrm{Id}}{\sigma^2})(y) - \frac{1}{2\sigma^2}\|x\|^2$$

$$= P(\sigma\mathbf{a} + \frac{x}{\sigma^2}, \mathbf{A} + \frac{\mathrm{Id}}{\sigma^2})(y) + h(x)$$

with $h(x) = -\frac{1}{2}\left(\frac{1}{\sigma^2}\|x\|^2 - \frac{1}{\sigma^2}(\sigma^2\mathbf{a} + x)^\top(\sigma^2\mathbf{A} + \mathrm{Id})^{-1}(\sigma^2\mathbf{a} + x)\right)$. Therefore, the convolution integral is finite if and only if $\mathbf{A} + \frac{\mathrm{Id}}{\sigma^2} \succ 0$ in which case we get the integral of a Gaussian density:

$$\frac{1}{(2\pi\sigma^2)^{\frac{n}{2}}} \int_{\mathbb{R}^d} \exp\left(Qf(\mathbf{a} + \frac{x}{\sigma^2}, \mathbf{A} + \frac{\mathrm{Id}}{\sigma^2})(y) + h(x)\right) \mathrm{d}(y) = \sqrt{\frac{\det(2\pi(\mathbf{A} + \frac{\mathrm{Id}}{\sigma^2})^{-1})}{(2\pi\sigma^2)^n}} e^{h(x)}$$

$$= \frac{e^{h(x)}}{\sqrt{\det(\sigma^2 \mathbf{A} + \mathrm{Id})}}$$

For the sake of clarity, let's separate the terms of $h$ depending on their order in $x$: $h(x) = -\frac{1}{2}(h_2(x) + h_1(x) + h_0)$ where:

$$h_2(x) = \frac{1}{\sigma^2}(\|x\|^2 - x^\top(\sigma^2\mathbf{A} + \mathrm{Id})^{-1}x$$

$$h_1(x) = -2x^\top(\sigma^2\mathbf{A} + \mathrm{Id})^{-1}\mathbf{a}$$

$$h_0 = -\sigma^2\mathbf{a}^\top(\sigma^2\mathbf{A} + \mathrm{Id})^{-1}\mathbf{a}$$

Finally, we can factorize $h_2$ and $h_0$ using Woodbury's matrix identity which holds even for a singular matrix $\mathbf{A}$:

$$(\sigma^2 \mathbf{A} + \mathrm{Id})^{-1} = \mathrm{Id} - \sigma^2(\sigma^2 \mathbf{A} + \mathrm{Id})^{-1}\mathbf{A} \qquad \text{(Woodbury's identity)}$$

Let $\mathbf{G} = (\sigma^2 \mathbf{A} + \mathrm{Id})^{-1}$.

$$h_2(x) = \frac{1}{\sigma^2}(\|x\|^2 - x^\top(\mathrm{Id} - \sigma^2(\sigma^2\mathbf{A} + \mathrm{Id})^{-1}\mathbf{A})x$$

$$= x^\top(\sigma^2\mathbf{A} + \mathrm{Id})^{-1}\mathbf{A}x$$

$$= x^\top \mathbf{G}\mathbf{A}x$$

$$h_1(x) = -2x^\top \mathbf{G}\mathbf{a}$$

$$h_0 = -\sigma^2\mathbf{a}^\top(\sigma^2\mathbf{A} + \mathrm{Id})^{-1}\mathbf{a}$$

$$= -\sigma^2\mathbf{a}^\top \mathbf{G}\mathbf{a}$$

Therefore, $h(x) = -\frac{1}{2}\left(x^\top \mathbf{G}\mathbf{A}x - 2x^\top \mathbf{G}\mathbf{a} - \sigma^2\mathbf{a}^\top \mathbf{G}\mathbf{a}\right) = Q(\mathbf{G}\mathbf{a}, \mathbf{G}\mathbf{A})(x) + \frac{\sigma^2\mathbf{a}^\top \mathbf{G}\mathbf{a}}{2}$. $\qquad \square$

## 5.4 Proof of theorem 3

In the balanced case, we showed that Sinkhorn's transform is stable for quadratic potentials and that the resulting sequence is a contraction. Similarly, the following proposition shows that the unbalanced Sinkhorn transform is stable for quadratic potentials. M

**Proposition 8.** *Let $\alpha$ be an unbalanced Gaussians given by $m_\alpha \mathcal{N}(\mathbf{a}, \mathbf{A})$. Let $\tau = \frac{\gamma}{2\sigma^2 + \gamma}$. Define the unbalanced Sinkhorn transform $T : \mathbb{R}^{\mathbb{R}^d} \to \mathbb{R}^{\mathbb{R}^d}$:*

$$T_\alpha(h)(x) \overset{\text{def}}{=} -\tau \log \int_{\mathbb{R}^d} e^{\frac{-\|x-y\|^2}{2\sigma^2} + h(y)} \mathrm{d}\alpha(y) \tag{79}$$

*Let $\mathbf{U} \in \mathcal{S}_d$, $\mathbf{u} \in \mathbb{R}^d$ and $m_u > 0$. If $h = \log(m_u) + \mathcal{Q}(\mathbf{u}, \mathbf{U})$ i.e $h(x) = \log(m_u) - \frac{1}{2}(x^\top \mathbf{U} x - 2x^\top \mathbf{u})$, then $T_\alpha(h)$ is well defined if and only if $\mathbf{F} \overset{\text{def}}{=} \sigma^2 \mathbf{U} + \sigma^2 \mathbf{A}^{-1} + \mathrm{Id} \succ 0$, in which case $T_\alpha(h) = \mathcal{Q}(\mathbf{v}, \mathbf{V}) + \log(m_v)$ with the identified parameters:*

$$\mathbf{V} = \tau \frac{1}{\sigma^2}(\mathbf{F}^{-1} - \mathrm{Id}) \tag{80}$$

$$\mathbf{v} = -\tau \mathbf{F}^{-1}(\mathbf{A}^{-1}\mathbf{a} + \mathbf{u}) \tag{81}$$

$$m_v = \left( \frac{\sqrt{\det(\mathbf{A})\det(\mathbf{F})}}{m_u m_\alpha e^{\frac{q_{u,\alpha}}{2}} \sigma^{2d}} \right)^\tau \tag{82}$$

*where $q_{u,\alpha} = \frac{\sigma^2}{\tau^2} \mathbf{v}^\top \mathbf{F} \mathbf{v} - \mathbf{a}^\top \mathbf{A}^{-1} \mathbf{a}$.*

*Proof.* The exponent inside the integral can be written as:

$$e^{\frac{-\|x-y\|^2}{2\sigma^2} + h(y)} \mathrm{d}\alpha(y) \propto e^{\frac{-\|x-y\|^2}{2\sigma^2} - \frac{1}{2}(y^\top \mathbf{X} y - y^\top \mathbf{A}^{-1} y)} \mathrm{d}y$$

$$\propto e^{-\frac{1}{2}(y^\top(\frac{\mathrm{Id}}{\sigma^2} + \mathbf{X} + \mathbf{A}^{-1}) y) + \frac{x^\top y}{\sigma^2}} \mathrm{d}y$$

which is integrable if and only if $\mathbf{U} + \mathbf{A}^{-1} + \frac{1}{\sigma^2} \mathrm{Id} \succ 0 \Leftrightarrow \mathbf{F} \succ 0$. Moreover, up to a multiplicative factor, the exponentiated Sinkhorn transform is equivalent to a Gaussian convolution of an exponentiated quadratic form. Lemma 5 applies:

$$e^{-T_\alpha(h)} = \int_{\mathbb{R}^d} e^{\frac{-\|x-y\|^2}{2\sigma^2} + f(y)} \mathrm{d}\alpha(y)$$

$$= m_u m_\alpha \frac{\exp(-\frac{1}{2}\mathbf{a}^\top \mathbf{A}^{-1}\mathbf{a})}{\sqrt{\det(2\pi\mathbf{A})}} \int_{\mathbb{R}^d} e^{\frac{-\|x-y\|^2}{2\sigma^2} + \mathcal{Q}(\mathbf{u},\mathbf{U})(y) + \mathcal{Q}(\mathbf{A}^{-1}\mathbf{a},\mathbf{A}^{-1})(y)} \mathrm{d}y$$

$$= m_u m_\alpha \frac{\exp(-\frac{1}{2}\mathbf{a}^\top \mathbf{A}^{-1}\mathbf{a})}{\sqrt{\det(2\pi\mathbf{A})}} \sqrt{(2\pi\sigma^2)^{2d}} \exp\left(\mathcal{N}(\sigma^2 \mathrm{Id})\right) \star \exp\left(\mathcal{Q}(\mathbf{u} + \mathbf{A}^{-1}\mathbf{a}, \mathbf{U} + \mathbf{A}^{-1})\right)$$

$$= m_u m_\alpha \frac{\sigma^{2d}\exp(-\frac{1}{2}\mathbf{a}^\top \mathbf{A}^{-1}\mathbf{a})}{\sqrt{\det(\mathbf{A})}} \exp\left(\mathcal{N}(\sigma^2 \mathrm{Id})\right) \star \exp\left(\mathcal{Q}(\mathbf{u} + \mathbf{A}^{-1}\mathbf{a}, \mathbf{U} + \mathbf{A}^{-1})\right)$$

$$= m_u m_\alpha \frac{\sigma^{2d}\exp(-\frac{1}{2}\mathbf{a}^\top \mathbf{A}^{-1}\mathbf{a})}{\sqrt{\det(\mathbf{A})}} c_\alpha \exp\left(\mathcal{Q}(\mathbf{F}^{-1}(\mathbf{u} + \mathbf{A}^1\mathbf{a}), \mathbf{F}^{-1}(\mathbf{U} + \mathbf{A}^{-1}))\right).$$

$$= m_u m_\alpha \frac{\sigma^{2d}\exp(-\frac{1}{2}\mathbf{a}^\top \mathbf{A}^{-1}\mathbf{a})}{\sqrt{\det(\mathbf{A})}} c_\alpha \exp\left(\mathcal{Q}(\mathbf{F}^{-1}(\mathbf{u} + \mathbf{A}^1\mathbf{a}), \frac{1}{\sigma^2}\mathbf{F}^{-1}(\mathbf{F} - \mathrm{Id}))\right).$$

$$= m_u m_\alpha \frac{\sigma^{2d}\exp(-\frac{1}{2}\mathbf{a}^\top \mathbf{A}^{-1}\mathbf{a})}{\sqrt{\det(\mathbf{A})}} c_\alpha \exp\left(\mathcal{Q}(\mathbf{F}^{-1}(\mathbf{u} + \mathbf{A}^1\mathbf{a}), \frac{1}{\sigma^2}(\mathrm{Id} - \mathbf{F}^{-1}))\right).$$

where $c_\alpha = \frac{\exp(\frac{1}{2}\sigma^2(\mathbf{u}+\mathbf{A}^{-1}\mathbf{a})^\top \mathbf{F}^{-1}(\mathbf{u}+\mathbf{A}^{-1}\mathbf{a}))}{\sqrt{\det(\mathbf{F})}}$.

Therefore, by applying $-\tau \log$ we can identify $\mathbf{V}$ and $\mathbf{v}$. Substituting $\mathbf{u} + \mathbf{A}^{-1}\mathbf{a}$ by $-\frac{1}{\tau}\mathbf{F}\mathbf{v}$ leads to the equation of $m_v$. $\qquad\square$

Unlike the balanced case, the unbalanced Sinkhorn iterations require 2 more parameters ($\mathbf{v}$ and $m_v$) with tangled updates. Proving the convergence of the resulting algorithm is more challenging. Instead, we directly solve the optimality conditions and show that a pair of quadratic potentials verifies (35).

**Proposition 9.** *The pair of quadratic forms $(f, g)$ of (38) verifies the optimality conditions (35) if and only if:*

$$\mathbf{F} \overset{\text{def}}{=} \sigma^2 \mathbf{A}^{-1} + \sigma^2 \mathbf{U} + \mathrm{Id} \succ 0$$
$$\mathbf{G} \overset{\text{def}}{=} \sigma^2 \mathbf{B}^{-1} + \sigma^2 \mathbf{V} + \mathrm{Id} \succ 0,$$
(83)

$$m_v \left( \frac{m_u m_\alpha e^{\frac{q_{u,\alpha}}{2}} \sigma^d}{\sqrt{\det(\mathbf{A})\det(\mathbf{F})}} \right)^\tau = 1 \qquad\qquad m_u \left( \frac{m_v m_\beta e^{\frac{q_{v,\beta}}{2}} \sigma^d}{\sqrt{\det(\mathbf{B})\det(\mathbf{G})}} \right)^\tau = 1$$

$$\mathbf{v} = -\tau \mathbf{F}^{-1}(\mathbf{A}^{-1}\mathbf{a} + \mathbf{u}) \qquad\qquad \mathbf{u} = -\tau \mathbf{G}^{-1}(\mathbf{B}^{-1}\mathbf{b} + \mathbf{v})$$
(84)

$$\mathbf{G} = \tau \mathbf{F}^{-1} + \sigma^2 \mathbf{B}^{-1} + (1-\tau)\,\mathrm{Id} \qquad \mathbf{F} = \tau \mathbf{G}^{-1} + \sigma^2 \mathbf{A}^{-1} + (1-\tau)\,\mathrm{Id}$$

$$q_{u,\alpha} = \frac{\sigma^2}{\tau^2}\mathbf{v}^\top \mathbf{F}\mathbf{v} - \mathbf{a}^\top \mathbf{A}^{-1}\mathbf{a} \qquad\qquad q_{v,\beta} = \frac{\sigma^2}{\tau^2}\mathbf{u}^\top \mathbf{G}\mathbf{u} - \mathbf{b}^\top \mathbf{B}^{-1}\mathbf{b}$$

*Proof.* The equations on $m_u, m_v, \mathbf{u}, \mathbf{v}$ follow immediately from Proposition 8. Using the definition of $\mathbf{F}$ and $\mathbf{G}$, substituting $\mathbf{U}$ and $\mathbf{F}$ leads to the equations in $\mathbf{F}$ and $\mathbf{G}$ $\qquad\qquad\square$

We now turn to solve the system (84). Notice that in general, the dual potentials can only be identified up to a an additive constant. Indeed, if a pair $(f, g)$ is optimal, then $(f + K, g - K)$ is also optimal for any $K \in \mathbb{R}$ (the transportation plan does not change). Thus, at optimality, it is sufficient to obtain the product $m_u m_v$. We start by identifying $(\mathbf{F}, \mathbf{G})$ then $(\mathbf{u}, \mathbf{v})$ and finally $m_u m_v$.

**Identifying $\mathbf{F}$ and $\mathbf{G}$.** The equations in $\mathbf{F}$ and $\mathbf{G}$ can be shown to be equivalent to those of the balanced case up to some change of variables. Let $\lambda \overset{\text{def}}{=} \frac{\sigma^2}{1-\tau} = \sigma^2 + \frac{\gamma}{2}$.

$$\begin{cases} \mathbf{F} &= \tau \mathbf{G}^{-1} + \sigma^2 \mathbf{A}^{-1} + (1-\tau)\,\mathrm{Id} \\ \mathbf{G} &= \tau \mathbf{F}^{-1} + \sigma^2 \mathbf{B}^{-1} + (1-\tau)\,\mathrm{Id} \end{cases}$$

$$\Leftrightarrow \begin{cases} \mathbf{F} &= \left(\frac{\mathbf{G}}{\tau}\right)^{-1} + \frac{\sigma^2}{\tau}\tau(\mathbf{A}^{-1} + \frac{1}{\lambda}\,\mathrm{Id}) \\ \frac{\mathbf{G}}{\tau} &= \mathbf{F}^{-1} + \frac{\sigma^2}{\tau}(\mathbf{B}^{-1} + \frac{1}{\lambda}\,\mathrm{Id}) \end{cases}$$

$$\Leftrightarrow \begin{cases} \mathbf{F} &= \widetilde{\mathbf{G}}^{-1} + \sigma^2 (\frac{\widetilde{\mathbf{A}}}{\tau})^{-1} \\ \widetilde{\mathbf{G}} &= \mathbf{F}^{-1} + \sigma^2 \widetilde{\mathbf{B}}^{-1} \end{cases}$$

which correspond to the balanced OT fixed point equations (20) associated with the pair $(\frac{\widetilde{\mathbf{A}}}{\tau}, \widetilde{\mathbf{B}})$ with the change of variables:

$$\widetilde{\mathbf{G}} \overset{\text{def}}{=} \frac{\mathbf{G}}{\tau} \tag{85}$$

$$\widetilde{\mathbf{A}} \overset{\text{def}}{=} \tau(\mathbf{A}^{-1} + \frac{1}{\lambda}\,\mathrm{Id})^{-1} \tag{86}$$

$$\widetilde{\mathbf{B}} \overset{\text{def}}{=} \tau(\mathbf{B}^{-1} + \frac{1}{\lambda}\,\mathrm{Id})^{-1} \tag{87}$$

Notice that since $0 < \tau < 1$, $\widetilde{\mathbf{A}}$ and $\widetilde{\mathbf{B}}$ are well-defined and positive definite. Therefore, Proposition 3 applies and we can write in closed form:

$$\mathbf{C} \overset{\text{def}}{=} \widetilde{\mathbf{A}}\widetilde{\mathbf{G}}^{-1} = \left(\frac{1}{\tau}\widetilde{\mathbf{A}}\widetilde{\mathbf{B}} + \frac{\sigma^4}{4}\,\mathrm{Id}\right)^{\frac{1}{2}} - \frac{\sigma^2}{2}\,\mathrm{Id}$$

$$= \widetilde{\mathbf{A}}^{\frac{1}{2}}\left(\frac{1}{\tau}\widetilde{\mathbf{A}}^{\frac{1}{2}}\widetilde{\mathbf{B}}\widetilde{\mathbf{A}}^{\frac{1}{2}} + \frac{\sigma^4}{4}\,\mathrm{Id}\right)^{\frac{1}{2}}\widetilde{\mathbf{A}}^{-\frac{1}{2}} - \frac{\sigma^2}{2}\,\mathrm{Id}$$
(88)

And similarly by symmetry:

$$\widetilde{\mathbf{B}}\mathbf{F}^{-1} = \left(\frac{1}{\tau}\widetilde{\mathbf{B}}\widetilde{\mathbf{A}} + \frac{\sigma^4}{4}\,\mathrm{Id}\right)^{\frac{1}{2}} - \frac{\sigma^2}{2}\,\mathrm{Id} = \mathbf{C}^\top \tag{89}$$

Therefore we obtain $\mathbf{F}$ and $\mathbf{G}$ in closed form:

$$\mathbf{F} = \widetilde{\mathbf{B}}\mathbf{C}^{-1} \tag{90}$$

$$\mathbf{G} = \mathbf{C}^{-1}\widetilde{\mathbf{A}} \tag{91}$$

Finally, to obtain the formulas of $\widetilde{\mathbf{A}}$ and $\widetilde{\mathbf{B}}$ of Theorem 3, use Woodburry's identity to write:

$$\widetilde{\mathbf{B}} = \tau\lambda(\mathrm{Id} - \lambda(\mathbf{B} + \lambda\,\mathrm{Id})^{-1})$$
$$= \frac{\gamma}{\gamma + 2\sigma^2}\frac{2\sigma^2 + \gamma}{2}(\mathrm{Id} - \lambda(\mathbf{B} + \lambda\,\mathrm{Id})^{-1})$$
$$= \frac{\gamma}{2}(\mathrm{Id} - \lambda(\mathbf{B} + \lambda\,\mathrm{Id})^{-1})$$

the same applies for $\widetilde{\mathbf{A}}$.

**Identifying u and v.** Combining the equations in $\mathbf{u}$ and $\mathbf{v}$ leads to:

$$\mathbf{v} = -\tau\mathbf{F}^{-1}(\mathbf{A}^{-1}\mathbf{a} + \tau\mathbf{u})$$
$$\Leftrightarrow \mathbf{F}\mathbf{v} = -\tau\mathbf{A}^{-1}\mathbf{a} - \tau\mathbf{u}$$
$$\Leftrightarrow \mathbf{F}\mathbf{v} = -\tau\mathbf{A}^{-1}\mathbf{a} + \tau^2\mathbf{G}^{-1}(\mathbf{B}^{-1}\mathbf{b} + \mathbf{v})$$
$$\Leftrightarrow \mathbf{G}\mathbf{F}\mathbf{v} = -\tau\mathbf{G}\mathbf{A}^{-1}\mathbf{a} + \tau^2(\mathbf{B}^{-1}\mathbf{b} + \mathbf{v})$$
$$\Leftrightarrow (\mathbf{G}\mathbf{F} - \tau^2\,\mathrm{Id})\mathbf{v} = -\tau\mathbf{G}\mathbf{A}^{-1}\mathbf{a} + \tau^2\mathbf{B}^{-1}\mathbf{b}$$

Similarly, $(\mathbf{F}\mathbf{G} - \tau^2\,\mathrm{Id})\mathbf{u} = -\tau\mathbf{F}\mathbf{B}^{-1}\mathbf{b} + \tau^2\mathbf{A}^{-1}\mathbf{a}$. Moreover, since $0 < \tau < 1$, it holds $(\mathbf{F} - \tau^2\mathbf{G}^{-1}) \succ (\mathbf{F} - \tau\mathbf{G}^{-1}) = \sigma^2\widetilde{\mathbf{A}}^{-1} \succ 0$. Therefore, $(\mathbf{F}\mathbf{G} - \tau^2\,\mathrm{Id}) = (\mathbf{F} - \tau^2\mathbf{G}^{-1}\,\mathrm{Id})\mathbf{G}$ is invertible. The same applies for $(\mathbf{G}\mathbf{F} - \tau^2\,\mathrm{Id})$.

Finally, both equations can be vectorized:

$$\begin{pmatrix} \mathbf{G}\mathbf{F} - \tau^2\,\mathrm{Id} & 0 \\ 0 & \mathbf{F}\mathbf{G} - \tau^2\,\mathrm{Id} \end{pmatrix}\begin{pmatrix} \mathbf{v} \\ \mathbf{u} \end{pmatrix} = \begin{pmatrix} -\tau\mathbf{G} & \tau^2\,\mathrm{Id} \\ \tau^2\,\mathrm{Id} & -\tau\mathbf{F} \end{pmatrix}\begin{pmatrix} \mathbf{A}^{-1} & 0 \\ 0 & \mathbf{B}^{-1} \end{pmatrix}\begin{pmatrix} \mathbf{a} \\ \mathbf{b} \end{pmatrix} \tag{92}$$

**Identifying $m_u m_v$.** Now that $\mathbf{F}, \mathbf{G}, \mathbf{u}$ and $\mathbf{v}$ are given in closed form, $m_u m_v$ is obtained by taking the product of both equations:

$$(m_u m_v)^{\tau+1} = \left(\frac{\sqrt{\det(\mathbf{A}\mathbf{B})\det(\mathbf{F}\mathbf{G})}}{\sigma^{2d}m_\alpha m_\beta}\right)^\tau \exp(-\frac{\tau}{2}(q_{u,\alpha} + q_{v,\beta})) \tag{93}$$

**Transportation plan.** Let $\omega \overset{\text{def}}{=} \frac{m_\alpha m_\beta}{\sqrt{\det(4\pi^2\mathbf{A}\mathbf{B})}}m_u m_v e^{-\frac{1}{2}(\mathbf{a}^\top\mathbf{A}^{-1}\mathbf{a} + \mathbf{b}^\top\mathbf{B}^{-1}\mathbf{b})}$. At optimality, the transport plan $\pi$ is given by:

$$\frac{\mathrm{d}\pi}{\mathrm{d}x\mathrm{d}y}(x,y) = \exp\left(\frac{f(x) + g(y) - \|x-y\|^2}{2\sigma^2}\right)\frac{\mathrm{d}\alpha}{\mathrm{d}x}(x)\frac{\mathrm{d}\beta}{\mathrm{d}y}(y)$$
$$= \omega\exp\left(\mathcal{Q}(\mathbf{A}^{-1}\mathbf{a} + \mathbf{u}, \mathbf{A}^{-1} + \mathbf{U})(x) - \frac{\|x-y\|^2}{2\sigma^2} + \mathcal{Q}(\mathbf{B}^{-1}\mathbf{b} + \mathbf{v}, \mathbf{B}^{-1} + \mathbf{V})(y)\right)$$
$$= \omega\exp\left(\mathcal{Q}(\mathbf{U} + \mathbf{A}^{-1})(x) + \mathcal{Q}(\mathbf{V} + \mathbf{B}^{-1})(y) + \mathcal{Q}\left(\begin{smallmatrix} \frac{\mathrm{Id}}{\sigma^2} & -\frac{\mathrm{Id}}{\sigma^2} \\ -\frac{\mathrm{Id}}{\sigma^2} & \frac{\mathrm{Id}}{\sigma^2} \end{smallmatrix}\right)(x,y)\right)$$
$$= \omega\exp\left(\mathcal{Q}\left(\begin{pmatrix} \mathbf{A}^{-1}\mathbf{a} + \mathbf{u} \\ \mathbf{B}^{-1}\mathbf{b} + \mathbf{v} \end{pmatrix}, \begin{pmatrix} \mathbf{U} + \mathbf{A}^{-1} + \frac{\mathrm{Id}}{\sigma^2} & 0 \\ 0 & \mathbf{V} + \mathbf{B}^{-1} + \frac{\mathrm{Id}}{\sigma^2} \end{pmatrix}\right)(x,y)\right)$$
$$= \omega\exp\left(\mathcal{Q}\left(\begin{pmatrix} \mathbf{A}^{-1}\mathbf{a} + \mathbf{u} \\ \mathbf{B}^{-1}\mathbf{b} + \mathbf{v} \end{pmatrix}, \frac{1}{\sigma^2}\begin{pmatrix} \mathbf{F} & -\mathrm{Id} \\ -\mathrm{Id} & \mathbf{G} \end{pmatrix}\right)(x,y)\right)$$
$$= \omega\exp\left(\mathcal{Q}(\mu, \Gamma)(x,y)\right)$$

with $\mu \stackrel{\text{def}}{=} \begin{pmatrix} \mathbf{A}^{-1}\mathbf{a} + \mathbf{u} \\ \mathbf{B}^{-1}\mathbf{b} + \mathbf{v} \end{pmatrix}$ and $\Gamma \stackrel{\text{def}}{=} \begin{pmatrix} \frac{\mathbf{F}}{\sigma^2} & -\frac{\text{Id}}{\sigma^2} \\ -\frac{\text{Id}}{\sigma^2} & \frac{\mathbf{G}}{\sigma^2} \end{pmatrix}$. Let's show that $\Gamma \succ 0$. Since $\frac{\mathbf{G}}{2\sigma^2} \succ 0$, it is sufficient to show that Schur complement $\frac{\mathbf{F}}{\sigma^2} - \frac{1}{\sigma^2}\mathbf{G}^{-1} \succ 0$. On one hand, with

$$\frac{\mathbf{F} - \mathbf{G}^{-1}}{\sigma^2} = \tau\widetilde{\mathbf{A}}^{-1} - \frac{1}{\lambda}\mathbf{G}^{-1}$$

On the other hand, almost by definition $\widetilde{\mathbf{A}} \prec \tau\lambda\,\text{Id}$ and $\widetilde{\mathbf{B}} \prec \tau\lambda\,\text{Id}$. Thus for any $x \in \mathbb{R}^d$:

$$x^\top \frac{\widetilde{\mathbf{A}}^{\frac{1}{2}}\widetilde{\mathbf{B}}\widetilde{\mathbf{A}}^{\frac{1}{2}}}{\tau} x \le \lambda\|\widetilde{\mathbf{A}}^{\frac{1}{2}}x\|^2 = \lambda x^\top \widetilde{\mathbf{A}} x \le \tau\lambda^2\|x\|^2,$$

which implies

$$\left( \frac{\widetilde{\mathbf{A}}^{\frac{1}{2}}\widetilde{\mathbf{B}}\widetilde{\mathbf{A}}^{\frac{1}{2}}}{\tau} + \frac{\sigma^4}{4}\,\text{Id} \right)^{\frac{1}{2}} \prec \sqrt{\tau\lambda^2 + \frac{\sigma^4}{4}}\,\text{Id} = \frac{\lambda}{2}(\sqrt{4\tau + (1-\tau)^2})\,\text{Id} = \frac{\lambda(1+\tau)}{2}\,\text{Id}\,.$$

Therefore, using the second equality of (88) and inverting (90) to obtain $\mathbf{G}^{-1}$:

$$x^\top\mathbf{G}^{-1}x = x^\top\widetilde{\mathbf{A}}^{-\frac{1}{2}} \left( \left( \frac{\widetilde{\mathbf{A}}^{\frac{1}{2}}\widetilde{\mathbf{B}}\widetilde{\mathbf{A}}^{\frac{1}{2}}}{\tau} + \frac{\sigma^4}{4}\,\text{Id} \right)^{\frac{1}{2}} - \frac{\sigma^2}{2}\,\text{Id}) \right) \widetilde{\mathbf{A}}^{-\frac{1}{2}}x$$

$$= (\widetilde{\mathbf{A}}^{-\frac{1}{2}}x)^\top \left( \left( \frac{\widetilde{\mathbf{A}}^{\frac{1}{2}}\widetilde{\mathbf{B}}\widetilde{\mathbf{A}}^{\frac{1}{2}}}{\tau} + \frac{\sigma^4}{4}\,\text{Id} \right)^{\frac{1}{2}} - \frac{\lambda(1-\tau)}{2}\,\text{Id}) \right) (\widetilde{\mathbf{A}}^{-\frac{1}{2}}x)$$

$$\le (\widetilde{\mathbf{A}}^{-\frac{1}{2}}x)^\top \left( \frac{\lambda(1+\tau)}{2}\,\text{Id} - \frac{\lambda(1-\tau)}{2}\,\text{Id}) \right) (\widetilde{\mathbf{A}}^{-\frac{1}{2}}x)$$

$$= \tau\lambda x^\top\widetilde{\mathbf{A}}^{-1}x.$$

Thus $\mathbf{G}^{-1} \prec \tau\lambda\widetilde{\mathbf{A}}^{-1}$. We can therefore conclude that the Schur complement $\frac{1}{\sigma^2}(\mathbf{F} - \mathbf{G}^{-1})$ is positive definite. By completing the square, we can factor $\frac{\mathrm{d}\pi}{\mathrm{d}x\mathrm{d}x}$ as a Gaussian density. Let $z \stackrel{\text{def}}{=} \binom{x}{y}$:

$$\frac{\mathrm{d}\pi}{\mathrm{d}x\mathrm{d}y}(x,y) = \omega\exp\left(\mathcal{Q}(\mu,\Gamma)(x,y)\right)$$

$$= \omega\exp\left(-\frac{1}{2}(z^\top\Gamma z - 2z^\top\mu)\right)$$

$$= \omega\exp\left(\frac{1}{2}\mu^\top\Gamma^{-1}\mu - \frac{1}{2}(z - \Gamma^{-1}\mu)^\top\Gamma(z - \Gamma^{-1}\mu))\right)$$

$$= \omega e^{\frac{1}{2}\mu^\top\Gamma^{-1}\mu}\mathcal{N}(\mathbf{H}\mu,\mathbf{H})(z),$$

where $\mathbf{H} = \Gamma^{-1}$.

**Detailed expressions.** To conclude the proof of Theorem 3, we need to simplify the formulas of $m$, $\mathbf{H}\mu$ and $\mathbf{H}$. First, we will start with the mean $\mathbf{H}\mu$.

$\mathbf{H}\mu$    Using the optimality conditions of Proposition 9 and the closed form formula of $\mathbf{v}$ and $\mathbf{u}$:

$$
\begin{aligned}
\mu &= \begin{pmatrix} \mathbf{A}^{-1}\mathbf{a} + \mathbf{u} \\ \mathbf{B}^{-1}\mathbf{b} + \mathbf{v} \end{pmatrix} \\[4pt]
&= -\frac{1}{\tau} \begin{pmatrix} \mathbf{F}\mathbf{v} \\ \mathbf{G}\mathbf{u} \end{pmatrix} \\[4pt]
&= -\frac{1}{\tau} \begin{pmatrix} \mathbf{F} & 0 \\ 0 & \mathbf{G} \end{pmatrix} \begin{pmatrix} \mathbf{v} \\ \mathbf{u} \end{pmatrix} \\[4pt]
&= -\frac{1}{\tau} \begin{pmatrix} \mathbf{F} & 0 \\ 0 & \mathbf{G} \end{pmatrix} \begin{pmatrix} \mathbf{GF} - \tau^2\,\mathrm{Id} & 0 \\ 0 & \mathbf{FG} - \tau^2\,\mathrm{Id} \end{pmatrix}^{-1} \begin{pmatrix} -\tau\mathbf{G} & \tau^2\,\mathrm{Id} \\ \tau^2\,\mathrm{Id} & -\tau\mathbf{F} \end{pmatrix} \begin{pmatrix} \mathbf{A}^{-1} & 0 \\ 0 & \mathbf{B}^{-1} \end{pmatrix} \begin{pmatrix} \mathbf{a} \\ \mathbf{b} \end{pmatrix} \\[4pt]
&= \begin{pmatrix} \mathbf{F} & 0 \\ 0 & \mathbf{G} \end{pmatrix} \begin{pmatrix} \mathbf{GF} - \tau^2\,\mathrm{Id} & 0 \\ 0 & \mathbf{FG} - \tau^2\,\mathrm{Id} \end{pmatrix}^{-1} \begin{pmatrix} \mathbf{G} & -\tau\,\mathrm{Id} \\ -\tau\,\mathrm{Id} & \mathbf{F} \end{pmatrix} \begin{pmatrix} \mathbf{A}^{-1} & 0 \\ 0 & \mathbf{B}^{-1} \end{pmatrix} \begin{pmatrix} \mathbf{a} \\ \mathbf{b} \end{pmatrix} \\[4pt]
&= \begin{pmatrix} \mathbf{F} & 0 \\ 0 & \mathbf{G} \end{pmatrix} \begin{pmatrix} (\mathbf{F} - \tau^2\mathbf{G}^{-1})^{-1} & -\tau(\mathbf{GF} - \tau^2\,\mathrm{Id})^{-1} \\ -\tau(\mathbf{FG} - \tau^2\,\mathrm{Id})^{-1} & (\mathbf{G} - \tau^2\mathbf{F}^{-1})^{-1} \end{pmatrix} \begin{pmatrix} \mathbf{A}^{-1} & 0 \\ 0 & \mathbf{B}^{-1} \end{pmatrix} \begin{pmatrix} \mathbf{a} \\ \mathbf{b} \end{pmatrix} \\[4pt]
&= \begin{pmatrix} \mathbf{F} & 0 \\ 0 & \mathbf{G} \end{pmatrix} \begin{pmatrix} \mathbf{F} & \tau\,\mathrm{Id} \\ \tau\,\mathrm{Id} & \mathbf{G} \end{pmatrix}^{-1} \begin{pmatrix} \mathbf{A}^{-1} & 0 \\ 0 & \mathbf{B}^{-1} \end{pmatrix} \begin{pmatrix} \mathbf{a} \\ \mathbf{b} \end{pmatrix} \\[4pt]
&= \begin{pmatrix} \mathrm{Id} & \tau\mathbf{G}^{-1} \\ \tau\mathbf{F}^{-1} & \mathrm{Id} \end{pmatrix}^{-1} \begin{pmatrix} \mathbf{A}^{-1} & 0 \\ 0 & \mathbf{B}^{-1} \end{pmatrix} \begin{pmatrix} \mathbf{a} \\ \mathbf{b} \end{pmatrix}
\end{aligned}
\tag{94}
$$

Therefore:

$$
\begin{aligned}
\mathbf{H}\mu &= \sigma^2 \begin{pmatrix} \mathbf{F} & -\mathrm{Id} \\ -\mathrm{Id} & \mathbf{G} \end{pmatrix}^{-1} \begin{pmatrix} \mathrm{Id} & \tau\mathbf{G}^{-1}\,\mathrm{Id} \\ \tau\mathbf{F}^{-1}\,\mathrm{Id} & \mathrm{Id} \end{pmatrix}^{-1} \begin{pmatrix} \mathbf{A}^{-1} & 0 \\ 0 & \mathbf{B}^{-1} \end{pmatrix} \begin{pmatrix} \mathbf{a} \\ \mathbf{b} \end{pmatrix} \\[4pt]
&= \sigma^2 \left( \begin{pmatrix} \mathrm{Id} & \tau\mathbf{G}^{-1}\,\mathrm{Id} \\ \tau\mathbf{F}^{-1}\,\mathrm{Id} & \mathrm{Id} \end{pmatrix} \begin{pmatrix} \mathbf{F} & -\mathrm{Id} \\ -\mathrm{Id} & \mathbf{G} \end{pmatrix} \right)^{-1} \begin{pmatrix} \mathbf{A}^{-1} & 0 \\ 0 & \mathbf{B}^{-1} \end{pmatrix} \begin{pmatrix} \mathbf{a} \\ \mathbf{b} \end{pmatrix} \\[4pt]
&= \sigma^2 \begin{pmatrix} \mathbf{F} - \tau\mathbf{G}^{-1} & -(1-\tau)\,\mathrm{Id} \\ -(1-\tau)\,\mathrm{Id} & \mathbf{G} - \tau\mathbf{F}^{-1} \end{pmatrix}^{-1} \begin{pmatrix} \mathbf{A}^{-1} & 0 \\ 0 & \mathbf{B}^{-1} \end{pmatrix} \begin{pmatrix} \mathbf{a} \\ \mathbf{b} \end{pmatrix} \\[4pt]
&= \sigma^2 \begin{pmatrix} \sigma^2\mathbf{A}^{-1} + (1-\tau)\,\mathrm{Id} & -(1-\tau)\,\mathrm{Id} \\ -(1-\tau)\,\mathrm{Id} & \sigma^2\mathbf{B}^{-1} + (1-\tau)\,\mathrm{Id} \end{pmatrix}^{-1} \begin{pmatrix} \mathbf{A}^{-1} & 0 \\ 0 & \mathbf{B}^{-1} \end{pmatrix} \begin{pmatrix} \mathbf{a} \\ \mathbf{b} \end{pmatrix} \\[4pt]
&= \begin{pmatrix} \mathbf{A}^{-1} + \mathrm{Id} & -\lambda\,\mathrm{Id} \\ -\lambda\,\mathrm{Id} & \mathbf{B}^{-1} + \lambda\,\mathrm{Id} \end{pmatrix}^{-1} \begin{pmatrix} \mathbf{A}^{-1} & 0 \\ 0 & \mathbf{B}^{-1} \end{pmatrix} \begin{pmatrix} \mathbf{a} \\ \mathbf{b} \end{pmatrix}
\end{aligned}
\tag{95}
$$

Let's compute the inverse of:

$$
\mathbf{Z} \stackrel{\mathrm{def}}{=} \begin{pmatrix} \mathbf{A}^{-1} + \frac{1}{\lambda}\,\mathrm{Id} & -\frac{1}{\lambda}\,\mathrm{Id} \\ -\frac{1}{\lambda}\,\mathrm{Id} & \mathbf{B}^{-1} + \frac{1}{\lambda}\,\mathrm{Id} \end{pmatrix}.
\tag{96}
$$

Let $\mathbf{S}$ and $\mathbf{S}'$ be the respective Schur complements of $\mathbf{A}^{-1} + \frac{1}{\lambda}\,\mathrm{Id}$ and $\mathbf{B}^{-1} + \frac{1}{\lambda}\,\mathrm{Id}$ in $\mathbf{Z}$. The block inverse formula writes:

$$
\mathbf{Z}^{-1} = \begin{pmatrix} \mathbf{S} & \frac{1}{\lambda}\mathbf{S}(\mathbf{B}^{-1} + \frac{1}{\lambda}\,\mathrm{Id})^{-1} \\ \frac{1}{\lambda}(\mathbf{A}^{-1} + \frac{1}{\lambda}\,\mathrm{Id})^{-1}\mathbf{S} & \mathbf{S}' \end{pmatrix}.
$$

Using Woodbury's identity twice and denoting $\mathbf{X} \stackrel{\mathrm{def}}{=} \mathbf{A} + \mathbf{B} + \lambda\,\mathrm{Id}$:

$$
\begin{aligned}
\mathbf{S} &= (\mathbf{A}^{-1} + \frac{1}{\lambda}\,\mathrm{Id} - \frac{1}{\lambda^2}(\mathbf{B}^{-1} + \frac{1}{\lambda}\,\mathrm{Id})^{-1})^{-1} \\
&= (\mathbf{A}^{-1} + (\mathbf{B} + \lambda\,\mathrm{Id})^{-1})^{-1} \\
&= (\mathbf{A} - \mathbf{A}(\mathbf{A} + \mathbf{B} + \lambda\,\mathrm{Id})^{-1}\mathbf{A}) \\
&= \mathbf{A} - \mathbf{A}\mathbf{X}^{-1}\mathbf{A}.
\end{aligned}
$$

And similarly: $\mathbf{S}' = \mathbf{B} - \mathbf{B}\mathbf{X}^{-1}\mathbf{B}$. The off-diagonal blocks can be simplified as well:

$$
\begin{aligned}
\frac{1}{\lambda}\mathbf{S}(\mathbf{B}^{-1} + \tfrac{1}{\lambda}\operatorname{Id})^{-1} &= \frac{1}{\lambda}(\mathbf{A}^{-1} + (\mathbf{B} + \lambda\operatorname{Id})^{-1})^{-1}(\mathbf{B}^{-1} + \tfrac{1}{\lambda}\operatorname{Id})^{-1} \\
&= (\mathbf{A}^{-1} + (\mathbf{B} + \lambda\operatorname{Id})^{-1})^{-1}(\lambda\operatorname{Id} + \mathbf{B}\operatorname{Id})^{-1}\mathbf{B} \\
&= \left((\mathbf{B} + \lambda\operatorname{Id}) - (\mathbf{B} + \lambda\operatorname{Id})(\mathbf{A} + \mathbf{B} + \lambda\operatorname{Id})^{-1}(\mathbf{B} + \lambda\operatorname{Id})\right)(\lambda\operatorname{Id} + \mathbf{B}\operatorname{Id})^{-1}\mathbf{B} \\
&= \mathbf{B} - (\mathbf{B} + \lambda\operatorname{Id})\mathbf{X}^{-1}\mathbf{B} \\
&= \mathbf{B} - (\mathbf{X} - \mathbf{A})\mathbf{X}^{-1}\mathbf{B} \\
&= \mathbf{A}\mathbf{X}^{-1}\mathbf{B}.
\end{aligned}
$$

Similarly, $\frac{1}{\lambda}(\mathbf{A}^{-1} + \tfrac{1}{\lambda}\operatorname{Id})^{-1}\mathbf{S} = \mathbf{B}\mathbf{X}^{-1}\mathbf{A}$. Thus, the inverse of $\mathbf{Z}$ is given by:

$$
\mathbf{Z}^{-1} = \begin{pmatrix} \mathbf{A} - \mathbf{A}\mathbf{X}^{-1}\mathbf{A} & \mathbf{A}\mathbf{X}^{-1}\mathbf{B} \\ \mathbf{B}\mathbf{X}^{-1}\mathbf{A} & \mathbf{B} - \mathbf{B}\mathbf{X}^{-1}\mathbf{B} \end{pmatrix}. \tag{97}
$$

and finally:

$$
\begin{aligned}
\mathbf{H}\mu &= \mathbf{Z}^{-1} \begin{pmatrix} \mathbf{A}^{-1} & 0 \\ 0 & \mathbf{B}^{-1} \end{pmatrix} \begin{pmatrix} \mathbf{a} \\ \mathbf{b} \end{pmatrix} = \begin{pmatrix} \operatorname{Id} - \mathbf{A}\mathbf{X}^{-1} & \mathbf{A}\mathbf{X}^{-1} \\ \mathbf{B}\mathbf{X}^{-1} & \operatorname{Id} - \mathbf{B}\mathbf{X}^{-1} \end{pmatrix} \begin{pmatrix} \mathbf{a} \\ \mathbf{b} \end{pmatrix} \\
&= \begin{pmatrix} \mathbf{a} + \mathbf{A}\mathbf{X}^{-1}(\mathbf{b} - \mathbf{a}) \\ \mathbf{b} + \mathbf{B}\mathbf{X}^{-1}(\mathbf{a} - \mathbf{b}) \end{pmatrix}
\end{aligned}
$$

**Finding the covariance matrix H.** To compute $\mathbf{H} = \left(\frac{1}{\sigma^2}\begin{pmatrix} \mathbf{F} & -\operatorname{Id} \\ -\operatorname{Id} & \mathbf{G} \end{pmatrix}\right)^{-1}$ one may use the block inverse formula. However, the Schur complement $(\mathbf{F} - \mathbf{G}^{-1})^{-1}$ is not easy to manipulate. Instead notice that the following holds:

$$
\begin{aligned}
\frac{1}{\sigma^2}\begin{pmatrix} \mathbf{F} & -\operatorname{Id} \\ -\operatorname{Id} & \mathbf{G} \end{pmatrix}\begin{pmatrix} \operatorname{Id} & \tau\mathbf{F}^{-1} \\ \tau\mathbf{G}^{-1} & \operatorname{Id} \end{pmatrix} &= \frac{1}{\sigma^2}\begin{pmatrix} \mathbf{F} - \tau\mathbf{G}^{-1} & -(1-\tau)\operatorname{Id} \\ -(1-\tau)\operatorname{Id} & \mathbf{G} - \tau\mathbf{F}^{-1} \end{pmatrix} \\
&= \begin{pmatrix} \mathbf{A}^{-1} + \tfrac{1}{\lambda}\operatorname{Id} & -\tfrac{1}{\lambda}\operatorname{Id} \\ -\tfrac{1}{\lambda}\operatorname{Id} & \mathbf{B}^{-1} + \tfrac{1}{\lambda}\operatorname{Id} \end{pmatrix},
\end{aligned}
$$

where the last equality follows from the optimality conditions (84). Therefore:

$$
\mathbf{H} = \begin{pmatrix} \operatorname{Id} & \tau\mathbf{F}^{-1} \\ \tau\mathbf{G}^{-1} & \operatorname{Id} \end{pmatrix}\begin{pmatrix} \mathbf{A}^{-1} + \tfrac{1}{\lambda}\operatorname{Id} & -\tfrac{1}{\lambda}\operatorname{Id} \\ -\tfrac{1}{\lambda}\operatorname{Id} & \mathbf{B}^{-1} + \tfrac{1}{\lambda}\operatorname{Id} \end{pmatrix}^{-1}.
$$

Notice that we have already computed the inverse matrix on the right side above in the developments of $\mathbf{H}\mu$. Thus:

$$
\begin{aligned}
\mathbf{H} &= \begin{pmatrix} \mathrm{Id} & \tau\mathbf{F}^{-1} \\ \tau\mathbf{G}^{-1} & \mathrm{Id} \end{pmatrix} \begin{pmatrix} \mathbf{A} - \mathbf{A}\mathbf{X}^{-1}\mathbf{A} & \mathbf{A}\mathbf{X}^{-1}\mathbf{B} \\ \mathbf{B}\mathbf{X}^{-1}\mathbf{A} & \mathbf{B} - \mathbf{B}\mathbf{X}^{-1}\mathbf{B} \end{pmatrix} \\
&= \begin{pmatrix} \mathrm{Id} & \tau\mathbf{C}\widetilde{\mathbf{B}}^{-1} \\ \mathbf{C}^{\top}\widetilde{\mathbf{A}}^{-1} & \mathrm{Id} \end{pmatrix} \begin{pmatrix} \mathbf{A} - \mathbf{A}\mathbf{X}^{-1}\mathbf{A} & \mathbf{A}\mathbf{X}^{-1}\mathbf{B} \\ \mathbf{B}\mathbf{X}^{-1}\mathbf{A} & \mathbf{B} - \mathbf{B}\mathbf{X}^{-1}\mathbf{B} \end{pmatrix} \\
&= \begin{pmatrix} \mathrm{Id} & \mathbf{C}(\mathbf{B}^{-1} + \frac{1}{\lambda}\mathrm{Id}) \\ \mathbf{C}^{\top}(\mathbf{A}^{-1} + \frac{1}{\lambda}\mathrm{Id}) & \mathrm{Id} \end{pmatrix} \begin{pmatrix} \mathbf{A} - \mathbf{A}\mathbf{X}^{-1}\mathbf{A} & \mathbf{A}\mathbf{X}^{-1}\mathbf{B} \\ \mathbf{B}\mathbf{X}^{-1}\mathbf{A} & \mathbf{B} - \mathbf{B}\mathbf{X}^{-1}\mathbf{B} \end{pmatrix} \\
&= \begin{pmatrix} \mathrm{Id} & \mathbf{C}(\mathbf{B}^{-1} + \frac{1}{\lambda}\mathrm{Id}) \\ \mathbf{C}^{\top}(\mathbf{A}^{-1} + \frac{1}{\lambda}\mathrm{Id}) & \mathrm{Id} \end{pmatrix} \begin{pmatrix} \mathbf{A} - \mathbf{A}\mathbf{X}^{-1}\mathbf{A} & \mathbf{A}\mathbf{X}^{-1}\mathbf{B} \\ \mathbf{B}\mathbf{X}^{-1}\mathbf{A} & \mathbf{B} - \mathbf{B}\mathbf{X}^{-1}\mathbf{B} \end{pmatrix} \\
&= \begin{pmatrix} \mathrm{Id} & \frac{1}{\lambda}\mathbf{C}(\lambda\mathrm{Id}+\mathbf{B})\mathbf{B}^{-1} \\ \frac{1}{\lambda}\mathbf{C}^{\top}\mathbf{C}(\lambda\mathrm{Id}+\mathbf{A})\mathbf{A}^{-1} & \mathrm{Id} \end{pmatrix} \begin{pmatrix} \mathbf{A} - \mathbf{A}\mathbf{X}^{-1}\mathbf{A} & \mathbf{A}\mathbf{X}^{-1}\mathbf{B} \\ \mathbf{B}\mathbf{X}^{-1}\mathbf{A} & \mathbf{B} - \mathbf{B}\mathbf{X}^{-1}\mathbf{B} \end{pmatrix} \\
&= \begin{pmatrix} \mathrm{Id} & \frac{1}{\lambda}\mathbf{C}(\mathbf{X} - \mathbf{A})\mathbf{B}^{-1} \\ \frac{1}{\lambda}\mathbf{C}^{\top}(\mathbf{X} - \mathbf{B})\mathbf{A}^{-1} & \mathrm{Id} \end{pmatrix} \begin{pmatrix} \mathbf{A} - \mathbf{A}\mathbf{X}^{-1}\mathbf{A} & \mathbf{A}\mathbf{X}^{-1}\mathbf{B} \\ \mathbf{B}\mathbf{X}^{-1}\mathbf{A} & \mathbf{B} - \mathbf{B}\mathbf{X}^{-1}\mathbf{B} \end{pmatrix} \\
&= \begin{pmatrix} \mathbf{A} - \mathbf{A}\mathbf{X}^{-1}\mathbf{A} + \frac{1}{\lambda}\mathbf{C}(\mathbf{A} - \mathbf{A}\mathbf{X}^{-1}\mathbf{A}) & \mathbf{A}\mathbf{X}^{-1}\mathbf{B} + \frac{1}{\lambda}\mathbf{C}(\mathbf{X} - \mathbf{A})(\mathrm{Id}-\mathbf{X}^{-1}\mathbf{B}) \\ \frac{1}{\lambda}\mathbf{C}^{\top}(\mathbf{X} - \mathbf{B})(\mathrm{Id}-\mathbf{X}^{-1}\mathbf{A}) + \mathbf{B}\mathbf{X}^{-1}\mathbf{A} & \frac{1}{\lambda}\mathbf{C}^{\top}(\mathbf{X} - \mathbf{B})\mathbf{X}^{-1}\mathbf{B} + \mathbf{B} - \mathbf{B}\mathbf{X}^{-1}\mathbf{B} \end{pmatrix} \\
&= \begin{pmatrix} (\mathrm{Id}+\frac{1}{\lambda}\mathbf{C})(\mathbf{A} - \mathbf{A}\mathbf{X}^{-1}\mathbf{A}) & \mathbf{A}\mathbf{X}^{-1}\mathbf{B} + \frac{1}{\lambda}\mathbf{C}(\mathbf{X} - \mathbf{A} - \mathbf{B} + \mathbf{A}\mathbf{X}^{-1}\mathbf{B}) \\ \lambda\mathbf{C}^{\top}(\lambda\mathrm{Id}+\mathbf{B}\mathbf{X}^{-1}\mathbf{A}) + \mathbf{B}\mathbf{X}^{-1}\mathbf{A} & \frac{1}{\lambda}\mathbf{C}^{\top}(\mathbf{X} - \mathbf{B})\mathbf{X}^{-1}\mathbf{B} + \mathbf{B} - \mathbf{B}\mathbf{X}^{-1}\mathbf{B} \end{pmatrix} \\
&= \begin{pmatrix} (\mathrm{Id}+\frac{1}{\lambda}\mathbf{C})(\mathbf{A} - \mathbf{A}\mathbf{X}^{-1}\mathbf{A}) & \mathbf{A}\mathbf{X}^{-1}\mathbf{B} + \frac{1}{\lambda}\mathbf{C}(\lambda\mathrm{Id}+\mathbf{A}\mathbf{X}^{-1}\mathbf{B}) \\ \mathbf{C}^{\top} + \frac{1}{\lambda}\mathbf{C}^{\top}\mathbf{B}\mathbf{X}^{-1}\mathbf{A} + \mathbf{B}\mathbf{X}^{-1}\mathbf{A} & (\mathrm{Id}+\frac{1}{\lambda}\mathbf{C}^{\top})(\mathbf{B} - \mathbf{B}\mathbf{X}^{-1}\mathbf{B}) \end{pmatrix} \\
&= \begin{pmatrix} (\mathrm{Id}+\frac{1}{\lambda}\mathbf{C})(\mathbf{A} - \mathbf{A}\mathbf{X}^{-1}\mathbf{A}) & \mathbf{C} + (\mathrm{Id}+\frac{1}{\lambda}\mathbf{C})\mathbf{A}\mathbf{X}^{-1}\mathbf{B} \\ \mathbf{C}^{\top} + (\mathrm{Id}+\frac{1}{\lambda}\mathbf{C}^{\top})\mathbf{B}\mathbf{X}^{-1}\mathbf{A} & (\mathrm{Id}+\frac{1}{\lambda}\mathbf{C}^{\top})(\mathbf{B} - \mathbf{B}\mathbf{X}^{-1}\mathbf{B}) \end{pmatrix}.
\end{aligned}
$$

**Finding the mass of the plan $\pi$.** The optimal transport plan is given by:

$$
\frac{\mathrm{d}\pi}{\mathrm{d}x\mathrm{d}y}(x, y) = \omega e^{\frac{1}{2}\mu^{\top}\Gamma^{-1}\mu}\sqrt{\det(2\pi\mathbf{H})}\mathcal{N}(\mathbf{H}\mu, \mathbf{H})(z), \tag{98}
$$

where

$$
\begin{aligned}
\omega &= \frac{m_{\alpha}m_{\beta}}{\sqrt{\det(4\pi^2\mathbf{A}\mathbf{B})}}m_u m_v e^{-\frac{1}{2}(\mathbf{a}^{\top}\mathbf{A}^{-1}\mathbf{a}+\mathbf{b}^{\top}\mathbf{B}^{-1}\mathbf{b})} \\
&= \frac{m_{\alpha}m_{\beta}}{\sqrt{\det(4\pi^2\mathbf{A}\mathbf{B})}}\left(\frac{\sqrt{\det(\mathbf{A}\mathbf{B})\det(\mathbf{F}\mathbf{G})}}{\sigma^{2d}m_{\alpha}m_{\beta}}\right)^{\frac{\tau}{\tau+1}} e^{-\frac{\tau}{2(\tau+1)}(q_{u,\alpha}+q_{v,\beta})}e^{-\frac{1}{2}(\mathbf{a}^{\top}\mathbf{A}^{-1}\mathbf{a}+\mathbf{b}^{\top}\mathbf{B}^{-1}\mathbf{b})} \\
&= \frac{1}{(2\pi)^d}\left(\frac{m_{\alpha}m_{\beta}}{\sqrt{\det(\mathbf{A}\mathbf{B})}}\right)^{\frac{1}{\tau+1}}\left(\frac{\sqrt{\det(\mathbf{F}\mathbf{G})}}{\sigma^{2d}}\right)^{\frac{\tau}{\tau+1}} e^{-\frac{\tau}{2(\tau+1)}(q_{u,\alpha}+q_{v,\beta})}e^{-\frac{1}{2}(\mathbf{a}^{\top}\mathbf{A}^{-1}\mathbf{a}+\mathbf{b}^{\top}\mathbf{B}^{-1}\mathbf{b})}.
\end{aligned}
$$

First, let's simplify the argument of the exponential terms. Isolating the terms that depend only on the input means $\mathbf{a}, \mathbf{b}$ it holds: $q_{u,\alpha} + q_{v,\beta} = \frac{\sigma^2}{\tau^2}(\mathbf{v}^{\top}\mathbf{F}\mathbf{v} + \mathbf{u}^{\top}\mathbf{G}\mathbf{u}) + \mathbf{a}^{\top}\mathbf{A}^{-1}\mathbf{a} + \mathbf{b}^{\top}\mathbf{B}^{-1}\mathbf{b}$. Therefore, the full exponential argument is given by:

$$
\phi \overset{\mathrm{def}}{=} \mu^{\top}\Gamma^{-1}\mu - \frac{\tau}{\tau+1}\frac{\sigma^2}{\tau^2}(\mathbf{v}^{\top}\mathbf{F}\mathbf{v} + \mathbf{u}^{\top}\mathbf{G}\mathbf{u}) - \frac{1}{\tau+1}(\mathbf{a}^{\top}\mathbf{A}^{-1}\mathbf{a}+\mathbf{b}^{\top}\mathbf{B}^{-1}\mathbf{b}) \tag{99}
$$

On one hand, using Equation (95) we replace $\mu$:

$$
\begin{aligned}
\mu^{\top}\Gamma^{-1}\mu &= \mu^{\top}\mathbf{H}\mu \\
&= \sigma^2 \begin{pmatrix} \mathbf{A}^{-1}\mathbf{a} \\ \mathbf{B}^{-1}\mathbf{b} \end{pmatrix}^{\top} \begin{pmatrix} \mathrm{Id} & \tau\mathbf{F}^{-1} \\ \tau\mathbf{G}^{-1} & \mathrm{Id} \end{pmatrix}^{-1} \begin{pmatrix} \mathbf{F} & -\mathrm{Id} \\ -\mathrm{Id} & \mathbf{G} \end{pmatrix}^{-1} \begin{pmatrix} \mathrm{Id} & \tau\mathbf{G}^{-1} \\ \tau\mathbf{F}^{-1} & \mathrm{Id} \end{pmatrix}^{-1} \begin{pmatrix} \mathbf{A}^{-1}\mathbf{a} \\ \mathbf{B}^{-1}\mathbf{b} \end{pmatrix}
\end{aligned}
$$

On the other hand:

$$\frac{\sigma^2}{\tau^2}(\mathbf{v}^\top \mathbf{F}\mathbf{v} + \mathbf{u}^\top \mathbf{G}\mathbf{u}) = \sigma^2((\mathbf{A}^{-1}\mathbf{a} + \mathbf{u})^\top \mathbf{F}^{-1}(\mathbf{A}^{-1}\mathbf{a} + \mathbf{u}) + (\mathbf{B}^{-1}\mathbf{b} + \mathbf{v})^\top \mathbf{G}^{-1}(\mathbf{B}^{-1}\mathbf{b} + \mathbf{v}))$$

$$= \sigma^2 \mu^\top \begin{pmatrix} \mathbf{F}^{-1} & 0 \\ 0 & \mathbf{G}^{-1} \end{pmatrix} \mu$$

$$= \sigma^2 \begin{pmatrix} \mathbf{A}^{-1}\mathbf{a} \\ \mathbf{B}^{-1}\mathbf{b} \end{pmatrix}^\top \begin{pmatrix} \mathrm{Id} & \tau\mathbf{F}^{-1} \\ \tau\mathbf{G}^{-1} & \mathrm{Id} \end{pmatrix}^{-1} \begin{pmatrix} \mathbf{F}^{-1} & 0 \\ 0 & \mathbf{G}^{-1} \end{pmatrix} \begin{pmatrix} \mathrm{Id} & \tau\mathbf{G}^{-1} \\ \tau\mathbf{F}^{-1} & \mathrm{Id} \end{pmatrix}^{-1} \begin{pmatrix} \mathbf{A}^{-1}\mathbf{a} \\ \mathbf{B}^{-1}\mathbf{b} \end{pmatrix}$$

Let $\mathbf{J} = \begin{pmatrix} \mathrm{Id} & \tau\mathbf{G}^{-1} \\ \tau\mathbf{F}^{-1} & \mathrm{Id} \end{pmatrix}$ and $\mathbf{K} = \begin{pmatrix} \mathbf{F} & 0 \\ 0 & \mathbf{G} \end{pmatrix}$. It holds:

$$\mu^\top \Gamma^{-1} \mu - \frac{\tau}{\tau+1}\frac{\sigma^2}{\tau^2}(\mathbf{v}^\top \mathbf{F}\mathbf{v} + \mathbf{u}^\top \mathbf{G}\mathbf{u}) = \begin{pmatrix} \mathbf{A}^{-1}\mathbf{a} \\ \mathbf{B}^{-1}\mathbf{b} \end{pmatrix}^\top \mathbf{J}^{\top-1}(\mathbf{H} - \frac{\sigma^2\tau}{\tau+1}\mathbf{K}^{-1})\mathbf{J}^{-1} \begin{pmatrix} \mathbf{A}^{-1}\mathbf{a} \\ \mathbf{B}^{-1}\mathbf{b} \end{pmatrix}$$

Let's compute the matrix $\mathbf{J}^{\top-1}(\mathbf{H} - \frac{\tau\sigma^2}{\tau+1}\mathbf{K}^{-1})\mathbf{J}^{-1}$. First keep in mind that $\mathbf{J}\mathbf{K} = \begin{pmatrix} \mathbf{F} & \tau\,\mathrm{Id} \\ \tau\,\mathrm{Id} & \mathbf{G} \end{pmatrix}$.
Now using Woodburry's identity:

$$\left(\mathbf{J}^{\top-1}(\mathbf{H} - \frac{\tau}{\tau+1}\mathbf{K}^{-1})\mathbf{J}^{-1}\right)^{-1} = \mathbf{J}(\mathbf{H} - \frac{\tau\sigma^2}{\tau+1}\mathbf{K}^{-1})^{-1}\mathbf{J}^\top$$

$$= \mathbf{J}\left(-\frac{\tau+1}{\tau\sigma^2}\mathbf{K} - \left(\frac{\tau+1}{\tau\sigma^2}\right)^2 \mathbf{K}(\mathbf{H}^{-1} - \frac{\tau+1}{\tau\sigma^2}\mathbf{K})^{-1}\mathbf{K}\right)\mathbf{J}^\top$$

$$= \frac{\tau+1}{\tau\sigma^2}\left(-\mathbf{J}\mathbf{K}\mathbf{J}^\top - \frac{\tau+1}{\tau\sigma^2}\mathbf{J}\mathbf{K}\left(\begin{pmatrix} -\frac{\mathbf{F}}{\tau\sigma^2} & -\frac{1}{\sigma^2}\mathrm{Id} \\ -\frac{1}{\sigma^2}\mathrm{Id} & -\frac{\mathbf{G}}{\tau\sigma^2} \end{pmatrix}\right)^{-1}(\mathbf{J}\mathbf{K}^\top)^\top\right)$$

$$= \frac{\tau+1}{\tau\sigma^2}\left(-\mathbf{J}\mathbf{K}\mathbf{J}^\top + (\tau+1)\mathbf{J}\mathbf{K}\left(\begin{pmatrix} \mathbf{F} & \tau\,\mathrm{Id} \\ \tau\,\mathrm{Id} & \mathbf{G} \end{pmatrix}\right)^{-1}(\mathbf{J}\mathbf{K}^\top)^\top\right)$$

$$= \frac{\tau+1}{\tau\sigma^2}\left(-\begin{pmatrix} \mathbf{F} & \tau\,\mathrm{Id} \\ \tau\,\mathrm{Id} & \mathbf{G} \end{pmatrix}\begin{pmatrix} \mathrm{Id} & \tau\mathbf{F}^{-1} \\ \tau\mathbf{G}^{-1} & \mathrm{Id} \end{pmatrix} + (\tau+1)\begin{pmatrix} \mathbf{F} & \tau\,\mathrm{Id} \\ \tau\,\mathrm{Id} & \mathbf{G} \end{pmatrix}\right)$$

$$= \frac{\tau+1}{\tau\sigma^2}\begin{pmatrix} -\mathbf{F} - \tau^2\mathbf{G}^{-1} + (\tau+1)\mathbf{F} & (-2\tau + \tau(\tau+1))\,\mathrm{Id} \\ (-2\tau + \tau(\tau+1))\,\mathrm{Id} & -\mathbf{G} - \tau^2\mathbf{F}^{-1} + (\tau+1)\mathbf{G} \end{pmatrix}$$

$$= \frac{\tau+1}{\sigma^2}\begin{pmatrix} \mathbf{F} - \tau\mathbf{G}^{-1} & -(1-\tau)\,\mathrm{Id} \\ -(1-\tau)\,\mathrm{Id} & \mathbf{G} - \tau\mathbf{F}^{-1} \end{pmatrix}$$

$$= (\tau+1)\begin{pmatrix} \mathbf{A}^{-1} + \frac{1}{\lambda}\mathrm{Id} & -\frac{1}{\lambda}\mathrm{Id} \\ -\frac{1}{\lambda}\mathrm{Id} & \mathbf{B}^{-1} + \frac{1}{\lambda}\mathrm{Id} \end{pmatrix}$$

$$= (\tau+1)\mathbf{Z}$$

Therefore:

$$\mu^\top \Gamma^{-1} \mu - \frac{\tau}{\tau+1}\frac{\sigma^2}{\tau^2}(\mathbf{v}^\top \mathbf{F}\mathbf{v} + \mathbf{u}^\top \mathbf{G}\mathbf{u}) = \frac{1}{\tau+1}\begin{pmatrix} \mathbf{A}^{-1}\mathbf{a} \\ \mathbf{B}^{-1}\mathbf{b} \end{pmatrix}^\top \mathbf{Z}^{-1}\begin{pmatrix} \mathbf{A}^{-1}\mathbf{a} \\ \mathbf{B}^{-1}\mathbf{b} \end{pmatrix} \qquad (100)$$

The full exponential argument $\phi$ defined in Equation (99) is given by:

$$
\begin{aligned}
\phi &= \frac{1}{\tau+1}\left(\begin{pmatrix}\mathbf{A}^{-1}\mathbf{a}\\\mathbf{B}^{-1}\mathbf{b}\end{pmatrix}^{\top}\mathbf{Z}^{-1}\begin{pmatrix}\mathbf{A}^{-1}\mathbf{a}\\\mathbf{B}^{-1}\mathbf{b}\end{pmatrix}-\mathbf{a}^{\top}\mathbf{A}^{-1}\mathbf{a}-\mathbf{b}^{\top}\mathbf{B}^{-1}\mathbf{b}\right)\\
&= \frac{1}{\tau+1}\begin{pmatrix}\mathbf{a}\\\mathbf{b}\end{pmatrix}^{\top}\begin{pmatrix}\mathbf{A}^{-1}&0\\0&\mathbf{B}^{-1}\end{pmatrix}\left(\mathbf{Z}^{-1}-\begin{pmatrix}\mathbf{A}&0\\0&\mathbf{B}\end{pmatrix}\right)\begin{pmatrix}\mathbf{A}^{-1}&0\\0&\mathbf{B}^{-1}\end{pmatrix}\begin{pmatrix}\mathbf{a}\\\mathbf{b}\end{pmatrix}\\
&= \frac{1}{\tau+1}\begin{pmatrix}\mathbf{a}\\\mathbf{b}\end{pmatrix}^{\top}\begin{pmatrix}\mathbf{A}^{-1}&0\\0&\mathbf{B}^{-1}\end{pmatrix}\begin{pmatrix}-\mathbf{A}\mathbf{X}^{-1}\mathbf{A}&\mathbf{A}\mathbf{X}^{-1}\mathbf{B}\\\mathbf{B}\mathbf{X}^{-1}\mathbf{A}&-\mathbf{B}\mathbf{X}^{-1}\mathbf{B}\end{pmatrix}\begin{pmatrix}\mathbf{A}^{-1}&0\\0&\mathbf{B}^{-1}\end{pmatrix}\begin{pmatrix}\mathbf{a}\\\mathbf{b}\end{pmatrix}\\
&= \frac{1}{\tau+1}\begin{pmatrix}\mathbf{a}\\\mathbf{b}\end{pmatrix}^{\top}\begin{pmatrix}-\mathbf{X}^{-1}&\mathbf{X}^{-1}\\\mathbf{X}^{-1}&-\mathbf{X}^{-1}\end{pmatrix}\begin{pmatrix}\mathbf{a}\\\mathbf{b}\end{pmatrix}\\
&= -\frac{1}{\tau+1}(\mathbf{a}-\mathbf{b})^{\top}\mathbf{X}^{-1}(\mathbf{a}-\mathbf{b})\\
&= \frac{1}{\tau+1}\|\mathbf{a}-\mathbf{b}\|_{\mathbf{X}^{-1}}^2
\end{aligned}
$$

Substituting in (98) leads to:

$$
\begin{aligned}
m_{\pi} &\overset{\text{def}}{=} \pi(\mathbb{R}^d\times\mathbb{R}^d)\\
&= \sqrt{\det(\mathbf{H})}\left(\frac{m_{\alpha}m_{\beta}}{\sqrt{\det(\mathbf{AB})}}\right)^{\frac{1}{\tau+1}}\left(\frac{\sqrt{\det(\mathbf{FG})}}{\sigma^{2d}}\right)^{\frac{\tau}{\tau+1}}e^{-\frac{1}{2(\tau+1)}\left(\|\mathbf{a}-\mathbf{b}\|_{\mathbf{X}^{-1}}^2\right)}.
\end{aligned}
$$

The determinants can be easily expressed as functions of $\mathbf{C}$. First notice that:

$$
\det(\mathbf{H}) = \frac{1}{\det(\Gamma)} = \frac{\sigma^{4d}}{\det(\mathbf{FG}-\mathrm{Id})},
$$

and using the definition of $\mathbf{C}$, it holds that

$$
\mathbf{FG} = \widetilde{\mathbf{B}}\mathbf{C}^{-2}\widetilde{\mathbf{A}}.
$$

Therefore, $\det(\mathbf{FG}) = \frac{\det(\widetilde{\mathbf{A}}\widetilde{\mathbf{B}})}{\det(\mathbf{C})^2}$. Keeping in mind that the closed form expression of $\mathbf{C}$ given in (90) is applied to the pair $(\frac{1}{\tau}\widetilde{\mathbf{A}},\widetilde{\mathbf{B}})$ in the unbalanced case, it holds: $\mathbf{C}^2+\sigma^2\mathbf{C}=\frac{1}{\tau}\widetilde{\mathbf{A}}\widetilde{\mathbf{B}}$. Thus:

$$
\begin{aligned}
\mathbf{FG}-\mathrm{Id} &= \widetilde{\mathbf{B}}\mathbf{C}^{-2}\widetilde{\mathbf{A}}(\mathrm{Id}-\widetilde{\mathbf{A}}^{-1}\mathbf{C}^2\widetilde{\mathbf{B}}^{-1})\\
&= \widetilde{\mathbf{B}}\mathbf{C}^{-2}\widetilde{\mathbf{A}}(\mathrm{Id}-\widetilde{\mathbf{A}}^{-1}(\frac{1}{\tau}\widetilde{\mathbf{A}}\widetilde{\mathbf{B}}-\sigma^2\mathbf{C})\widetilde{\mathbf{B}}^{-1})\\
&= \widetilde{\mathbf{B}}\mathbf{C}^{-2}\widetilde{\mathbf{A}}(\frac{(1-\tau)}{\tau}\mathrm{Id}+\sigma^2\widetilde{\mathbf{A}}^{-1}\mathbf{C}\widetilde{\mathbf{B}}^{-1})\\
&= \sigma^2\widetilde{\mathbf{B}}\mathbf{C}^{-2}\widetilde{\mathbf{A}}(-\frac{2}{\gamma}\mathrm{Id}+\widetilde{\mathbf{A}}^{-1}\mathbf{C}\widetilde{\mathbf{B}}^{-1})\\
&= \sigma^2\widetilde{\mathbf{B}}\mathbf{C}^{-2}(-\frac{2}{\gamma}\widetilde{\mathbf{A}}\widetilde{\mathbf{B}}+\mathbf{C})\widetilde{\mathbf{B}}^{-1},
\end{aligned}
$$

therefore

$$
\det(\mathbf{FG}-\mathrm{Id}) = \sigma^{2d}\frac{\det((-\frac{2}{\gamma}\widetilde{\mathbf{A}}\widetilde{\mathbf{B}}+\mathbf{C})}{\det(\mathbf{C})^2}.
$$

Replacing the determinant formulas of $\mathbf{FG}$ and $\mathbf{FG} - \mathrm{Id}$ and re-arranging the common terms $\det(\mathbf{C})$ and $\sigma$ leads to:

$$
\begin{aligned}
\pi(\mathbb{R}^d \times \mathbb{R}^d) &= \frac{\left(m_\alpha m_\beta \sigma^{2d} \det(\mathbf{C}) \sqrt{\frac{\det(\widetilde{\mathbf{A}}\widetilde{\mathbf{B}})^\tau}{\det(\mathbf{A}\mathbf{B})}}\right)^{\frac{1}{\tau+1}}}{\sqrt{\frac{\det(\mathbf{C} - \frac{2}{\gamma}\widetilde{\mathbf{A}}\widetilde{\mathbf{B}})}{\sigma^{2d}}}} e^{-\frac{1}{2(\tau+1)}\left(\|\mathbf{a}-\mathbf{b}\|^2_{\mathbf{X}^{-1}}\right)} \\
&= \sigma^{d\left(\frac{2}{\tau+1}-1\right)} \frac{\left(m_\alpha m_\beta \det(\mathbf{C}) \sqrt{\frac{\det(\widetilde{\mathbf{A}}\widetilde{\mathbf{B}})^\tau}{\det(\mathbf{A}\mathbf{B})}}\right)^{\frac{1}{\tau+1}}}{\sqrt{\det(\mathbf{C} - \frac{2}{\gamma}\widetilde{\mathbf{A}}\widetilde{\mathbf{B}})}} e^{-\frac{1}{2(\tau+1)}\left(\|\mathbf{a}-\mathbf{b}\|^2_{\mathbf{X}^{-1}}\right)} \\
&= \sigma^{d\frac{1-\tau}{\tau+1}} \frac{\left(m_\alpha m_\beta \det(\mathbf{C}) \sqrt{\frac{\det(\widetilde{\mathbf{A}}\widetilde{\mathbf{B}})^\tau}{\det(\mathbf{A}\mathbf{B})}}\right)^{\frac{1}{\tau+1}}}{\sqrt{\det(\mathbf{C} - \frac{2}{\gamma}\widetilde{\mathbf{A}}\widetilde{\mathbf{B}})}} e^{-\frac{1}{2(\tau+1)}\left(\|\mathbf{a}-\mathbf{b}\|^2_{\mathbf{X}^{-1}}\right)} \\
&= \sigma^{\frac{d\sigma^2}{\sigma^2+\gamma}} \frac{\left(m_\alpha m_\beta \det(\mathbf{C}) \sqrt{\frac{\det(\widetilde{\mathbf{A}}\widetilde{\mathbf{B}})^\tau}{\det(\mathbf{A}\mathbf{B})}}\right)^{\frac{1}{\tau+1}}}{\sqrt{\det(\mathbf{C} - \frac{2}{\gamma}\widetilde{\mathbf{A}}\widetilde{\mathbf{B}})}} e^{-\frac{1}{2(\tau+1)}\left(\|\mathbf{a}-\mathbf{b}\|^2_{\mathbf{X}^{-1}}\right)}
\end{aligned}
\tag{101}
$$

**Deriving a closed form for** $\mathrm{UOT}_\sigma$. Using Equation (101), a direct application of Proposition 7 yields

$$
\mathrm{UOT}_\sigma(\alpha, \beta) = \gamma(m_\alpha + m_\beta) + 2\sigma^2(m_\alpha m_\beta) - 2(\sigma^2 + 2\gamma)m_{\pi^\star}.
\tag{102}
$$

This ends the proof of Theorem 3.