[Reviews · NeurIPS 2020]

Review 1

Summary and Contributions: This paper shows that entropy-regularized optimal transport between two Gaussian measures has a closed form, and presents an explicit form for it. Moreover, they provide the closed form for the corresponding barycenter problem as well as the entropy regularized OT problem for unbalanced measures.

Strengths: Optimal transport has gain a lot of interest in recent years. Up to now, closed form solutions only exists for a rather small set of measures. The main contribution of this paper is having a closed form for the entropy-regularized problem as well. The contribution is the extension of the fundamental understanding of the problem. The reviewer thinks this results will become classical for the literature.

Weaknesses: The paper is well written and the results are fundamental. However, the authors fail to bring the result to their impact of the current state of OT. Some discussion is missing on how having a closed form solution could lead to better algorithms or better analysis.

Correctness: Yes, the reviewer checked the proofs and they are correct.

Clarity: The paper is well written, with detailed explanations and context.

Relation to Prior Work: Yes, the paper puts their results in context with existing literature.

Reproducibility: Yes

Additional Feedback: After the rebuttal I have read the rebuttal from the authors. They replied to my comment. I still feel the result of this paper is important, but the discussion about the connections with the implications for ML community in general is lacking. Still this is a very good paper. I will update my score accordingly. %%%%%%%%%%%%%%%%%%%%%%%%%%%%%%%%%%555 The paper is well written and explained. My only suggestion is to discussed further the implications of the result for the community of Optimal transport research. Does having this closed form solution provide better ways to design algorithms, faster algorithms, etc? Line 110: Please define what is a centered measure.


Review 2

Summary and Contributions: This paper gives closed-form solution for an entropically-regularized unbalanced OT between Gaussians problem.

Strengths: The significance of the contribution is that - The parameters in the closed-form-solution give intuition for how the different parameters in unbalanced-OT interact with each other. Unbalanced OT is kind of hard to think about, even though for many applications it is clear that balanced OT is pretty much nonsense. So I think this intuition-giving aspect is pretty important. - Having a "ground truth" solution to this problem gives us a new way to test the limitations of the usual sinkorn-on-empirical-distribution approach. - Conceivably, it could be used to help accelerate more general unbalanced OT problems (if the marginals can be understood as mixture of gaussians and... if something clever could be done to use this result to say anything remotely useful about mixtures of Gaussians.)

Weaknesses: There are a few ways to do unbalanced OT. For example, you can do it by controlling the total variation between the target marginals and the transport marginals, instead of the KL. If the paper could show the formula for that case that would be supercool. Though maybe there isn't a closed-form for that. Worth looking into. Also, I think the authors could skip the balanced-OT section and move a lot of propositions to the appendix. This would give more room for some more empirical studies. Specificially, here's an empirical study I'd like to see: 1) you're given samples from p and samples from q 2) you use unbalanced sinkhorn iterates to get the best transport plan for the empirical distributions 3) how close is your transport plan to the best transport plan from p to q? This is a question that always makes everyone a little nervous, and having some examples to show where it works and where it doesn't would be really valueable to the field. How many samples do you "need" before you stop making huge mistakes? With this closed form solution it would be easy to simulate this as the ambient dimension explodes, which might be pretty informative. Post-author-feedback-feedback: neato! Thanks for running that extra sim. That's pretty cool.

Correctness: It all looks right to me.

Clarity: It is clear.

Relation to Prior Work: They discuss how it connects to prior work pretty well.

Reproducibility: Yes

Additional Feedback:


Review 3

Summary and Contributions: In this paper, the authors show that the entropy-regularized Wasserstein distance between two Gaussian measures can be computed with a closed-form formula (the optimal transport plan is a Gaussian measure on the product, with explicit parameters). Based on this formula, they are also able to provide an explicit expression for the barycenter between K Gaussian measures (for the Sinkhorn divergence, i.e. debiased Wasserstein distance). Finally, the authors extend these formulae to the unbalanced case. They also provide a few numerical experiments to validate the theoretical results.

Strengths: The authors solve a boiling question, with a formula that will be useful for a very large community. The method to prove this formula is both natural and elegant (expressing the Sinkhorn iterations directly on the parameters of Gaussian measures, and explicitly solving the fixed point equation). The proof is decomposed in several intermediate steps, in a very clear manner. Even if it contains many technical details, a reader who is not interested in the details can very easily focus on the main theorems, and understand the formulae that she/he may need. This will be really useful for practitioners. The authors also obtain Propositions 4, 5, 6 about convexity, duality, and differentiability as by-products of their main theorems. These results are also interesting in themselves (which thus provides a useful generalization of the Wasserstein-Bures distance on matrices). The extension to the unbalanced case is also of interest, and helps to get an intuition on how the mass is either transported or re-created depending on the parameters (according to Remark 2).

Weaknesses: - This paper is quite dense (with nearly 20 pages of proof in supplementary material). One can very well wonder why such a paper is not submitted to a journal. - The only (small) weakness of the paper is that the last section about numerical experiments is too short, and thus difficult to understand: -> "Figure 1 illustrates the convergence"... the convergence of what? -> Figure 2 is also difficult to understand. Both source and target measures are sampled, and then Sinkhorn algorithm is used on a discretized measure?

Correctness: Everything I've read seems correct. I have checked the part of the appendix concerning the balanced case. The main proofs seems correct, but there may be a few typos (see below). So the authors are invited to correct possible typos in the appendix. I have not checked the proof of the unbalanced case. I have a minor concern in the proof of Proposition 2: in order to show that the Sinkhorn iterations are contractive, the authors show that the norm of d\phi(X) is < 1 everywhere, but do not show that it is bounded by a uniform constant c < 1. Could the authors explain why the uniform bound is not necessary here? Anyway, the study of the sequence (F_n, G_n) is not mandatory since the authors explicitly solve the fixed point equation on matrices (and thus show that there is indeed a unique solution). Also, on l. 423, I think the authors could better explain why AB and C have same eigenvectors (the fact that C has a diagonal decomposition can be used earlier). At the top of page 17, the authors can add a sentence to explain the following calculation.

Clarity: Globally, this is a very clear paper. I have just a small comment on the introduction of the unbalanced case: since the measures alpha, beta are not normalized in (32), the Kullback-Leibler divergence should be re-defined exactly here (and not after Equation (33)). Also, the sentence l. 201 "Moreover, the objective admits a lower bound if and only if ..." has not a clear meaning. Does it mean that the objective is finite as soon as \pi has a density. Or that there is a uniform lower bound for all \pi with L2 density?

Relation to Prior Work: The introduction clearly states the relation to prior works. First, the authors recall the case when the OT distance can be computed in closed form (univariate measures, or Gaussian measures). They also recall approximations (sliced Wassertein, or Gaussian mixture transport) which rely on these two first cases. The authors could have also cited a recent paper that studies in more depth the Gaussian mixture case: https://arxiv.org/abs/1907.05254 The author also properly refers to the work by Gerolin et al [24] which states a simplified result in the one-dimensional case. However, this article should already be cited in the introduction.

Reproducibility: Yes

Additional Feedback: - The fact that all proofs can be found in appendix could be written earlier in the paper. - Could the authors add a reference for the dual formulation of (9)? If spaces X,Y are not assumed to be compact, the existence of the maximum may not be so easy to see. (Is the result of [11] sufficient to show that the maximum exists?) - In Equation (6), the expression seems to be a differential, and the identification to a gradient is not so obvious (depends on the scalar product). - As noticed in Remark 1, Gaussian measures may be defined for singular matrices. So it would be useful to recall in Theorem 1 the hypothesis that A, B should be non singular. - l. 149: The authors say that a sequence is contractive, but for which distance? Typos: l.64 ground truth value ? l.147 the indices in equation (17) should be changed according to Equation (407): G_{n+1} = F_n^{-1} + \sigma^2 B^-1 F_{n+1} = G_{n+1}^{-1} + \sigma^2 A^{-1} l.176 statement using for Gaussians Typos (appendix) l.360 missing ref l.401 (in the equations: Q(\sigma^2 Id) should be Q(\sigma^{-2} Id) ) l.402 Q(V_n) = \frac{g_n}{2 \sigma^2} l.412 The first differential l.448 This equation is not the definition of B_\sigma^2 l.450 there exists l.477 Could the authors add a reference for Danskin's theorem? l.479 The equation seems to have a missing term l.523 g_k and h_{\beta} are actually used in the previous equation (65) l.526 equation (66) index \beta missing on h (same in equation (67) ) Reference [5] and [6] are the same Conference missing in [21] and [33]

[Author Response · NeurIPS 2020]

We thank the reviewers for their appreciative and thoughtful feedback.

**Reviewer 1.** *"However, the authors fail to bring the result to their impact of the current state*
*of OT [...[ Does having this closed form solution provide better ways to design algorithms, faster*
*algorithms, etc?"* As discussed in L59-L73, the first obvious impact of our contribution is that it
provides the first example for which regularized (unbalanced) OT admits a closed form expression.
These formulas provide a testbed for any theoretical conjecture that tries to understand better entropic
OT, or any novel stochastic optimization algorithm designed to compute it faster. Additionally, our
formulas offer a principled solution to alleviate the differentiability issues of the Bures metric that
arise for singular matrices (L.129-130). Finally, one can foresee that applications relying on entropic
OT might benefit from some local Gaussian approximations to use these closed form, in the spirit of
sliced Wasserstein approaches. We will further emphasize these aspects. *"Line 110: Please define*
*what is a centered measure."* A measure with 0 mean. We will clarify this.

**Reviewer 2.** *"If the paper could show the formula for that case [TV] that would be*
*supercool. Though maybe there isn't a closed-form for that."* A glance at the prox-
div operator of TV (https://arxiv.org/pdf/1607.05816.pdf) shows that after the first iter-
ation, the (log) dual variable would be the pointwise projection of a quadratic func-
tion over the box $\left[-\frac{\lambda}{\varepsilon}, \frac{\lambda}{\varepsilon}\right]$ which is not obvious to convolve with a Gaussian kernel.

*"[Transport plan experiment] How many samples do you "need" before you stop making huge mistakes? (...) as the ambient dimension explodes"* To answer this question we computed the distance between the ground truth (formulas of Thm 3) and the empirical moments $(\mu_n, \Sigma_n)$ of the transport plan using random inputs and fixed parameters $\sigma = 0.01, \gamma = 0.2, m_\alpha = 1, m_\beta = 1.1$. See Fig.1 on left.

27 Figure 1: Large dimensions need more samples to approxi-
28 mate the moments of the unbalanced optimal transport plan.

29 **Reviewer 3.** *""Figure 1 illustrates the convergence"... the convergence of what?"* This conver-
30 gence is in terms of number of samples from 2 Gaussian distributions: $\lim_{n \to +\infty} \mathrm{OT}_\sigma(\alpha_n, \beta_n) \to$
31 $\mathrm{OT}_\sigma(\alpha, \beta)$. We will clarify this. *"Figure 2 is also difficult to understand. Both source and*
32 *target measures are sampled, and then Sinkhorn algorithm is used on a discretized measure?"*
33 Exactly, both measures are sampled, we run Sinkhorn to obtain an empirical transportation plan that
34 we visualize by computing a histogram on a uniform 2D grid. *"[On the proof of prop 2] Could*
35 *the authors explain why the uniform bound is not necessary here?"* Thank you for pointing this
36 out, a uniform bound is indeed required and we will update our proof. From (42) and using Weyl's
37 inequality, we can bound the smallest eigenvalue of $\mathbf{F}_n$ from below: $\forall n, \lambda_d(\mathbf{F}_n) \geq \frac{\sigma^2}{\lambda_1(\mathbf{A})}$ (where
38 $\lambda_d(\mathbf{F})$ is the smallest eigenvalue of $\mathbf{F}$ and $\lambda_1(\mathbf{A})$ is the biggest eigenvalue of $\mathbf{A}$). Hence, the iterates
39 live in $\mathcal{A} \stackrel{\mathrm{def}}{=} \mathcal{S}_{++}^d \cap \{\mathbf{X} : \lambda_d(\mathbf{X}) \geq \frac{\sigma^2}{\lambda_1(\mathbf{A})}\}$. Finally, for all $\mathbf{X} \in \mathcal{A}$, $\|(\mathrm{Id} + \sigma^2 \mathbf{B}^{-\frac{1}{2}} \mathbf{X} \mathbf{B}^{-\frac{1}{2}})^{-1}\|_{\mathrm{op}} =$
40 $\frac{1}{\lambda_d(\mathrm{Id} + \sigma^2 \mathbf{B}^{-1/2} \mathbf{X} \mathbf{B}^{-1/2})} = \frac{1}{1+\sigma^2 \lambda_d(\mathbf{B}^{-1/2} \mathbf{X} \mathbf{B}^{-1/2})} \leq \frac{1}{1+\sigma^2 \lambda_d(\mathbf{B}^{-1}) \lambda_d(\mathbf{X})} \leq \left(1 + \frac{\sigma^4}{\lambda_1(\mathbf{B}) \lambda_1(\mathbf{A})}\right)^{-1}$.
41 Which proves the uniform bound. *"on l. 423, I think the authors could better explain why AB*
42 *and C have same eigenvectors"* Because $AB$ is a quadratic polynomial of $C$ (Eq 21). As explained
43 in L157-158, C has positive eigenvalues, writing its EVD as: $C = QDQ^{-1}$ in eq (21) leads to:
44 $Q(D^2 + \sigma^2 D)Q^{-1} = AB$ which is an EVD of $AB$. AB and C have the same eigenvectors $Q$. In
45 the appendix, we wrote EVD of AB instead of C, we will correct this. *"l. 201 "Moreover, the*
46 *objective admits a lower bound if and only if ..." has not a clear meaning"* This is the condition
47 for the entropy KL to be finite. We will replace the previous statement. *"Could the authors add a*
48 *reference for the dual formulation of (9)? [...] is [11] enough [...]?"* We have cited [33], the context
49 of [11] is more restrictive since they considered probability spaces (X, Y with reference measures
50 having unit mass). *"In Equation (6), [...] depends on the scalar product"* the gradient is given for
51 the Frobenius inner product. We will clarify this point. *"- l. 149: The authors say that a sequence*
52 *is contractive, but for which distance?"* In the matrix operator norm, as per the proof of Proposition
53 2. We will explicitly state this.



[Meta-Review · NeurIPS 2020]

Reviewers agree that this is a valuable and substantial contribution to the literature. While there is some question about whether such a paper fits in the conference page limit, the AC sees no conflict here -- the main result can be (and is) stated in the paper, and proofs found in the appendix.